

# Coupled climate-carbon cycle simulation of the Last Glacial Maximum atmospheric CO$_2$ decrease using a large ensemble of modern plausible parameter sets

Krista M.S. Kemppinen[1], Philip B. Holden[2], Neil R. Edwards[2], Andy Ridgwell[3,4], Andrew D. Friend[1]

[1]Department of Geography, University of Cambridge, Cambridge CB2 3EN, UK
[2]Environment, Earth and Ecosystem Sciences, The Open University, Milton Keynes, MK7 6AA, UK
[3]School of Geographical Sciences, Bristol University, Bristol BS8 1SS, UK
[4]Department of Earth Sciences, University of California, Riverside, California 92521, USA

*Correspondence to*: Krista M.S. Kemppinen (Krista.Kemppinen@asu.edu)

**Abstract.** During the Last Glacial Maximum (LGM), atmospheric CO$_2$ was around 90 ppmv lower than during the preindustrial period. Despite years of research, however, the exact mechanisms leading to the glacial atmospheric CO$_2$ drop are still not entirely understood. Here, a large (471-member) ensemble of GENIE-1 simulations is used to simulate the equilibrium LGM minus preindustrial atmospheric CO$_2$ concentration difference ($\Delta$CO$_2$). The ensemble has previously been weakly constrained with modern observations and was designed to allow for a wide range of large-scale feedback response strengths. Out of the 471 simulations, 315 complete without evidence of numerical instability, and with a $\Delta$CO$_2$ that centres around -20 ppmv. Roughly a quarter of the 315 runs predict a more significant atmospheric CO$_2$ drop, between $\sim$30 and 90 ppmv. This range captures the error in the model's process representations and the impact of processes which may be important for $\Delta$CO$_2$ but are not included in the model. These runs jointly constitute what we refer to as the "plausible glacial atmospheric CO$_2$ change-filtered (PGACF) ensemble".

Our analyses suggest that decreasing LGM atmospheric CO$_2$ tends to be associated with decreasing SSTs, increasing sea ice area, a weakening of the Atlantic Meridional Overturning Circulation (AMOC), a strengthening of the Antarctic Bottom Water (AABW) cell in the Atlantic Ocean, a decreasing ocean biological productivity, an increasing CaCO$_3$ weathering flux, an increasing terrestrial biosphere carbon inventory and an increasing deep-sea CaCO$_3$ burial flux. The increases in terrestrial biosphere carbon are predominantly due to our choice to preserve rather than destroy carbon in ice sheet areas. However, the ensemble soil respiration also tends to decrease significantly more than net photosynthesis, resulting in relatively large increases in non-burial carbon. In a majority of simulations, the terrestrial biosphere carbon increases are also accompanied by decreases in ocean carbon and increases in lithospheric carbon. In total, however, we find there are 5 different ways of achieving a plausible $\Delta$CO$_2$ in terms of the sign of individual carbon reservoir changes. The PGACF ensemble members also predict both positive and negative changes in global particulate organic carbon (POC) flux, AMOC and AABW cell strengths, and global CaCO$_3$ burial flux.



Comparison of the PGACF ensemble results against observations suggests that the simulated LGM physical climate and biogeochemical changes are mostly of the right sign and magnitude or within the range of observational error, except for the change in global deep-sea $CaCO_3$ burial flux – which tends to be overestimated. We note that changing $CaCO_3$ weathering flux is a variable parameter (included to account for variation in both the $CaCO_3$ weathering rate and the un-modelled $CaCO_3$ shallow water deposition flux), and this parameter is strongly associated with changes in global $CaCO_3$ burial rate. The increasing terrestrial carbon inventory is also likely to have contributed to the LGM increase in deep-sea $CaCO_3$ burial flux via the process of carbonate compensation. However, we do not yet rule out either of these processes as causes of $\Delta CO_2$ since missing processes such as Si fertilisation, Si leakage and the effect of decreasing SSTs on $CaCO_3$ production may have introduced a high LGM global $CaCO_3$ burial rate bias. Including these processes would, all else held constant, lower the rain ratio seen by the sediments and result in a decrease in atmospheric $CO_2$ and increase in ocean carbon. Despite not modelling $\Delta^{14}C_{atm\,(DIC)}$ and $\delta^{13}C_{atm\,(DIC)}$, we also highlight some ways in which our results may potentially be reconciled with these records.

## 1 Introduction

Analyses of Antarctic ice core records suggest that the atmospheric $CO_2$ concentration at the Last Glacial Maximum (LGM), about 21 kyr ago, was around 190 ppmv, well below the preindustrial atmospheric concentration of around 280 ppmv. The most commonly accepted mechanisms to explain the atmospheric $CO_2$ decrease include lower sea surface temperatures (Martin et al., 2005; Menviel et al., 2012), iron fertilisation (Bopp et al., 2003; Oka et al., 2011; Jaccard et al., 2013; Ziegler et al., 2013; Martínez-Garcia et al., 2014; Lambert et al., 2015), sea-ice capping of air-sea gas exchange (Stephens and Keeling, 2000; Sun and Matsumoto, 2010; Chikamoto et al., 2012) and ocean circulation/stratification changes (Adkins et al., 2002; Lynch-Stieglitz et al., 2007; Skinner et al., 2010; Lippold et al., 2012; Gebbie, 2014; Skinner et al., 2014; Tiedemann et al., 2015; de la Fuente et al., 2015; Freeman et al., 2015), due to a range of possible mechanisms such as increased brine rejection (Shin et al., 2003; Bouttes et al., 2010, 2011; Zhang et al., 2013; Ballarotta et al., 2014), a shift in/weakening of the westerly wind belt over the Southern Ocean (Toggweiler et al., 2006; Anderson et al., 2009; Völker and Köhler, 2013), stronger westerly winds over the North Atlantic (Muglia and Schmittner, 2015), and a reduced or reversed buoyancy flux from the atmosphere to the ocean surface in the Southern Ocean (Watson and Garabato, 2006; Ferrari et al., 2014). A process that is conversely assumed to have contributed to increasing atmospheric $CO_2$ is increasing salinity and ocean total dissolved inorganic carbon (DIC) concentration in response to decreasing sea level (Ciais et al., 2013).

A dominant assumption is also that the terrestrial biosphere carbon inventory was reduced (Crowley et al., 1995; Adams and Faure, 1998; Ciais et al., 2012; Peterson et al., 2014), in line with independent estimates of an ocean carbon inventory that was

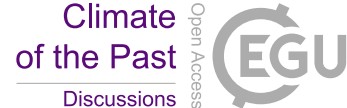



enhanced by several hundred petagrams (Goodwin and Lauderdale, 2013; Sarnthein et al., 2013; Allen et al., 2015; Skinner et al., 2015; Schmittner and Somes, 2016). The decrease in terrestrial carbon is generally attributed to unfavourable climatic conditions for photosynthesis, and the destruction of organic material by moving ice sheets (e.g. Otto et al., 2002; Prentice et al., 2011; Brovkin et al., 2012; O'ishi and Abe-Ouchi, 2013). The hypothesis that there was an increase in terrestrial carbon

has, however, also been put forward (e.g. Zeng, 2003; Zimov, 2006), with some studies additionally suggesting little net change (e.g. Brovkin and Ganopolski, 2015). Processes proposed to be responsible for the terrestrial carbon increase include slower soil respiration rates, continental shelf regrowth, and the preservation rather than destruction of terrestrial biosphere carbon in areas to be covered by the expanding Laurentide and Eurasian ice sheets (Weitemeyer and Buffett, 2006; Franzén and Cropp, 2007; Zeng et al., 2007; Zimov et al., 2009; Zech et al., 2011).

Other mechanisms which may have affected the LGM atmospheric $CO_2$ change include changes in carbonate weathering rate (Munhoven, 2002; Jones et al., 2002; Foster and Vance, 2006; Vance et al., 2009; Brovkin et al., 2012; Crocket et al., 2012; Lupker et al., 2013; Simmons et al., 2016), dissolved organic carbon inventory (Ma and Tian, 2014), reduced shallow water carbonate deposition (Opdyke and Walker, 1992; Kleypas et al., 1997; Brovkin et al., 2007), reduced marine bacterial

metabolic rate (Matsumoto et al., 2007; Roth et al., 2014), silicic acid leakage (Matsumoto et al., 2002, 2014), Si fertilisation (Harrison, 2000; Tréguer and Pondaven, 2000) and increased oceanic $PO_4$ inventory (Tamburini and Follmi, 2009; Wallmann, 2014, 2015).

To investigate the causes of the LGM atmospheric $CO_2$ drop, this study utilises an ensemble of sets of parameters which are

thought to contribute to variability of atmospheric $CO_2$ on glacial/interglacial timescales. The ensemble is a modified version of the emulator-filtered plausibility-constrained (EFPC) ensemble of Holden et al. (2013a), using the GENIE-1 EMIC, in its coupled climate-carbon cycle configuration. The philosophy behind the design of the ensemble was initially outlined in Holden et al. (2010a): by applying only weak constraints to the ensemble parameters and the modern climate states accepted as plausible, a wide range of large-scale feedback response strengths is allowed for, in an attempt to encompass the range of

behaviour exhibited by higher-resolution multi-model ensembles (Holden et al., 2010a; Edwards et al 2011; Holden et al., 2013a). It is recognised, however, that what constitutes an implausible or plausible climate state is somewhat subjective, and that not all sources of model uncertainty, such as the unknown error due to interacting ice sheets and ocean circulation, or the unknown error due to potential $\Delta CO_2$ mechanisms not included in the model (although see below for some estimates), are accounted for.





## 2 Methods

### 2.1 The model

The GENIE-1 configuration is as described in Holden et al. (2013a). The physical model consists of a three-dimensional

frictional geostrophic ocean model (GOLDSTEIN) coupled to a thermodynamic/dynamic sea ice model (Edwards and

Marsh, 2005; Marsh et al., 2011) and a two-dimensional Energy-Moisture Balance Model (EMBM). Atmospheric tracers are

a sub-component of the EMBM, with a simple module (ATCHEM) used to store the concentration of atmospheric gases and

their relevant isotopic properties (Lenton et al., 2007). The model land surface physics and terrestrial carbon cycle are

represented by ENTS (Williamson et al., 2006). The ocean biogeochemistry model (BIOGEM) is as described in Ridgwell et

al. (2007) but includes a representation of iron cycling (Annan and Hargreaves, 2010), and the biological uptake scheme of

Doney et al. (2006). The model sediments are represented by SEDGEM (Ridgwell and Hargreaves, 2007). GENIE-1 also

includes a land surface weathering model, ROKGEM (Colbourn, 2011), which redistributes prescribed weathering fluxes

according to a fixed river-routing scheme. The model is on a 36 x 36 equal-area horizontal grid, with 16 vertical levels in the

ocean.

### 2.2 The ensemble

The GENIE-1 ensemble consists of 471 parameter sets, varying 29 key model parameters over the ranges in Table 1. It

derives from the 471-member EFPC ensemble of Holden et al. (2013a), which varies 24 active parameters and 1 dummy

parameter (as a check against over-fitting).  The parameter values in Holden et al. (2013a) were derived by building

emulators of eight preindustrial climate metrics and applying a rejection sampling method known as approximate bayesian

computation (ABC) to find parameter sets that the emulators predicted were modern plausible. Two parameters were later

added to the EFPC ensemble in Holden et al. (2013b) to describe the un-modelled response of clouds to global average

temperature change (OL1), and the uncertain response of photosynthesis to changing atmospheric $CO_2$ concentration (VPC).

We add two further parameters here that represent uncertain processes specific to the LGM. The first (FFX) scales ice-sheet

meltwater fluxes to account for uncertainty in un-modelled isostatic depression at the ice-bedrock interface due to ice sheet

growth (Holden et al., 2010b). The second (GWS) scales the global average carbonate preindustrial weathering rates for the

LGM, to account for uncertainty in carbonate weathering and un-modelled shallow water carbonate deposition rate changes.

For both FFX and GWS, uniform random values were derived using the generation function runif in R.

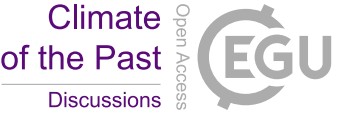



**Table 1. Ensemble parameters.** Ranges are from (a) Holden et al. (2013a), (b) Holden et al. (2013b), and (c) Holden et al. (2010b), with the exception of GWS (see main text). The table also precludes the dummy parameter.

| Module | Code | Description | Range | Ref. |
|---|---|---|---|---|
| EMBM | AHD | Atmospheric heat diffusivity ($m^2\ s^{-1}$) | 1118875 to 4368143 | a |
| | AMD | Atmospheric moisture diffusivity ($m^2\ s^{-1}$) | 50719 to 2852835 | a |
| | APM | Atlantic-Pacific moisture flux scaling | 0.1 to 2.0 | a |
| | OL0 | Clear skies OLR reduction ($W\ m^{-2}$) | 2.6 to 10.0 | a |
| | OL1 | OLR feedback ($W\ m^{-2}\ K^{-1}$) | -0.5 to 0.5 | b |
| GOLDSTEIN | OHD | Isopycnal diffusivity ($m^2\ s^{-1}$) | 312 to 5644 | a |
| | OVD | Reference diapycnal diffusivity ($m^2\ s^{-1}$) | 0.00002 to 0.0002 | a |
| | OP1 | Power law for diapycnal diffusivity depth profile | 0.008 to 1.5 | a |
| | ODC | Ocean inverse drag coefficient (days) | 0.5 to 5.0 | a |
| | WSF | Wind scale factor | 1.0 to 3.0 | a |
| | FFX | Freshwater flux scaling factor | 1.0 to 2.0 | c |
| SEA-ICE | SID | Sea ice diffusivity ($m^2\ s^{-1}$) | 5671 to 99032 | a |
| ENTS | VFC | Fractional vegetation dependence on vegetation carbon density ($m^2\ kgC^{-1}$) | 0.4 to 1.0 | a |
| | VBP | Base rate of photosynthesis ($kgC\ m^{-2}\ yr^{-1}$) | 3.0 to 5.5 | a |
| | VRA | Vegetation respiration activation energy ($J\ mol^{-1}$) | 24211 to 71926 | a |
| | LLR | Leaf litter rate ($yr^{-1}$) | 0.08 to 0.3 | a |
| | SRT | Soil respiration activation temperature (K) | 198 to 241 | a |
| | VPC | Photosynthesis half-saturation to $CO_2$ (ppmv) | 30 to 697 | b |
| BIOGEM | PHS | $PO_4$ half-saturation concentration ($mol\ kg^{-1}$) | 5.3e-8 to 9.9e-7 | a |
| | PRP | Initial proportion of POC export as recalcitrant fraction | 0.01 to 0.1 | a |
| | PRD | e-folding remineralisation depth of non-recalcitrant POC (m) | 106 to 995 | a |
| | RRS | Rain ratio scalar | 0.02 to 0.1 | a |
| | TCP | Thermodynamic calcification rate power | 0.2 to 2.0 | a |
| | PRC | Initial proportion of $CaCO_3$ export as recalcitrant fraction | 0.1 to 1.0 | a |
| | CRD | e-folding remineralisation depth of non-recalcitrant $CaCO_3$ (m) | 314 to 2962 | a |
| | FES | Iron solubility | 0.001 to 0.01 | a |
| | ASG | Air-sea gas exchange parameter | 0.1 to 0.5 | a |
| ROKGEM | GWS | Land-to-ocean bicarbonate flux scaling factor | 0.5 to 1.5 | n/a |



## 2.3 Experimental set-up of the model

The preindustrial ensemble simulation results were repeated to verify reproducibility of Holden et al. (2013a). The plausibility metrics are summarised in Table 2. The simulations were performed in two stages, each lasting 10 kyr, on the Cambridge High Performance Computing (HPC) Cluster Darwin. The first stage involved spinning up the model with atmospheric $CO_2$

concentration relaxed to 278 ppmv and a closed biogeochemistry system. The $CaCO_3$ weathering flux is diagnosed in the model to balance the modelled $CaCO_3$ burial rate and conserve alkalinity. In the second stage, atmospheric $CO_2$ was allowed to evolve with interacting oceans and sediments, applying a fixed $CaCO_3$ weathering rate diagnosed from the end of stage 1. To allow the sediments to reach equilibrium as fast as possible, no bioturbation was modelled in either stage 1 or stage 2.

Each parameter set was then applied to LGM simulations. The modelled preindustrial equilibrium states were used as initial conditions and the ensemble members were integrated for 10 kyr, with freely evolving $CO_2$. These 10 kyr simulations are variously referred to here as the "LGM equilibrium simulation" or "stage 3", and the LGM equilibrium state refers to the end of stage 3, unless otherwise specified. Thus, $\Delta CO_2$, for example, corresponds to end of stage 3 minus end of stage 2 atmospheric $CO_2$ concentration. The ensemble was integrated for another 10 kyr after stage 3 (yielding 20 kyr of LGM climate in total) to

simply verify, by analysing a subset of the ensemble, that the sediments (being the slowest component in the model) were in equilibrium by 10 kyr. These next 10 kyr of LGM simulation are referred to as "stage 4".

Boundary conditions applied in the LGM simulations included orbital parameters (Berger, 1978) and aeolian dust deposition fields (Mahowald et al., 2006). The model also requires a detrital flux field to the sediments, containing contributions from

opal and material from non-aeolian sources (Ridgwell and Hargreaves, 2007). The representation of the ice sheets is as described in Holden et al. (2010b), using the terrestrial ice sheet fraction and orography from the ICE-4G reconstruction of Peltier (1994). Rather than initialising the ensemble with the ice sheet extent and orography at 21 kyr BP, the ice sheets are configured to grow from their preindustrial to LGM extent in 1 kyr, at the beginning of the LGM simulation (i.e. years 0-1 kyr) in order to account for the impact of sea level change on ocean tracers. An important assumption is that the preindustrial

terrestrial carbon is preserved beneath the LGM ice sheets, though it is allowed to interact with the atmosphere prior to burial. Sensitivity simulations were performed to verify the simulated equilibrium state is insensitive to the choice of timescale of ice-sheet growth. Weathering fluxes from the preindustrial simulation were applied, scaled by GWS.

A distinctive feature of the simulations is that the atmospheric $CO_2$ used in the radiative code is internally generated, rather

than prescribed as in e.g. Tagliabue et al. (2009); Bouttes et al. (2011); Brovkin et al. (2012); Brovkin and Ganopolski (2015); Crichton et al. (2016). The radiative forcing from dust, and gases other than $CO_2$ was neglected.





An error in the experimental set-up precludes analysis of simulated carbon isotope tracers. We acknowledge that an evaluation of the simulated $\Delta^{14}C_{atm\,(DIC)}$ and $\delta^{13}C_{atm\,(DIC)}$ against observations would have provided useful constraints (see discussion in conclusions section).

## 3 Preindustrial simulations

### 3.1 Ensemble filtering

After application to stages 2 and 3, the 471-member ("EFPC") ensemble was filtered to 315 simulations, being those with a stage 2 atmospheric $CO_2$ concentration in the range 268 to 288 ppmv (c.f. Prentice et al., 2001), which did not enter a snowball Earth state in stage 3 and which did not evidence numerical instability (c.f. Holden et al., 2013b). These 315 simulations comprise the "EFPC2" ensemble, and form the basis of all the results reported in this study, unless otherwise specified. Individual ensemble members are referred to by their "run IDs", corresponding to the 1000 emulator-filtered parameter sets that were originally applied to GENIE-1 (Holden et al., 2013a).

### 3.2 Modern plausibility and other global metrics

EFPC2 preindustrial atmospheric $CO_2$ has a mean concentration of $278.1 \pm 1.3$ ppmv (standard deviation). The ensemble mean and range for the eight modern climate plausibility metrics are shown in Table 2, and compared against the results of the 471-member EFPC ensemble (Holden et al., 2013a). Metrics are also reported separately for the annual average global ocean carbon inventory, sea surface temperature and sea ice area, and compared with observations (Table 3). The ensemble mean ocean carbon inventory is close to the 36,000 PgC equilibrium preindustrial ocean carbon inventory predicted by GENIE-1 in Lenton et al. (2006), below reconstructed estimates of ca. 38,000 PgC (Houghton et al., 1990), largely attributable to an underestimated ocean volume at our low resolution (Lenton et al 2006). The ensemble mean SST exceeds observations but the error is still comparable to that associated with previous model predictions (e.g. Kim et al., 2003). The ensemble mean sea ice area (SIA) lies within the range of observed estimates.





**Table 2. The EFPC and EFPC2 ensembles modern climate plausibility metrics**. The EFPC2 ensemble values are as reported in Holden et al. (2013a), Table 2. All values are annual averages, except for the Antarctic sea ice area, which is the end-of-year value. The ensemble means are presented as the mean plus minus one standard deviation.

| | EFPC2 Ensemble mean | EFPC2 Ensemble range | EFPC Ensemble mean | EFPC Ensemble range |
|---|---|---|---|---|
| Global SAT (°C) | 13.7 ± 1.1 | 11.8 to 16.2 | 13.6 ± 1.1 | 11.7 to 16.2 |
| Atlantic overturning stream function maximum (Sv) | 16.4 ± 3.8 | 4.5 to 27.3 | 17.5 ± 3.2 | 10.0 to 25.8 |
| Atlantic overturning stream function minimum (Sv) | -4.1 ± 1.0 | -6.8 to -0.8 | -4.1 ± 1.0 | -6.8 to -1.0 |
| 31/12 Antarctic sea ice area (million $km^2$) | 6.7 ± 2.7 | 1.2 to 13 | 6.8 ± 2.8 | 1.2 to 12.9 |
| Global VegC (PgC) | 499.9 ± 94.5 | 328.6 to 765.5 | 492 ± 94 | 326 to 762 |
| Global SoilC (PgC) | 1329.7 ± 279.2 | 896.1 to 2353.2 | 1351 ± 308 | 896 to 2430 |
| wt% $CaCO_3$ | 34.4 ± 7.9 | 19.1 to 51.5 | 34.1 ± 7.8 | 20.0 to 50.0 |
| Global ocean $O_2$ ($\mu m$ $kg^{-1}$) | 164.1 ± 19.3 | 121.8 to 217.5 | 165 ± 20 | 117 to 216 |

**Table 3. Preindustrial ocean carbon inventory, sea surface temperature and sea ice area.** All values are annual averages.

| | Ensemble mean | Ensemble range | Observations |
|---|---|---|---|
| Global ocean carbon inventory (PgC) | 36056.2 ± 252.4 | 35280.5 to 36655.7 | 38000 Houghton et al. (1990) |
| Global sea surface temperature (°C) | 18.9 ± 1.2 | 16.4 to 21.9 | 15.9 NCDC, 2015 |
| Global sea ice area (million $km^2$) | 23 ± 4.2 | 16.3 to 38.6 | 19 to 27 Lemke et al. (2007) |



## 4 LGM ensemble simulations

### 4.1 Atmospheric carbon dioxide

The ensemble $\Delta CO_2$ distribution is centered around -20 ppmv, with a range of -88 to 74 ppmv, although the ensemble member corresponding to the latter $\Delta CO_2$ is an outlier (Fig. 1). A more negative $\Delta CO_2$, between $\sim$ -90 and -30 ppmv is achieved by 104 ensemble members. This range roughly accommodates the impact of error in the model's process representations and the impact of certain potentially important $\Delta CO_2$ mechanisms that are missing from the model, such as changing marine bacterial metabolic rate, wind speed (via its effect on gas transfer) and Si fertilization (Kohfeld and Ridgwell, 2009). This is not a

comprehensive assessment, however, as our model also does not include processes such as the effect of changing winds on ocean circulation (Toggweiler et al., 2006), Si leakage (Matsumoto et al., 2002, 2013, 2014), the effect of decreasing SSTs on $CaCO_3$ production (Iglesias-Rodriguez et al., 2002), or changing oceanic $PO_4$ inventory (Menviel et al., 2012). The fact that it is difficult for our model to achieve a $\Delta CO_2$ of $\sim$-90 ppmv without the missing processes is a significant result because of our extensive exploration of model uncertainty when building the model ensemble. Our assumption moving forward is that the

role of processes which *are* included in the model add linearly to the missing ones and that our $\sim -30$ to $-90$ ppmv $\Delta CO_2$ therefore represents a "plausible" glacial atmospheric $CO_2$ change. Rather than trying to identify the "best" parameter set in this "plausible glacial atmospheric $CO_2$ change-filtered (PGACF) ensemble" (Table 4), we investigate what the emergent model output relationships are and what magnitude and direction responses can be observed. This is to avoid focusing on a candidate parameter with potentially the wrong balance of processes. We also compare the behaviour of the PGACF ensemble

(with a mean $\Delta CO_2$ of -45±13 ppmv) with that of the EFPC2 ensemble to better understand the emergent relationships. Moreoever, we identify which members in the PGACF ensemble predict $\Delta CO_2$ from the highly negative end of the ensemble range to help diagnose any significant differences in behaviour along the plausible $\Delta CO_2$ spectrum. The lower $\Delta CO_2$ limit for this subset of the PGACF ensemble is $\sim$ -60 ppmv, roughly equivalent to allowing for an extra atmospheric $CO_2$ decrease due to changing marine bacterial metabolic rate, wind speed (via its effect on gas transfer) and Si fertilization between the best and

upper estimate of Kohfeld and Ridgwell (2009). The ensemble members in the subset are referred to collectively as the "PGACF-16" ensemble (Table 4), with the numeral denoting the number of ensemble members.

We also note here that we have apparently succeeded in reproducing one of the most remarkable properties of the 800-kyear temporal record of $CO_2$, namely the nearly constant lower bound across different glacial states. There is a significant caveat,

however, which is that 47 of the 471 LGM states subsequently diverge to an unrealistic snowball glaciation response to LGM forcing through feedback mechanisms, probably involving low $CO_2$, that have to be inferred to be unrealistically strong in this context. These simulations have to be rejected, thus it is difficult to draw any firm conclusions regarding the dynamical controls that maintain the constancy of the lower $CO_2$ bound in the real system.





(a)    (b)

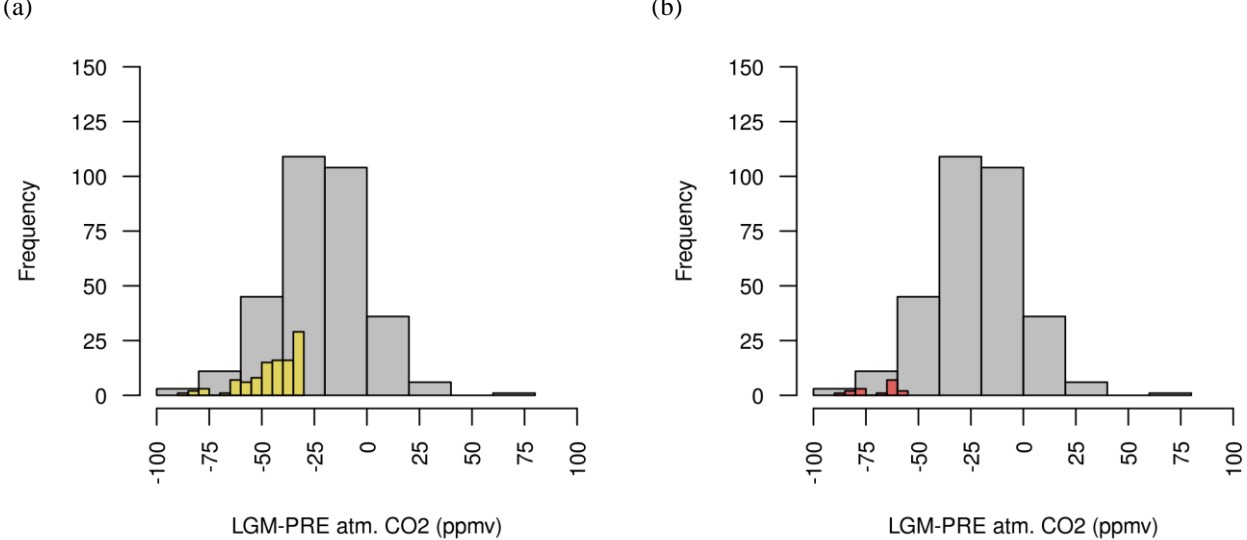

**Fig. 1. LGM change in atmospheric CO$_2$ (a-b) distribution.** The EFPC2 ensemble response is shown in grey, the PGACF ensemble response in yellow and the PGACF-16 ensemble response in purple. Unless otherwise specified, the same colour legend applies to all figures in the manuscript.





**Table 4. The different ensembles and their $\Delta CO_2$.** As discussed in the main text, the EPFC ensemble is the EFPC ensemble of Holden et al. (2013a), which is weakly constrained by modern climate, but includes four additional parameters (that have no effect on modern climate). The EFPC2 ensemble is the same ensemble as EFPC but excludes ensemble members with preindustrial atmospheric $CO_2$ concentration outside of 268 to 288 ppmv, which entered a snowball Earth state in the LGM simulation, or which evidenced numerical instability. The PGACF is the subset of the EFPC2 ensemble with $\Delta CO_2$ between ~-30 and -90 ppmv, while PGCAF-16 represents those PGCAF ensemble members (16 in total) with $\Delta CO_2$ no higher than ~-60 ppmv.

| Ensembles | $\Delta CO_2$ range (ppmv) | Number of members | Details |
| --- | --- | --- | --- |
| EFPC | n/a | 471 | Derived from Holden et al. (2013a) |
| EFPC2 | -88 to 74 | 315 | Subset of the EFPC ensemble |
| PGACF | -88 to -30 | 104 | Subset of the EPFC2 ensemble |
| PGACF-16 | -88 to -59 | 16 | Subset of the PGACF ensemble |

## 4.2 Climate, sea level and ocean circulation

**Temperature**

The LGM SAT anomaly (ΔSAT) in the PGACF ensemble varies between -2.5 and -10.4 °C, with a mean of -4.6 ± 1.7 °C, close to the observed ΔSAT of -4 ± 1 °C (Annan and Hargreaves, 2013). The LGM SST anomaly (ΔSST) varies between -4.5 and -0.7 °C, with a mean of -1.8 ± 0.8 °C, also close to the -1.7 °C observational data-constrained model estimate of Schmittner et al. (2011), and within the -0.7°C to -2.7°C range inferred from proxy data (MARGO Project Members 2009 in Masson-Delmotte et al., 2013). Compared to the EFPC2 ensemble, both the PGACF ensemble ΔSAT and ΔSST tend to be more negative (Fig. 2). There is also a positive correlation between ΔSAT and $\Delta CO_2$ in the PGACF and EFPC2 ensembles (r = 0.75 and 0.74), reflecting the radiative impact of atmospheric $CO_2$ on SAT, and also potentially the effect of changing SAT on $\Delta CO_2$. As suggested above, decreasing SST may contribute to decreasing $CO_2$ via the $CO_2$ solubility temperature dependence. Changing SAT may also affect $\Delta CO_2$ via its effects on sea ice, ocean circulation, terrestrial and marine productivity (see below). In the PGACF-16 ensemble, ΔSAT and ΔSST are from the extreme or at least lower end of the PGACF ensemble range and mostly



well below observations. It is thus possible that the contribution of individual temperature-dependent processes to $\Delta CO_2$ may have been distorted in the PGACF-16 ensemble.

The PGACF ensemble mean ΔSAT spatial distribution is shown in Fig. 3, and in line with observations (Annan and Hargreaves, 2013), the largest LGM decreases (> 10 °C) are found in North America, coinciding with the Laurentide ice sheet, and in the eastern Atlantic/ northwestern Eurasia region, coinciding with the Eurasian ice sheet. The maximum observed SAT decrease, however, is -20 °C, compared to just -16 °C in the ensemble mean. The simulated temperature changes in the North Pacific range between -4 and -7 °C, roughly consistent with the observed -2 to -8 °C changes. At low latitudes conversely, the Pacific, Atlantic and Indian Ocean SAT decreases in the Northern Hemisphere tend to be larger than in the Southern Hemisphere whereas observations show no such divide. The ensemble mean also does not capture the somewhat larger temperature decreases over low latitude land areas (2 to 4 °C) compared to low latitude ocean areas (1 to 2 °C). Another discrepancy is the underestimation of southern high latitudes SAT decreases, with simulated ΔSATs ranging between -4 and -7 °C, compared to between -4 and  -12 °C in observations. At least a fraction of the discrepancy is likely caused by the underestimation of LGM SH sea ice (see below), and which has previously been attributed to excessive atmospheric heat diffusion in GENIE-1 (Lenton et al., 2006; Lunt et al., 2006). This is conversely not an issue in the preindustrial ensemble simulations as winter SH sea ice area was used as a constraint during the modern plausibility filtering process (Table 2).

The PGACF ensemble mean spatial distribution of ΔSST is also shown in Fig. 3, and exhibits the largest decreases (≥ 4 °C) in the North Atlantic and northeast Pacific. The smallest (< 1°C) decreases are conversely found in the polar regions, with ΔSST ranging between -2 and -3 °C in the tropics. These patterns are roughly consistent with the observations (Annan and Hargreaves, 2013) which show ΔSSTs of mostly ≤ -2 °C and ≤ -1 °C in the tropics and polar regions respectively. However, while the ensemble mean polar  ΔSSTs are consistently negative, the observed polar ΔSSTs include SST increases of up to 2 °C. The latter can also be observed in the northernmost North Atlantic.  In the North Pacific, the ensemble mean ΔSSTs are somewhat underestimated, particularly in the west, with the observed ΔSSTs ranging between -2 and -8 °C. The only exception is the Central Pacific, around 60 °N, where SST increases of up to 1 °C can be found in observations. Cooling across the southern mid-latitudes is more severely underestimated, with simulated ΔSSTs ranging between -2 and -3 °C compared to between -2 to -8 °C in observations.



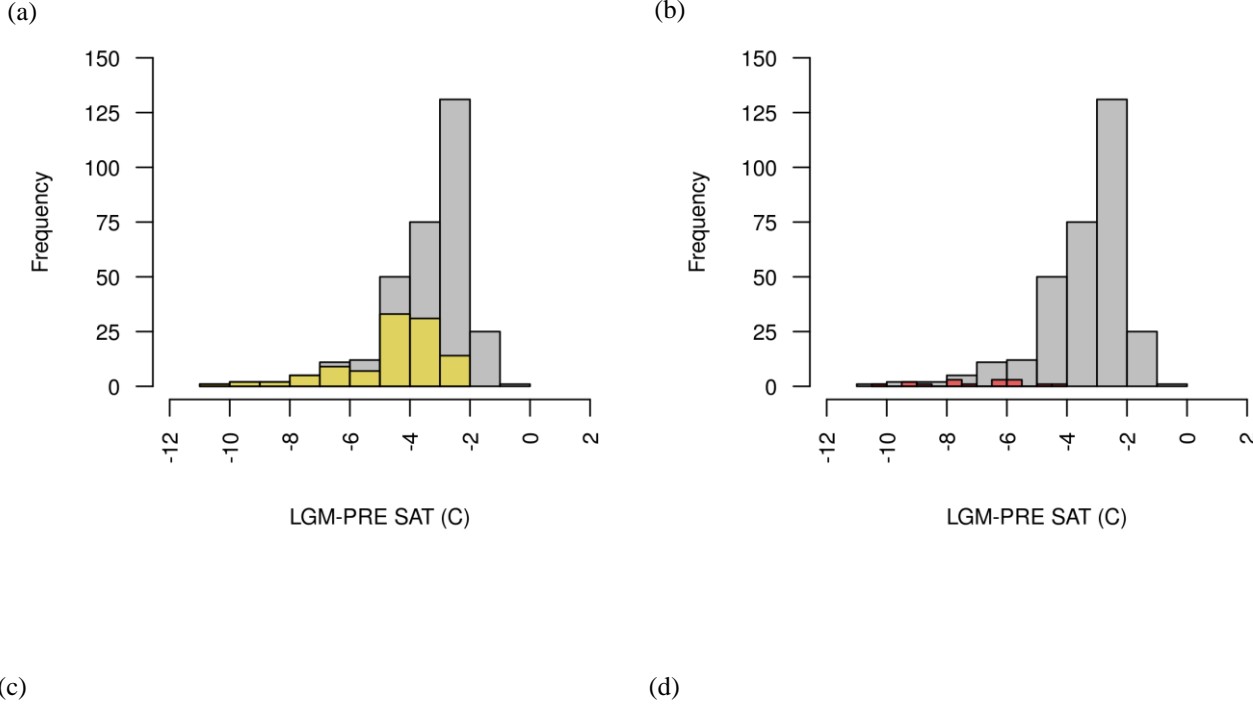

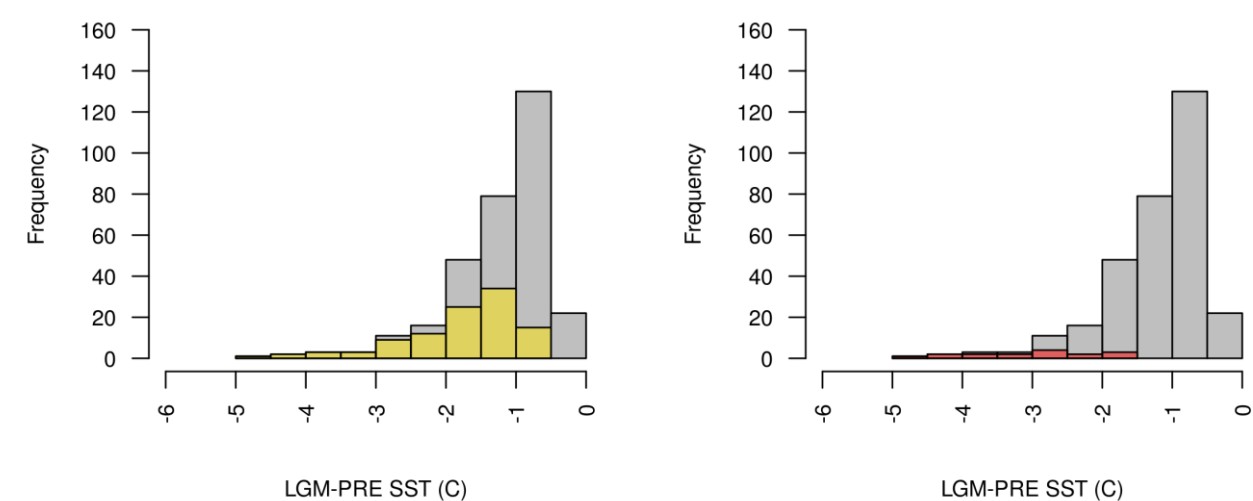

**Fig. 2. LGM change in surface air temperature (a-b) and sea surface temperature (c-d) distributions.**



(a)

(b)

(c)

(d)

**Fig. 3. LGM change in surface air temperature (a-b) and sea surface temperature (c-d) (°C) PGACF ensemble mean (left) and standard deviation (right).**

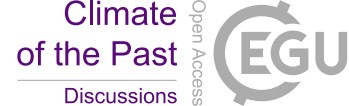



**Sea level**

The percentage increase in LGM salinity (%S) (and DIC, ALK, $PO_4$, etc) due to decreasing sea level in the PGACF
ensemble varies between 2 and 4, with a mean of $2.84 \pm 0.62$. Its distribution is similar to that found in the EFPC2 and the
PGACF-16 ensembles (Fig. 4). Thereis also no significant relationship between %S and $\Delta CO_2$ in the PGACF ensemble,
although there is a weak positive one (r = 0.17) in the EFPC2 ensemble.

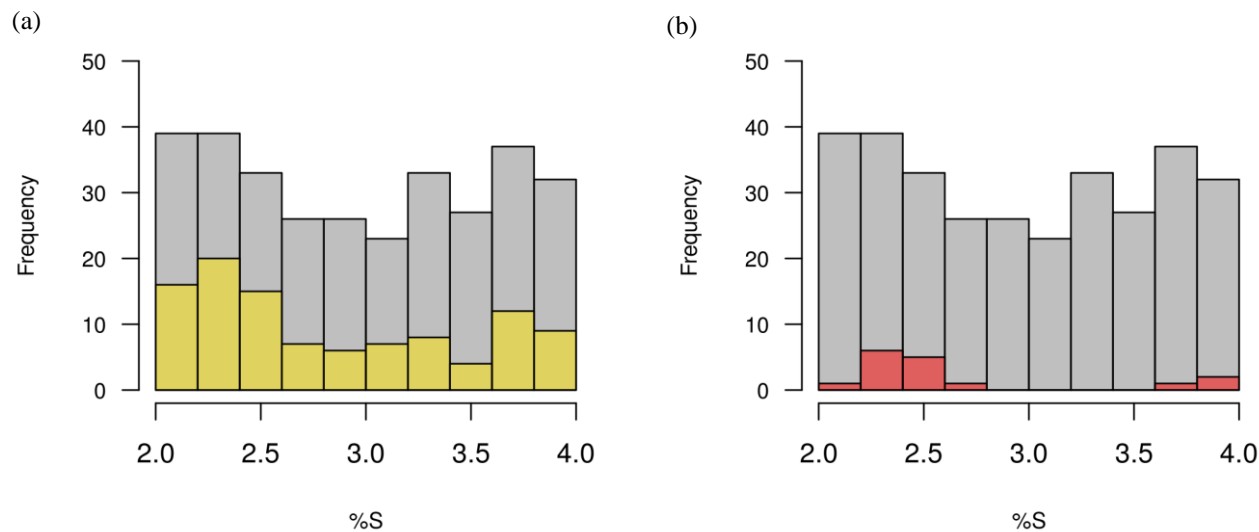

(a)                                                        (b)

**Fig. 4. Percentage increase in LGM salinity due to decreasing seal level (a-b) distribution.**

**Sea ice**

The PGACF ensemble LGM global annual sea ice area anomaly (ΔSIA) has a mean of $18.6 \pm 7.4$ million $km^2$, and a range
of 9.9 to 44 million $km^2$. The ΔSIA in the PGACF ensemble tends to be higher than in the EFPC2 ensemble, and smaller
than in the PGACF-16 ensemble (Fig. 5). There is also a negative correlation between ΔSIA and both ΔSAT (r = -0.97 and -
0.96) and $\Delta CO_2$ (r = -0.74 and -0.74) in both the PGACF and EFPC2 ensembles. The negative correlation between ΔSIA
and $\Delta CO_2$ likely reflects the impact of changing atmospheric $CO_2$ on ΔSIA but may also include a smaller contribution from
changing sea ice area to $\Delta CO_2$. Increasing LGM sea ice area may have for instance reduced the outgassing of $CO_2$ from the
ocean, particularly in the Southern Ocean via sea ice capping, and reduced the net ocean-atmosphere $CO_2$ flux by decreasing
the AMOC strength (see below). As shown in Fig. 6, fractional sea ice cover increases in all regions where sea ice is present
in preindustrial simulations, although the largest increases take place in the North Atlantic.



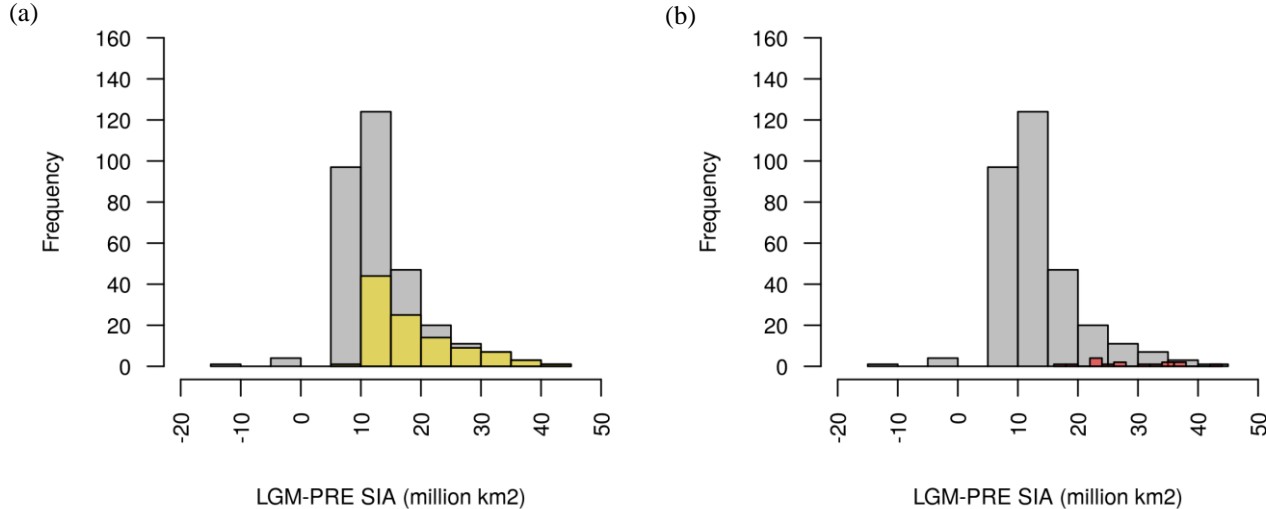

**Fig. 5. LGM change in global sea ice area (a-b) distribution.**



(a)

(b)

(c)

(d)

**Fig. 6. LGM-PRE (a-b) and PRE (c-d) fractional sea ice cover PGACF ensemble means (left) and standard deviations (right).**



### Precipitation

The spatial distribution of the PGACF ensemble mean LGM precipitation rate anomaly ($\Delta$PP) is shown in Fig. 7. The largest LGM precipitation decreases (> 1.5 mm day $^{-1}$, with a maximum of 2.25 mm day $^{-1}$) are found over northern North America, and from around the eastern North Atlantic to northwest Asia, coinciding with the location of the Laurentide and

Eurasian Ice Sheets (and the largest increases in fractional sea ice cover) respectively. The changes are roughly consistent with pollen-based precipitation reconstructions (Bartlein et al., 2011 in Alder and Hostetler, 2015) which suggest that precipitation decreased by $\sim$ 0.27 to 1.1 mm day $^{-1}$ in northern Europe/northwest Asia, and by $\geq \sim$1.1 mm day $^{-1}$ (with a maximum of 3.4 mm day $^{-1}$) in northwest and eastern North America. The observations additionally point toward increased precipitation in the Great Basin region but this is not captured by the ensemble mean potentially because the precipitation

change may be caused by a displacement of the track of the westerlies by the Laurentide Ice Sheet, and the current model does not allow for such dynamical changes (Bartlein et al., 2011). In the observations also, precipitation decreases in the 0.27 to 1.1 mm day $^{-1}$ range extend to southern Europe and to the Middle East, where the ensemble mean conversely predicts precipitation decreases no larger than $\sim$ 0.5 mm day $^{-1}$ and precipitation increases of up to $\sim$ 0.1 mm day $^{-1}$ respectively. Precipitation decreases in the 0.27 to 1.1 mm day $^{-1}$ range are also observed in equatorial Africa where the ensemble mean

predicts somewhat comparable changes (-0.75 mm day $^{-1}$). The main exception is the rainforest region where observations suggest precipitation increased by up to 1.1 mm day $^{-1}$. Relatively large precipitation decreases, of around 1 mm day $^{-1}$, are further simulated for the Tibetan plateau region, for which there is no observational data, and for mid-latitude east Asia, for which there is just one observation, in Japan. The latter suggests a precipitation decrease of 2.4 mm day $^{-1}$ (Bartlein et al., 2011, supplementary information). A potentially more significant discrepancy is the simulated precipitation increases of

up to 0.1 mm day $^{-1}$ in continental Siberia, where observations conversely indicate precipitation *decreases* in the 0.07 to 1.1 mm day $^{-1}$ range. In Beringia also, precipitation decreases by > 0.5 mm day $^{-1}$ in the ensemble mean but exhibits both increases and decreases of up to $\sim$ 1.1 mm day $^{-1}$ in the observations. Although not shown here, comparison of the PGACF ensemble mean against the EFPC2 ensemble mean suggests that the precipitation patterns in the two are very similar, but the decreases generally tend to be higher in the PGACF ensemble mean. The precipitation decreases in the latter conversely tend

to be smaller than in the PGACF-16 ensemble.





(a)                                                          (b)

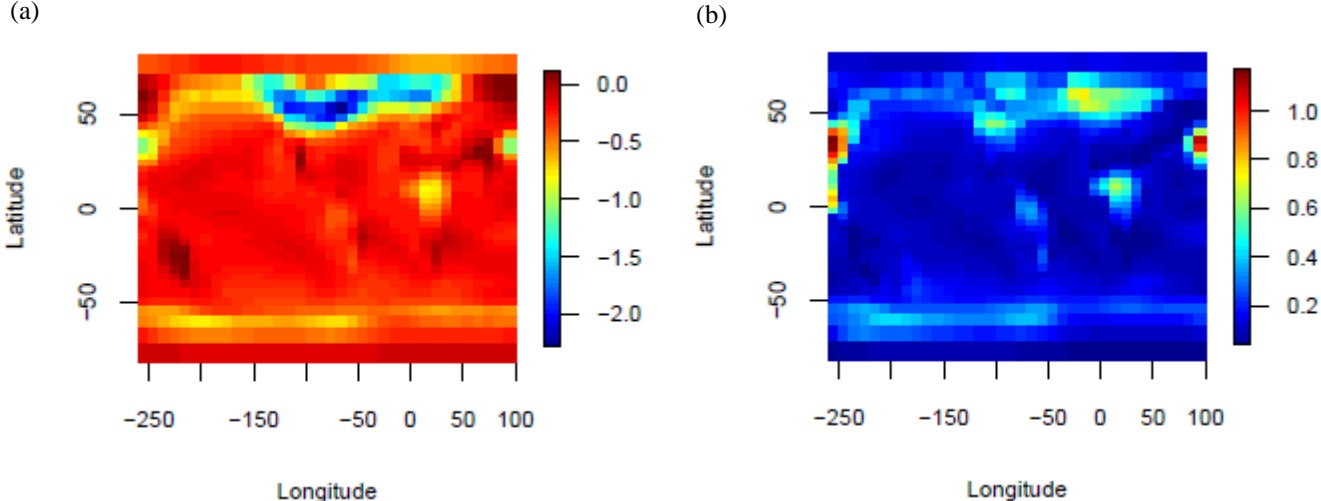

**Fig. 7. LGM change in precipitation rate (mm day$^{-1}$) PGACF mean (a) and standard deviation (b).**

**Ocean circulation**

The PGACF ensemble LGM AMOC strength anomaly ($\Delta\psi_{max}$) has a mean of -2.8 ± 2.8 Sv and a range of -8 to 4.7 Sv (Fig.

8), which is at the low end of the $\Delta\psi_{max}$ predicted by 9 PMIP2 (Weber et al., 2007) and 8 PMIP3 (Muglia and Schmittner,

10   2015) coupled model simulations ([-6.2, 7.3 Sv] and [1.8, 10.4] respectively). It does not, however, include the more

negative $\Delta\psi_{max}$ predicted by Völker and Köhler (2013) for instance ($\Delta\psi_{max}$ = -11.6 Sv). The LGM-PRE AABW cell

strength in the Atlantic Ocean ($\Delta\psi_{min}$) PGACF ensemble mean is 0.1 ± 1.2 Sv, which represents an LGM decrease in cell

strength as the latter is negative to denote the anticlockwise flow of Antarctic water. A negative $\Delta\psi_{min}$ here conversely

always represents an LGM increase in cell strength unless otherwise specified. The total range of $\Delta\psi_{min}$ is from -4.3 to 4.3

15   Sv, roughly comparable to the range of $\Delta\psi_{min}$ (-3.4 to 3.5 Sv) predicted in Weber et al. (2007) (see also Muglia and

Schmittner, 2015), but excluding the much larger $\psi_{min}$ increase (14 Sv) predicted by Kim et al. (2003) for example. Out of

the 104 simulations in the PGACF ensemble, one ensemble member predicts no formation of AABW (LGM $\psi_{min}$ = 0). The

northern limits of the PGCAF ensemble mean LGM AMOC and AABW cells (where $\psi$ again approaches 0 Sv) are roughly

the same latitude as in the preindustrial simulations (Fig. 9). The maximum depth reached by the ensemble mean AMOC

20   base is also similar to preindustrial. Observations (Lynch-Stieglitz et al., 2007; Lippold et al., 2012; Gebbie, 2014; Böhm et

al., 2015), conversely, suggest that the LGM AMOC shoaled to < 2 km, raising its base depth by 2500 and 600 m at the

north and south end of the return flow respectively. The LGM AABW, in turn, is thought to have filled the deep Atlantic

below 2 km, reaching as far north as 65 °N, which is ca. 25 degrees north of its modern northern limit (Oppo et al., 2015).



Although not show here, the PGACF-16 ensemble members tend to exhibit a shoaling of the AMOC and enhanced penetration of AABW. With regard to $\Delta\psi_{max}$ and $\Delta\psi_{min}$, these tend to be more negative (i.e. weaker AMOC and stronger AABW) than in the PGACF-16. The $\Delta\psi_{max}$ and $\Delta\psi_{min}$ in the EFPC2 ensemble tend to conversely be more positive. Analyses of the PGACF and EFPC2 ensembles also reveals a positive relationship between $\Delta\psi_{max}$ and $\Delta CO_2$ (r = 0.57 and

0.59), and a negative relationship between $\Delta\psi_{min}$ and $\Delta CO_2$ (r = -0.42 and -0.36). There is, moreover, a negative correlation between $\Delta\psi_{max}$ and $\Delta\psi_{min}$ (r = -0.62 and -0.63), a positive correlation between $\Delta\psi_{max}$ and $\Delta$SAT (r = 0.68 and 0.66), a negative correlation between $\Delta\psi_{min}$ and $\Delta$SAT (r = -0.4 and -0.4), a negative correlation between $\Delta\psi_{max}$ and $\Delta$SIA (r = -0.62 and -0.66), and a positive correlation between $\Delta\psi_{min}$ and $\Delta$SIA (r = 0.37 and 0.42). Based on these relationships, we hypothesise that increasing LGM AABW (AMOC) strength led to an expansion of the AABW (AMOC) cell, with the latter

restricting the AMOC (AABW) to lower depths (higher latitudes), and reducing its overturning rate (e.g. Shin et al., 2003). Increasing sea ice may have also contributed to increasing $\psi_{min}$ and decreasing $\psi_{max}$ by increasing SH brine rejection and reducing North Atlantic deep convection respectively. The relationships between $\Delta\psi_{max}$, $\Delta\psi_{min}$ and $\Delta CO_2$ may in turn suggest that increasing $\psi_{min}$(decreasing $\psi_{max}$) contributed to decreasing atmospheric $CO_2$. The replacement of NADW by AABW in the North Atlantic would have for instance led to a dissolution of deep sea sediment $CaCO_3$ due to AABW having

a lower bottom water $CO_3^{2-}$concentration than NADW. The increased $CaCO_3$ dissolution flux in turn raises the whole ocean alkalinity, lowering the atmospheric $CO_2$. Enhanced AABW production would have also caused the deep ocean to become more stratified, allowing more DIC to accumulate at depth, and promoting further $CaCO_3$ dissolution. A decrease in NADW formation on its own may have moreover lowered atmospheric $CO_2$ by reducing the outgassing of $CO_2$ at the ocean surface and the burial rate of deep sea $CaCO_3$ through the concomitant increase in deep sea DIC accumulation. Further investigation

is, however, required to verify these causal relationships. An alternative, or more likely complementary, explanation for the correlation between $\Delta\psi_{max}$ and $\Delta CO_2$ for instance is that decreasing $CO_2$ leads to decreasing $\psi_{max}$, and increasing $\psi_{min}$ by increasing the amount of sea ice.



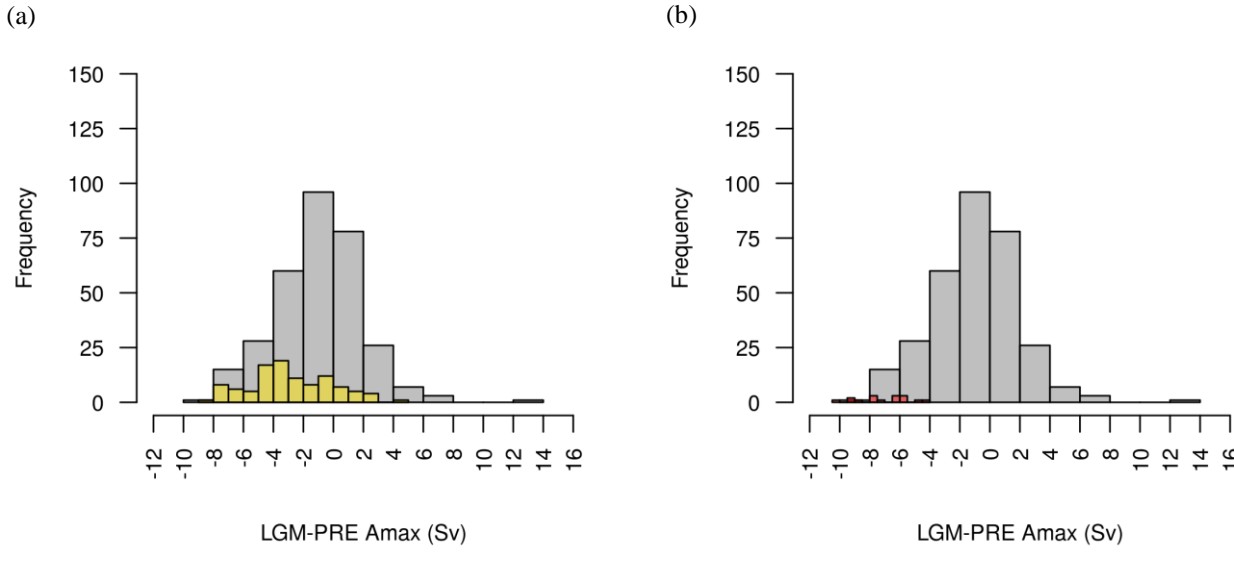

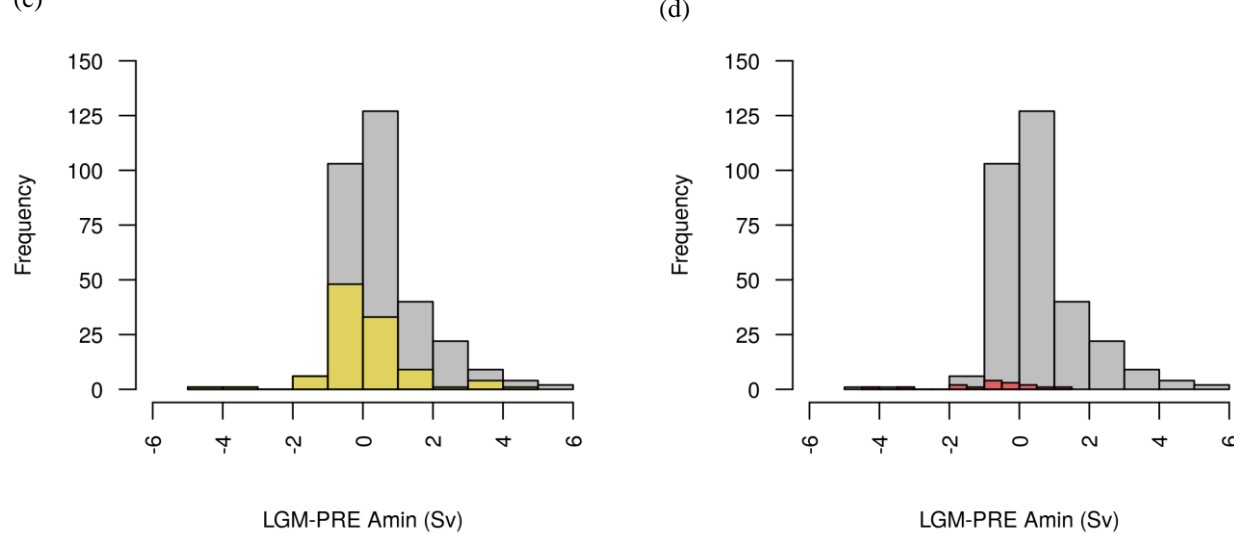

**Fig. 8. LGM change in $\psi_{max}$ (a-b) and $\psi_{min}$ (c-d) distributions.**





**Fig. 9. LGM (a-b) and PRE (c-d) Atlantic overturning stream functions (Sv) PGACF ensemble means (left) and standard deviations (right).**





### 4.3 Terrestrial biosphere, ocean and lithospheric carbon

Most of the ensemble members in the PGACF ensemble predict an LGM increase in terrestrial biosphere (ΔTerrC), and lithospheric[1] (ΔLithC) carbon inventory and a decrease in ocean carbon inventory (ΔOceanC) (Fig. 10). The same trends are

5 also observed in the EFPC2 and PGACF-16 ensembles. More specifically, the EFPC2 predicts there are 7 different scenarios of carbon partitioning between the terrestrial biosphere, ocean and lithospheric carbon reservoirs (Table 5). The dominant scenario, namely an increase in terrestrial biosphere and lithospheric carbon inventory, and a decrease in ocean carbon inventory, includes roughly 89% of ensemble members. In the PGACF and PFACF-16 ensembles, the proportions are 79 and 63% respectively. The second most common scenario is an increase in terrestrial biospere and ocean carbon, and a decrease

10 in lithospheric carbon, comprising 5% of EPFC2 ensemble members, and 11% and 19% of PGACF and PGACF-16 ensemble members respectively. The third most common scenario is an increase in terrestrial biosphere carbon and a decrease in ocean and lithospheric carbon, comprising 3, 8 and 13% of members in the EFPC2, PGACF and PGACF-16 ensembles respectively.

(a)

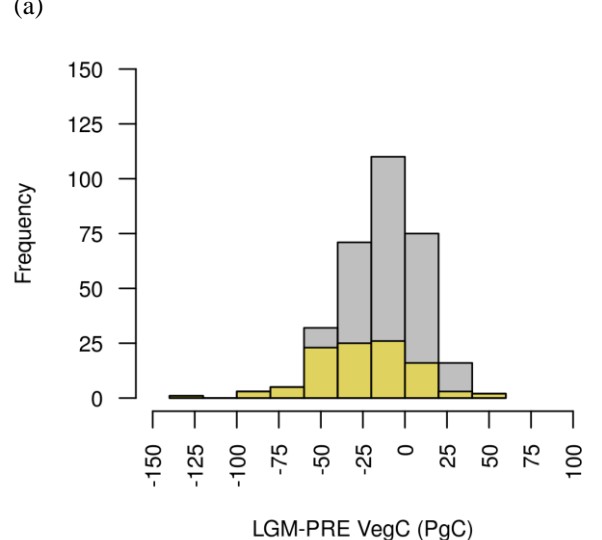

(b)

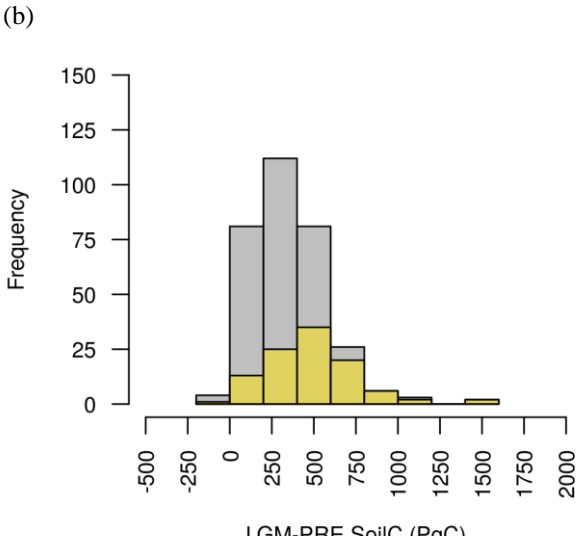

---

[1] The ΔLithC stems from changes in the deep-sea $CaCO_3$ burial flux and/or $CaCO_3$ weathering/shallow water deposition flux and was initially calculated to ensure that carbon was being conserved over the LGM simulation.





(c) 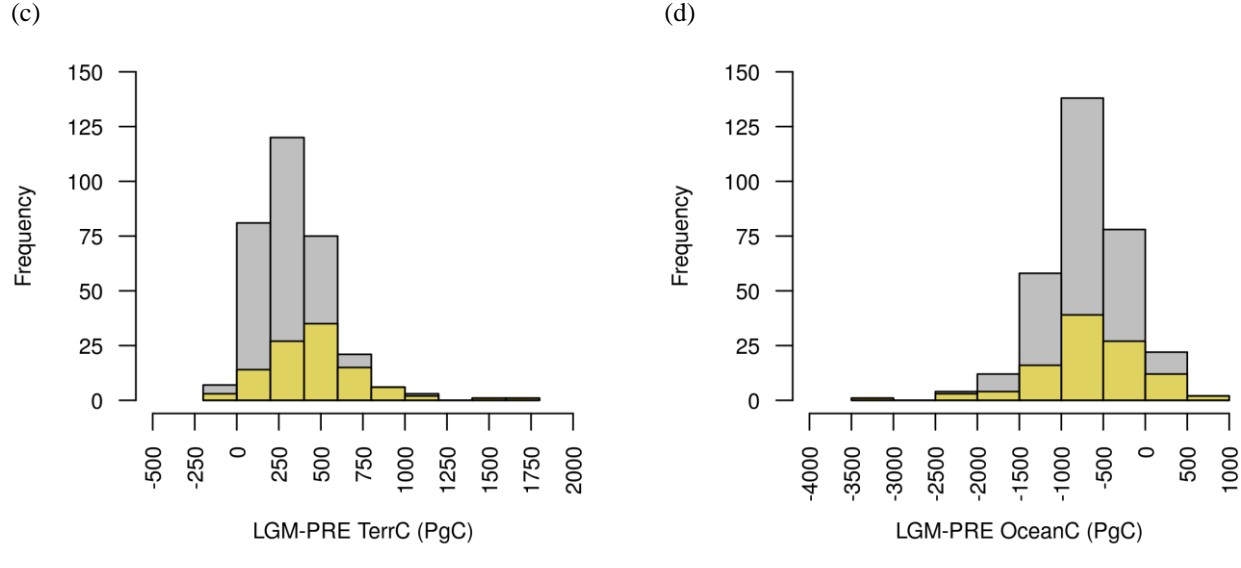

(d)

(e) 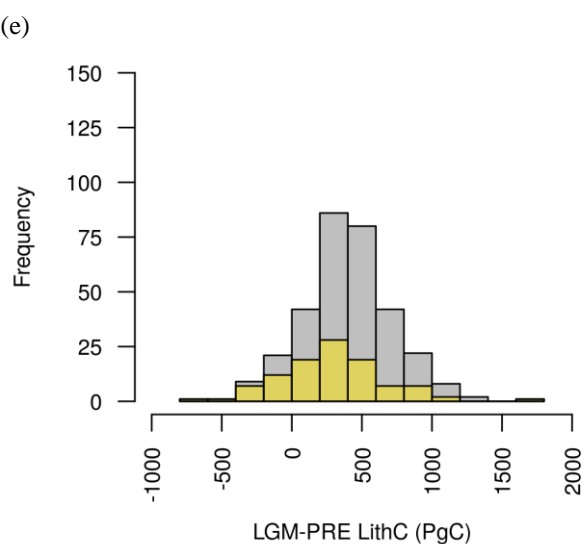

**Fig. 10. LGM change in vegetation (a), soil (b), terrestrial (vegetation + soil) (c), ocean (d) and lithospheric (e) carbon inventory distributions.**



**Table 5. LGM-PRE carbon partitioning scenarios in the EFPC2 (n=315), PGACF (n=104) and PGACF-16 (n=16) ensembles.**

| | Scenarios | | N=315 Total counts | (% of total) | N=104 | | N=16 | |
|---|---|---|---|---|---|---|---|---|
| **1.** $\Delta TerrC(+)$ | $\Delta LithC(+)$ | $\Delta OceanC(-)$ | 279 | (89) | 82 | (79) | 10 | (63) |
| **2.** $\Delta TerrC(+)$ | $\Delta OceanC(+)$ | $\Delta LithC(-)$ | 16 | (5) | 11 | (11) | 3 | (19) |
| **3.** $\Delta TerrC(+)$ | $\Delta LithC(-)$ | $\Delta OceanC(-)$ | 11 | (3) | 8 | (8) | 2 | (13) |
| **4.** $\Delta TerrC(-)$ | $\Delta OceanC(+)$ | $\Delta LithC(+)$ | 1 | (1) | 1 | (1) | 0 | (0) |
| **5.** $\Delta TerrC(-)$ | $\Delta LithC(-)$ | $\Delta OceanC(+)$ | 5 | (2) | 2 | (1) | 1 | (6) |
| **6.** $\Delta TerrC(-)$ | $\Delta OceanC(-)$ | $\Delta LithC(+)$ | 1 | (1) | 0 | (0) | 0 | (0) |
| **7.** $\Delta TerrC(+)$ | $\Delta LithC(+)$ | $\Delta OceanC(+)$ | 2 | (1) | 0 | (0) | 0 | (0) |

As shown in Table 6, a positive $\Delta TerrC$ and a negative $\Delta OceanC$ are only consistent with a handful of observational data- and model- based studies. In the absence of other reservoir changes, a flux of $CO_2$ from the atmosphere to the land (or vice-versa) will lead to immediate outgassing of $CO_2$ from (uptake by) the ocean to remove the atmospheric $pCO_2$ difference. Carbonate compensation of the increased terrestrial carbon storage will also result in an increased deep sea $CaCO_3$ burial

10 flux, and surface ocean $pCO_2$, leading to an increase in lithospheric carbon and atmospheric $CO_2$, and a further decrease in ocean carbon. Here, at least another part of the ensemble changes in lithospheric carbon will have been driven by changes in the land to ocean bicarbonate flux since the latter is prescribed in the simulations. There is also a positive correlation between $\Delta \psi_{max}$ and $\Delta LithC$ in both the PGACF and EFPC2 ensembles (r = 0.25 and r = 0.32), which potentially suggests that reducing the AMOC strength indeed leads to a decrease in deep sea $CaCO_3$ burial rate, as suggested above.

The PGACF ensemble mean $\Delta TerrC$ is of the same sign and order of magnitude as the $\Delta TerrC$ predicted by Zimov et al. ( 2006, 2009), Zech et al. (2011) and Zeng (2003, 2007). $\Delta OceanC$ is not directly calculated in these studies and is instead typically assumed to have decreased by several hundred PgC as a result of the terrestrial biosphere carbon gain. The previous positive $\Delta TerrC$ studies attribute the increase in terrestrial biosphere carbon to different factors: Zimov, N.S. and Zech. R,

20 predict large increases in permafrost carbon while Zeng, N. ignores permafrost and instead attributes most of the glacial terrestrial carbon increase to ice sheet burial carbon. Another contributor is the storage of carbon on exposed continental shelves. Here, neither this carbon accumulation mechanism nor that of peat growth (absent also in Zeng, 2003), are included in our model. As discussed above, permafrost growth is also not represented explicitly but there is an attempt to capture the





very slow rates of soil decomposition characteristic of permafrost (Williamson et al., 2006). Similarly to Zeng (2003), the

model gradually buries carbon in LGM ice sheet areas.

**Table 6. ΔTerrC, ΔOceanC and ΔLithC (PgC) in this study (PGACF ensemble mean, standard deviation and range)
and previous studies.** PE14 = Peterson et al. (2014), OA13 = O'ishi and Abe-Ouchi (2013), C12 = Ciais et al. (2012), Z03
= Zeng (2003), SA13 = Sarnthein et al. (2013), BG15 = Brovkin and Ganopolski (2015), PR11= Prentice et al. (2011),
SK15 = Skinner et al. (2015), SS16 = Schmittner and Somes (2016), GL13 = Goodwin and Lauderdale (2013), A15 =
Allen et al. (2015), CR95 = Crowley et al. (1995), AF98 = Adams and Faure (1998), BR12 = Brovkin et al. (2012), ZI09 =
Zimov et al. (2009), ZE11 = Zech et al. (2011).

|  | This study | Previous studies | Ref. | Details |
|---|---|---|---|---|
| ΔTerrC | 467.5±286.5 [-51.6,1603.8] | [-1160, 530] | CR95 | Pollen database |
|  |  | -1500 | AF98 | Ecological data |
|  |  | [-694, -550] | PR11 | Simulation with LPX |
|  |  | -600 | BR12 | Simulation with CLIMBER-2 |
|  |  | -597 | OA13 | Simulation with MIROC-LPJ |
|  |  | -511 | PE14 | Benthic foraminiferal $\delta^{13}$C records |
|  |  | -330 | C12 | Benthic foraminiferal, ice core and terrestrial $\delta^{13}$C records + simulation with LPJ land ecosystem model |
|  |  | 0 | BG15 | Simulation with CLIMBER-2 (+ permafrost, peat, glacial burial carbon) |
|  |  | 547 | Z03 | Simulation with a coupled atmosphere-land-ocean-carbon model |
|  |  | [200, 400] | ZE11 | Soil carbon measurements |
|  |  | <1000 | ZI09 | Soil carbon measurements |
| ΔOceanC | -664±626.9 [-3187.7, 662.4] | [730, 980] | SA13 | Ocean radiocarbon records |
|  |  | 687 | SK15 | Ocean radiocarbon records |
|  |  | 654 | A15 | Ocean [$CO_3^{2-}$] reconstructions + benthic foraminiferal $\delta^{13}$C records |
|  |  | [570, 970] | GL13 | Ocean [$CO_3^{2-}$] reconstructions |
|  |  | 520 | C12 | Benthic foraminiferal, ice core and terrestrial $\delta^{13}$C records + simulation with LPJ land ecosystem model |
|  |  | [510,670] | SS16 | Simulation with MOBI 1.5 coupled to Uvic |
| ΔLithC | 292.5±373.9 [-654.9,1700.9] | n/a | n/a | n/a |





Analysis of the PGACF-16 ensemble members' terrestrial carbon reservoirs suggests that if the preindustrial ice sheet carbon inventory (the terrestrial biosphere carbon in grid cells to be buried by the LGM ice sheets) were to be destroyed instead of preserved at the LGM, the ΔTerrC would be negative in all but 3 simulations (Tables 7 and 8). During the 1000 years of LGM ice sheet build-up, the ice sheet carbon increases by between 6 and 444 PgC, yielding LGM ice-sheet carbon

inventories (or "burial" carbon inventories) between 318 and 1341 PgC (Table 8), and accounting for less than half of the total LGM change in terrestrial carbon (i.e. ΔTerrC) in all but one simulation (EM 442). Most of the ice sheet carbon increase is due to soil carbon, with vegetation carbon decreasing in all but one simulation (EM 871). The range of carbon inventory changes includes the 116 PgC increase in ice sheet carbon, and consequent total burial carbon inventory (431 PgC), estimated by Zeng (2003). It would also allow for an additional 250-550 PgC (Franzen, 1994) from increased glacial

peat accumulation to be buried under the ice sheets, as suggested by Zeng (2003). No observational data-based estimates of the LGM burial carbon inventory are available since there is so far only limited evidence for organic material being preserved by ice during glaciations (Franzen, 1994 and references in Weitemeyer and Buffett, and Zeng, 2007). Outside of the ice sheets, the terrestrial carbon inventory increases in all but one simulation (EM 219), by between 111 and 1089 PgC. This is mostly due to soil carbon, which increases in all simulations. The vegetation carbon, conversely, decreases in all but

three simulations (EM 837, EM 863 and EM 837). The range of non-ice sheet carbon changes includes the 198 PgC increase in non-burial non-shelf terrestrial carbon predicted by Zeng (2003) as a result of reduced soil respiration.

The ensemble LGM terrestrial biosphere carbon increases both inside and outside of the ice sheet areas are presumed to reflect the decrease in soil respiration rate due to colder SATs exceeding the decrease in net photosynthesis (i.e. total

photosynthesis-respiration) rate due to lower $CO_2$, SAT and precipitation, as they are mainly driven by soil carbon increases. In most previous model- and observational data-based studies it is conversely suggested that the reduced photosynthesis effect due to climate and $CO_2$ changes outcompetes the reduced respiration effect on a global scale. In O'ishi and Abe-Ouchi (2013) for example, the terrestrial carbon that would be gained if ice sheet carbon was preserved rather than destroyed (383 PgC) is smaller than the amount of carbon that is lost in response to the LGM climate and $CO_2$ changes (502 PgC). Although

not evaluated directly, it is likely that similar ice sheet/non-ice sheet terrestrial carbon proportions to those in the PGACF-16 are found in the PGACF and EFPC2 ensembles because of the similar climate change distributions exhibited in all three instances. The spatial distribution of ΔTerrC in the PGACF ensemble (described below) is also characteristic of that found in the PGACF-16 and EFPC2 ensembles.

The largest PGCAF ensemble mean LGM increases (≥ 20 kgC m$^{-2}$) in total terrestrial carbon are found in North America and Europe/western Asia, both within and south of the Laurentide and Eurasian ice sheet margins (Fig. 11). Regions with smaller but still relatively large (≥ 10 kgC m$^{-2}$) increases include the Andes and Patagonia regions, the southern tip of the African continent, eastern north Siberia and the grid cells just south of the Tibetan plateau. The largest LGM decreases in terrestrial carbon (≥ 10 kgC m$^{-2}$) conversely tend to be found in northwest North America, Beringia and the Tibetan plateau



region. Other regions with relatively large ($\geq 5$ kgC m$^{-2}$) decreases include equatorial Africa and the deserts in central Asia. Everywhere else the LGM terrestrial carbon density increases by between 0 and 10 kgC m$^{-2}$. Comparison against paleoecological reconstruction studies (Crowley et al., 1995) suggests that the simulated terrestrial carbon changes within the Laurentide and Eurasian ice sheet areas are of the wrong sign, except in northwest North America, since these studies

assume the complete destruction of vegetation and soils in ice sheet areas. Discrepancies between the PGACF ensemble mean and observations further arise from the rainforest regions, where the ensemble mean predicts terrestrial biosphere carbon density changes between -5 and 10 kgC m$^{-2}$, well above observed changes of $\sim$ -23 kgC m$^{-2}$. It is important to note, however, that as suggested in Zeng (2007), the rate of decomposition of soil carbon at the LGM may have been slower than assumed in pollen data-based studies. The largest increases in terrestrial carbon density ($\sim 40$ kgC m$^{-2}$) produced by

the ensemble mean are comparable to those found in areas with permafrost growth (Zimov et al., 2006). However, the peaks are potentially misplaced, being located within and south of the Laurentide and Eurasian ice sheet covered areas, rather than in eastern Siberia and Alaska. Alternatively, terrestrial carbon increases in eastern Siberia and Alaska are simply underestimated in the ensemble mean and  large increases in terrestrial carbon indeed took place within the ice sheet areas during glacial periods. The highly negative LGM terrestrial carbon changes in northwest North America and adjacent

Beringia are likely caused by precipitation decreasing comparatively more than SAT, and causing the decrease in photosynthesis to exceed the decrease in soil respiration. However, it is also noteworthy that, although not shown here, the regions with the largest decreases in terrestrial carbon density, namely northwest North America, Beringia and the Tibetan plateau area, are also the regions with the largest terrestrial carbon densities in the preindustrial simulations.





**Table 7. Preindustrial ice sheet carbon (PgC) and contribution (%) to preindustrial total land (i.e. vegetation + soil), vegetation and soil carbon (PgC) in each PGACF-16 ensemble member (EM).**

| EM | Ice-sheet | % Total Land | % Vegetation | % Soil |
|---|---|---|---|---|
| 442 | 456 | 24 | 14 | 28 |
| 873 | 896 | 35 | 20 | 40 |
| 511 | 677 | 27 | 14 | 31 |
| 99 | 372 | 22 | 13 | 25 |
| 871 | 404 | 23 | 15 | 28 |
| 786 | 502 | 26 | 15 | 30 |
| 107 | 540 | 33 | 18 | 36 |
| 701 | 549 | 28 | 16 | 33 |
| 801 | 707 | 35 | 19 | 40 |
| 219 | 312 | 21 | 14 | 24 |
| 694 | 389 | 25 | 14 | 29 |
| 623 | 697 | 32 | 16 | 36 |
| 522 | 713 | 37 | 18 | 42 |
| 863 | 408 | 24 | 16 | 28 |
| 478 | 573 | 31 | 16 | 35 |
| 837 | 784 | 47 | 31 | 53 |

**Table 8. LGM ice sheet carbon change and contribution (%) to the total LGM terrestrial carbon change (i.e. ΔTerrC) in each PGACF-16 ensemble member (EM).** Also shown is the amount of carbon (in PgC) buried underneath the ice sheets during the LGM and the LGM non-ice sheet carbon change. All units are PgC.

| EM | LGM-PRE ice sheet | % LGM-PRE Total land | LGM Burial | LGM-PRE non-ice sheet |
|---|---|---|---|---|
| 442 | 117 | 51 | 573 | 111 |
| 873 | 444 | 29 | 1341 | 1089 |
| 511 | 262 | 32 | 939 | 567 |
| 99 | 33 | 20 | 405 | 130 |
| 871 | 149 | 26 | 553 | 425 |
| 786 | 131 | 21 | 633 | 486 |
| 107 | 86 | 22 | 626 | 310 |
| 701 | 161 | 36 | 710 | 283 |
| 801 | 275 | 39 | 982 | 423 |
| 219 | 6 | 16 | 318 | -34 |
| 694 | 95 | 30 | 484 | 227 |
| 623 | 181 | 36 | 879 | 319 |
| 522 | 210 | 28 | 923 | 531 |
| 863 | 73 | 16 | 480 | 380 |
| 478 | 165 | 41 | 739 | 233 |
| 837 | 395 | 33 | 1179 | 796 |





(a)

(b)

(c)

(d)

(e)

(f)

**Fig. 11. LGM vegetation (a-b), soil (c-d) and total terrestrial carbon changes (e-f) PGACF ensemble mean (left) and standard deviation (right).** Units are kgC m$^{-2}$.





## 4.4 Ocean primary productivity

The PGACF ensemble LGM total POC export flux change ($\Delta POC_{exp}$) has a mean of -0.19 $\pm$ 1 PgC yr$^{-1}$ and varies between

5  -2.57 and 2.56 PgC yr$^{-1}$, roughly within the range of previous model-based estimates (e.g. Brovkin et al., 2002; Bopp et al., 2003; Brovkin et al., 2007; Chikamoto et al., 2012; Palastanga et al., 2013; Schmittner and Somes, 2016; Buchanan et al., 2016) (Fig. 12). The POC flux decreases in the EFPC2 and PGACF-16 ensembles tend to be smaller and larger respectively. The $\Delta POC_{exp}$ is also positively correlated with $\Delta\psi_{max}$ (r = 0.72 and 0.79) and negatively correlated with $\Delta\psi_{min}$ (r = -0.62 and -0.58) in both the PGACF and the EFPC2 ensembles. The correlations potentially suggest that decreasing AMOC

10  strength and increasing AABW production lead to decreasing POC export. One possible mechanism is enhanced deep ocean stratification due to increasing AABW formation leading to more efficient trapping of nutrients at depth and hence reduced availability in the euphotic zone. All else held constant, a weaker and shallower AMOC cell would also inhibit the transfer of nutrients from the deep ocean to the surface. A negative correlation can additionally be found between $\Delta POC_{exp}$ and $\Delta SIA$ (r = -0.55 and -0.6), probably because no primary production occurs beneath the sea ice surface and increasing sea ice area at

15  the LGM therefore leads to decreasing POC export flux. This would also explain the largest PGACF ensemble mean decreases in POC export flux coinciding with increases in sea ice fraction (Fig. 13).

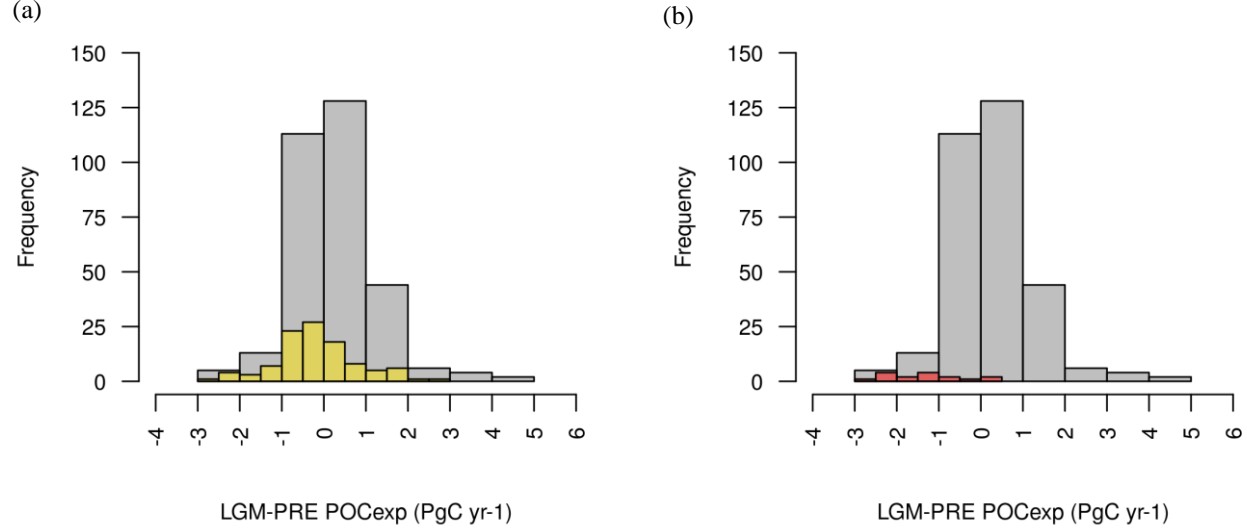

**Fig. 12. LGM change in POC export flux (a-b) distributions.**



(a)                                                    (b)

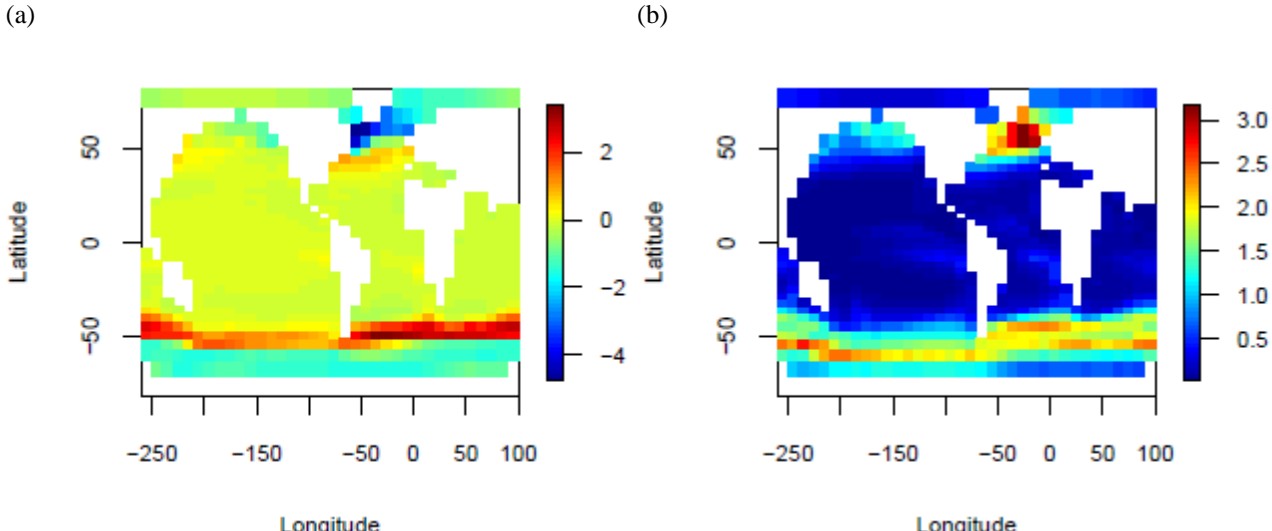

**Fig. 13. LGM surface POC export flux change (molC m$^{-2}$ yr$^{-1}$) PGACF ensemble mean (a) and standard deviation (b).**

The largest LGM PGACF ensemble *increases* in POC export flux conversely occur at around 50 °S, roughly in front of the Antarctic sea ice margins. Increases in POC export are also simulated close to the North Pacific and Atlantic sea ice margins, as well as in the eastern equatorial Pacific and the southwest Atlantic upwelling region. The sea ice margin POC export flux increases are likely caused by the advection of unutilised nutrients from underneath the sea ice. However, they may additionally be caused by the enhanced iron availability from the increased supply of aeolian dust, particularly in the Southern Ocean and North Pacific since these are strongly limited by iron (Ridgwell et al., 2007). Iron fertilization may also explain the increases in POC export flux in the eastern equatorial Pacific and in the southwest Atlantic upwelling region. Comparison against observations suggests that the ensemble mean POC flux changes immediately north, and south of the Antarctic sea ice margins align with observations of increased and reduced marine productivity in the Subantarctic (∼45 to 60 °N) and Southern Ocean respectively (Kohfeld et al., 2005; Kohfeld et al., 2013; Jaccard et al., 2013; Martínez-García et al., 2014). The simulated decreases in export flux in the Arctic and subarctic Atlantic (i.e. above ca. 50 °N), and the increases in export flux immediately south of 50°N are also in agreement with previous reconstructions (Kohfeld et al., 2005; Radi and de Vernal, 2008).

The mostly lower LGM export fluxes at the equator and in the South Atlantic are conversely inconsistent with the observational data of Kohfeld et al., (2005). The decreases may be caused by the increases in productivity in HNLC regions reducing the phosphate (the other limiting nutrient in GENIE-1 besides iron) availability for photosynthesis in other regions. They may additionally be due to the model not simulating enhanced nutrient inventories in response to enhanced weathering



or reduced shallower water deposition of organic matter. The model also does not vary wind speed which may have resulted in stronger tropical upwelling in the Atlantic at the LGM. The evidence for the North Pacific is somewhat ambiguous. Jaccard et al. (2010) suggest there was a decrease in productivity in the subarctic western Pacific (at around 50 °N) while Kohfeld and Chase (2011) suggest that there was a decrease in productivity in the coastal northwest Pacific but also that the

export fluxes in the open ocean may have been higher. The evidence for the equatorial Pacific is also mixed, with Kohfeld et al. (2005) suggesting that the export fluxes were higher during the LGM while Costa et al. (2016) suggest that productivity was reduced around the centre of the region. Ocean productivity data for the South Pacific appears to still be rather limited. In the Indian ocean, Kohfeld et al. (2005) suggest that LGM POC export fluxes were higher rather than lower, as in the PGACF ensemble mean, particularly in the equatorial region. Singh et al. (2011) conversely suggest that the fluxes were

lower.

### 4.5 Carbonate weathering and shallow water deposition

The PGACF ensemble glacial weathering factor (GWS) has a mean of $1.16 \pm 0.24$ (corresponding to a percentage change in the land to ocean bicarbonate flux (%LOC) of 38.67), and a range of 0.52 to 1.5 (corresponding to a %LOC between -49.33 and 50), roughly covering the range of prescribed glacial weathering changes in EFPC2 (Fig. 14). Compared to the EFPC2 ensemble, however, the GWS in the PGACF ensemble tends to be larger, while conversely tending to be smaller than in the PGACF-16 ensemble. There is also a negative correlation between GWS and $\Delta CO_2$ (r = -0.52) in the EFPC2 ensemble,

suggesting that increasing the input of bicarbonate to the ocean leads to a decrease in $CO_2$ by raising the inventories of ALK and DIC in a 2:1 ratio. The total effect of varying GWS over its full range is $\sim 40$ ppmv (Fig. 15). In the PGACF ensemble, however, r is below the 0.05 significance level, suggesting that it is less important.



(a)                                                    (b)

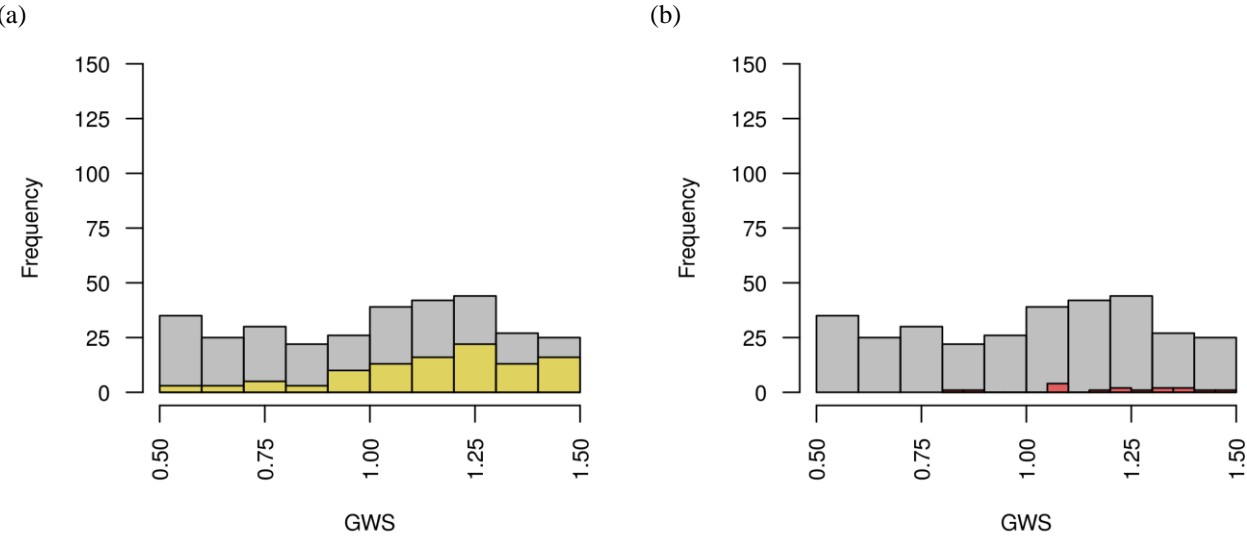

**Fig. 14. GWS (a-b) distributions.**

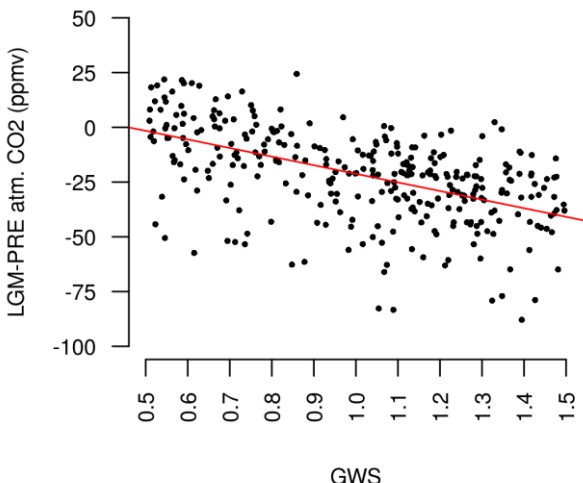

**Fig. 15. Scatterplot of LGM change in atmospheric $CO_2$ versus GWS in the EFPC2 ensemble.**





### 4.6 Deep-sea carbonate burial

The PGACF ensemble global deep sea sediment $CaCO_3$ burial flux anomaly ($\Delta CaCO3_{bur}$) has a mean of $0.036 \pm 0.045$ PgC $yr^{-1}$, and a range of -0.098 to 0.139 PgC $yr^{-1}$ (Fig. 16). The ensemble mean value is ca. 3 times larger than the observed

value (Catubig et al., 1998), although the latter still falls within the range of simulated values. The $\Delta CaCO3_{bur}$ in the PGACF ensemble tend to be higher than in the EFPC2 ensemble, and lower than in PGACF-16 ensemble. It is strongly determined by GWS, as suggested by the positive correlation (r = 0.88 and 0.9) between the two, in both the PGACF and the EFPC2 ensembles. Increasing %LOC should indeed cause the $CaCO_3$ burial flux to increase as increasing ALK means the deep ocean $CO_3^{2-}$ will eventually increase, causing the saturation horizon to fall and allowing $CaCO_3$ to accumulate over greater

areas (which are now exposed to undersaturated waters) (Sigman and Boyle, 2000). The input of ALK to the surface ocean will also increase the rate of $CaCO_3$ export production (which will in turn increase the sediment deposition flux of $CaCO_3$) since as discussed in Chikamoto et al. (2008), the latter is proportional to the production rate of POC (which is equal to the POC export flux), together with the sea surface saturation state with respect to $CaCO_3$, in GENIE-1. There is indeed also a positive correlation between %LOC and the global change in $CaCO_3$ export (r = 0.27 and 0.4), and between the latter and

$\Delta CaCO3_{bur}$ (r = 0.34 and 0.45) in both the PGACF and the EFPC2 ensembles.

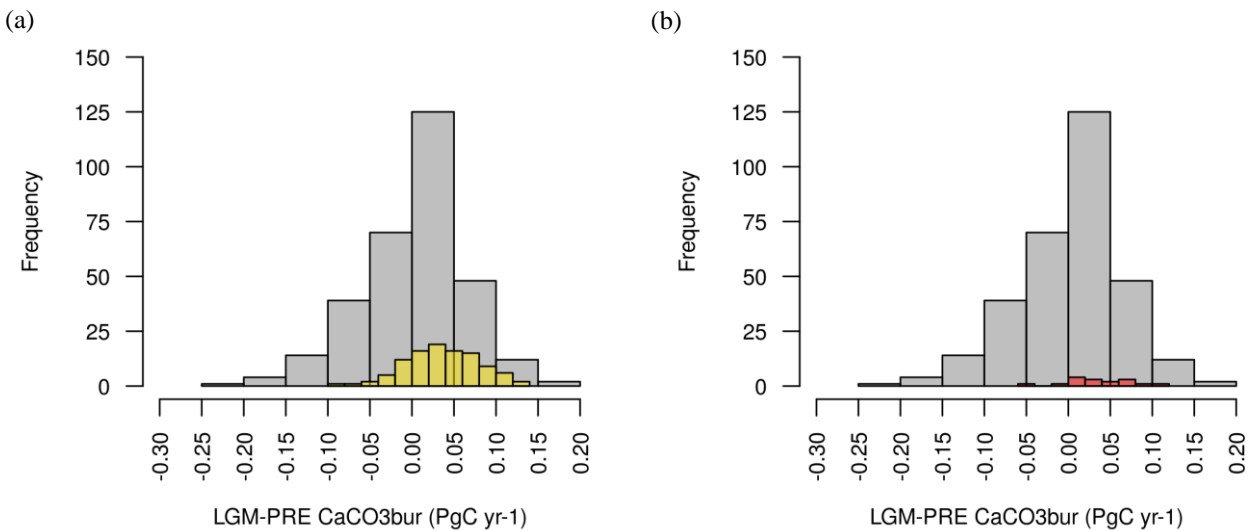

**Fig. 16. LGM change in deep sea sediment $CaCO_3$ burial flux (a-b) distributions.**





The PGACF ensemble mean spatial distribution of $\Delta CaCO3_{bur}$ is shown in Fig. 17, with the largest LGM increases in burial flux ($\geq 0.5 \times 10^{-5}$ mol cm$^{-2}$ yr$^{-1}$) roughly coinciding with the largest increases in POC export flux, at around 50 °S, and in the North Atlantic, particularly around the west coast. Large increases can also be found in the North Pacific, but within rather than immediately south of the sea ice covered area. The largest decreases in burial flux ($\leq$ -0.5 $\times 10^{-5}$ mol cm$^{-2}$

5   yr$^{-1}$) are found in the Arctic and North Atlantic, north of the sea ice limits, as well as in the Southern Hemisphere immediately south of the peak increases in burial flux. Comparison against the reconstructions of Catubig et al. (1998) suggests that the mostly negative burial changes in the Pacific Ocean are at odds with the observations, except in the western tropical Pacific. The largest increases in burial flux are also observed in the eastern equatorial Pacific, rather than in the North Pacific, as in the ensemble mean. A possible reason for the discrepancy is the underestimation of increases in

10   productivity (and therefore increases in $CaCO_3$ export flux) in the eastern equatorial Pacific. Other disagreements between the ensemble mean and the observations include the large  ensemble mean increases in burial flux across the Atlantic, Indian and Pacific basins at around 50 °S, where observations conversely suggest the fluxes were lower, not higher. The simulated increases may be caused by an overestimation of ocean primary productivity (and therefore $CaCO_3$ export), or alternatively, an overestimation of $CaCO_3$ export fluxes only. The latter could be due to the lack of direct mechanism for reducing the

15   surface rain ratio, such as Si fertilization, or the absence of a temperature control on $CaCO_3$ production since the latter may cease with SSTs below 10 °C (Iglesias-Rodriguez et al., 2002 in Brovkin et al., 2007).

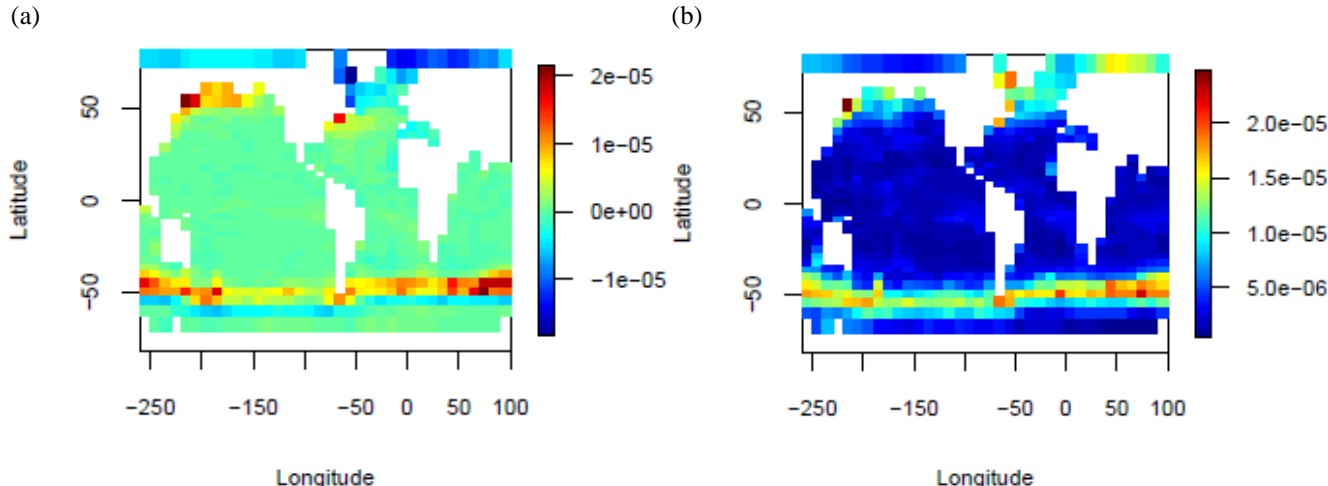

**Fig. 18. LGM deep sea CaCO$_3$ burial rate change (mol cm$^{-2}$ yr$^{-1}$) PGACF ensemble mean (a) and standard deviation (b).**



## 5 Conclusions

We have described the application, to preindustrial and LGM equilibrium simulations, of an ensemble of sets of model parameters which are thought to contribute to variability of atmospheric $CO_2$ on glacial/interglacial timescales. The ensemble (denoted EFPC2) was weakly constrained in Holden et al. (2013a) with eight preindustrial/modern metrics of the atmosphere, ocean, sea ice, terrestrial carbon, ocean biogeochemistry and ocean sediments, and were designed to allow for a wide range of large-scale feedback response strengths. The simulations were forced with orbital parameters, 2-D fields of orography and ice sheet fraction as well as oceanic dust deposition fields. The atmospheric $CO_2$ for the radiative code was generated internally. In the LGM simulations, a range of $CaCO_3$ weathering flux changes was prescribed in an attempt to take into account uncertainty in the observed weathering and shallow water deposition fluxes of $CaCO_3$. Changing sea level was also included as a variable parameter in order to capture the uncertainty in the magnitude of the sea level drop and its effects on the carbon cycle. The terrestrial biosphere carbon in ice sheet areas was, furthermore, configured to be preserved rather than destroyed, and to interact with the atmosphere prior to its burial.

The EFPC2 ensemble preindustrial global SAT, $\psi_{max}$, $\psi_{min}$, $SIA_{SHw}$, VegC, SoilC, wt% $CaCO_3$ and global ocean $O_2$ were compared against the results of Holden et al. (2013a) to verify reproducibility. The evaluation of the ensemble mean response against observations of ocean temperature, salinity, dissolved $PO_4$, dissolved $O_2$, alkalinity and DIC in Holden et al., 2013a was conversely not repeated. The ensemble results were also compared against new metrics, namely OceanC, SST, SIA to verify plausibility. In line with previous GENIE-1 studies, the ensemble mean OceanC was underestimated due to constraints on the model ocean volume. However, for ensemble mean SIA and SST, it was found that the values were in agreement with observations and had an error of comparable magnitude to that found in previous model studies respectively.

Analysis of the LGM-PRE results revealed a centering of the EFPC2 ensemble $\Delta CO_2$ around -20 ppmv and a simulated $\Delta CO_2$ between ~-90 and -30 ppmv in roughly a third of ensemble members. This range was deemed to capture the impact of error in the model's process representations and the impact of a few potentially important missing processes (acknowledging that a larger number of processes have previously been proposed). The fact that our model struggled to achieve a $\Delta CO_2$ of ~ -90 ppmv was argued to be significant because of the extensive exploration of model uncertainty that underpinned the building of our model ensemble. To move forward, we assumed that the impact of processes included in the model added linearly to the missing ones and that the $\sim -30$ to $-90$ ppmv $\Delta CO_2$ therefore represented a plausible glacial atmospheric $CO_2$ change (the PGACF ensemble). Rather than trying to determine the best parameter set within the PGACF ensemble, we explored the magnitude and direction of responses across the ensemble, to identify the emergent model output relationships. We also examined these across larger (EFPC2 ensemble: -88 to 74 ppmv) and smaller (PGACF-16: -88 to -59 ppmv) $\Delta CO_2$ ranges, to check against sample size artefacts and to determine whether processes' behaviour might vary across the $\Delta CO_2$ spectrum. The



focus on a larger number of parameter sets was to prevent us from proposing a $\Delta CO_2$ solution with potentially the wrong balance of processes.

The ensemble members in the PGACF-16 ensemble, with $\Delta CO_2$ closer to observation than in the PGACF ensemble, were
found to be associated, for the most part, with extreme cooling, which may have affected other parts of the model's behaviour. At the same time, however, our analyses revealed that the behaviour of the PGACF-16 ensemble tended to be consistent with that of the PGACF ensemble. The behaviour of the latter also aligned with that of the EFCP2 ensemble. In the PGACF ensemble, decreasing atmospheric $CO_2$ tended to be associated with decreasing SSTs, increasing sea ice area, a weakening of the AMOC, a strengthening of the AABW cell, a decreasing ocean biological productivity, an increasing land to ocean
bicarbonate flux, an increasing terrestrial biosphere carbon inventory and an increasing deep-sea $CaCO_3$ burial flux. The majority of ensemble members were also found to not only predict an increase in terrestrial biosphere carbon but a decrease in ocean carbon and an increase in lithospheric carbon. In total, however, there were five different ways of achieving a plausible $\Delta CO_2$ in terms of the sign of individual carbon reservoir changes. The PGACF ensemble also predicted both positive and negative changes in global POC flux, AMOC and AABW cell strengths, and global $CaCO_3$ burial flux. The bidirectional
change in global $CaCO_3$ burial flux is likely to be a consequence of prescribing both increases and decreases in the land to ocean bicarbonate flux, as our analysis suggests the two are strongly correlated. The bidirectionality in AMOC/AABW strength change in turn suggests a non-linear response to the climate forcings, and may involve atmosphere, ocean and sea ice processes.

The predominantly positive $\Delta$Terr in our ensemble was attributed primarily to the burial of ice sheet carbon. However, it was
also shown that in the ensemble soil respiration tends to decrease significantly more than net photosynthesis, resulting in a relatively large increase in non-burial carbon. Assuming 90% of the terrestrial biosphere carbon induced atmospheric $CO_2$ perturbation gets removed by the oceans and the sediments (e.g. Zeng, 2003; Joos et al., 2004; Kohfeld and Ridgwell, 2009; Zech, 2012), the $\Delta CO_2$ due to the PGACF ensemble mean $\Delta$TerrC is -22 ppmv. Assuming a SST sensitivity of ca. 10 ppmv/°C (e.g. Menviel et al., 2012; Martin et al., 2005) in turn suggests a $\Delta CO_2$ due to $\Delta$SST of -18 ppmv. Their total effect would
account for most of the simulated $\Delta CO_2$. Taking into account the impact of decreasing sea level, however, would raise the atmospheric $CO_2$, as would decreasing POC export as less $CO_2$ is now being stored in ocean biota. A process which conversely will have further decreased atmospheric $CO_2$ is increasing $CaCO_3$ weathering (/decreasing $CaCO_3$ shallow water deposition), particularly since our regression analyses suggest an impact of around 40 ppmv when varying GWS over its full range.

To determine how realistic the aforementioned LGM changes are, the PGACF ensemble mean response was evaluated against observations of physical climate, ocean circulation, terrestrial biosphere carbon, and ocean and sediment biogeochemistry. Naturally some discrepancies were expected due to the differing $\Delta CO_2$ between our ensemble and ice core records. The sign and magnitude of the global temperature responses, however, were aligned with observations. Our evaluation of the PGACF





ensemble mean precipitation changes also suggested that the model reproduces certain key features such as the strong LGM decreases in precipitation around the North Atlantic region but fails to reproduce certain other features that are seemingly related to the model atmosphere's tendency to produce precipitation that is too evenly distributed and limited inter-basin moisture transport. A potential consequence of the latter at least is the overestimation of vegetation carbon density in Siberia,

assuming that moisture is a limiting factor.

Due to the range of observed LGM AMOC and AABW cell strength responses, it was not possible to evaluate the model AMOC and AABW cell strength responses with great accuracy. It was conversely possible to evaluate the model AMOC cell's position as most data-based studies agree that the latter was shallower and that AABW penetrated further north. Despite the

PGACF ensemble mean decrease in $\psi_{max}$, there was no apparent change in the position of the AMOC cell, potentially because of the concomitant decrease in $\psi_{min}$ (stronger AABW cell). In the PGACF-16 ensemble, however the decreases in $\psi_{min}$ tended to be greater than in the PGACF ensemble and associated with a shoaling of the AMOC cell and enhanced penetration of AABW into the Atlantic. The agreement between the PGACF ensemble mean LGM terrestrial biosphere carbon change spatial distribution and observations was found to largely depend on the observations used and the amount of error attributed

to these. As such, we consider our LGM terrestrial biosphere carbon change spatial distribution to not be implausible. With regard to LGM POC export flux changes, observations generally agree on the pattern of change in the Southern Ocean, and this was reproduced by the PGACF ensemble mean. However, there is uncertainty with regard to the sign of the LGM global POC export flux change. The general consensus with regard to the LGM global $CaCO_3$ burial flux change is conversely that it was very small. Deep ocean $[CO_3^{2-}]$ reconstructions also reveal LGM and Late Holocene (0-5 kyr BP) global mean deep ocean

$[CO_3^{2-}]$ that are roughly similar (Yu et al., 2014). In contrast, the PGACF ensemble was found to predict a relatively large mean LGM global $CaCO_3$ burial flux increase. A potential cause of the discrepancy is our range of permissible $CaCO_3$ weathering/shallow water deposition flux changes. Increased terrestrial biosphere carbon inventories may have also contributed since they lead to an increase in $CaCO_3$ burial flux via the process of carbonate compensation. However, we do not yet rule out either of these processes as causes of $\Delta CO_2$ since a high LGM global $CaCO_3$ burial rate bias may have been

introduced by processes missing from the model, such as Si fertilisation, Si leakage and the effect of decreasing SSTs on $CaCO_3$ production. The impact of these processes would be to decrease the rain ratio at the sea bed, leading to a decrease in atmospheric $CO_2$ and increase in ocean carbon inventory. Wallmann (2014) also shows that changes in global $CaCO_3$ burial flux over the last $\sim 100$ kyr BP can be roughly reproduced without invoking a decrease in terrestrial carbon stocks, although their model does simulate an increase in DIC stock and an increase, not a decrease in POC export flux.

To further evaluate the sign and magnitude of our simulated LGM changes, and in particular $\Delta$TerrC, a key future test would be to add carbon isotopes into GENIE-1's terrestrial biosphere carbon module and re-run the PGACF LGM simulations with $\Delta^{14}C_{atm\,(DIC)}$ and $\delta^{13}C_{atm\,(DIC)}$ spun up. As shown in Table 6, a frequent argument for a lower glacial terrestrial carbon





inventory is the reconstructed mean glacial ocean $\delta^{13}$C value of ca. 0.35‰ lower than present due to the fact that plants discriminate against $^{13}$C during photosynthesis. In our simulations conversely, it follows that the increase in glacial terrestrial carbon inventory would have resulted in an *increase* in ocean $\delta^{13}$C. However, other processes such as reduced marine productivity, lower SSTs (Schmitt et al., 2012) and greater sea ice area (Stephen and Keeling, 2000) may have

counteracted at least some of this increase. Increasing LGM $CaCO_3$ weathering flux would have conversely raised ocean $\delta^{13}$C although our model does not account for changes in the organic carbon weathering flux, which if increased would result in a decrease in $\delta^{13}$C. Wallmann (2014) attribute most of the observed LGM ocean $\delta^{13}$C decrease to enhanced weathering and reduced deposition of organic carbon at continental margins due to lower sea levels. Menviel et al. (2015) have also shown that weaker surface winds can contribute to a lower ocean $\delta^{13}$C while these are held fixed in our model. It

has further been argued that enhanced glacial carbonate ion concentrations may have reduced the $\delta^{13}$C in foraminera shells without altering mean ocean $\delta^{13}$C (Lea et al., 1999).

As discussed in Zeng (2003) the glacial increase in terrestrial carbon inventory may also potentially explain the increase in atmospheric $\delta^{13}$C over the glaciation, as well the decrease of about 0.3 ‰ at the beginning of the deglaciation (Smith et al.,

1999). Another feature of the deglacial record, namely the rise in $\delta^{13}C_{atm}$ between ca. 12 and 7 kyr BP, is in turn attributed to increasing SSTs and terrestrial biosphere regrowth on previously ice-covered areas. A more common explanation for the deglacial $\delta^{13}C_{atm}$ variation, which resembles the letter W, is conversely that the beginning of the pattern was caused by the release of old carbon from the deep ocean while the end of the pattern was largely due to terrestrial biosphere regrowth. The middle section of the W, characterised by subdued variation in $\delta^{13}C_{atm}$, is attributed to abrupt climate changes (Schmitt et

al., 2012). More recently, however, it has also been suggested that the deglacial decrease in $\delta^{13}C_{atm}$ at ca. 17.5 kyr BP may be caused at least partly by the demise of iron-stimulated Southern Ocean biological productivity (Fischer et al., 2015) and the release of carbon from thawing permafrost (Crichton et al., 2016). A better test, therefore, would be for our PGACF ensemble members to not only simulate LGM-PRE $\delta^{13}C_{atm}$ but also the transient changes in $\delta^{13}C_{atm}$ over the deglaciation.

Another frequent but more indirect argument for a lower glacial terrestrial carbon inventory is the deglacial drop in atmospheric $\Delta^{14}$C, which is typically attributed to the release of old ocean carbon accumulated over the previous glacial period. Zeng (2007) and others (e.g. Zech, 2012) conversely propose that it is caused by the release of very old, and therefore $^{14}$C-depleted, carbon from the terrestrial biosphere. More recent higher resolution records of deglacial $\Delta^{14}$C and atmospheric $CO_2$ (e.g. Durand et al., 2013 in Köhler et al., 2014; Marcott et al., 2014) have, however, so far not been discussed by the enhanced glacial terrestrial

carbon inventory studies. Another challenge is that the early deglacial decrease in atmospheric $\Delta^{14}$C would have also led to a decrease in ocean $\Delta^{14}$C and this is yet to be reconciled with ocean $\Delta^{14}$C data. Studies so far have suggested there was a decrease in deep ocean $\Delta^{14}$C during the glaciation, corresponding to an increase in ventilation age, and an increase in deep ocean $\Delta^{14}$C at deglaciation, corresponding to a decrease in ventilation age, and also coinciding with the decrease in atmospheric $\Delta^{14}$C



(Hughen et al., 2006; Skinner et al., 2010; Skinner et al., 2015; de la Fuente et al., 2015; Freeman et al., 2015). With a reduced deep ocean ventilation at the LGM, it is assumed that the carbon sequestration capacity of the ocean was enhanced, and that the magnitude and spread of these changes resulted in a global increase rather than decrease in ocean carbon.

5   A potential limitation, however, with the use of ocean $\Delta^{14}C$ to infer larger LGM ocean carbon inventory pools is the presence of $\Delta^{14}C_{DIC}$ which indicate no change in ventilation age, or conversely an increase (Broecker and Barker, 2007; Broecker and Clark, 2010; Cléroux et al., 2010; Lund et al., 2011), and processes besides water mass aging which may have contributed to decreasing glacial $\Delta^{14}C_{DIC}$ such as decreasing atmospheric $^{14}C$ production rate, increased weathering of $^{14}C$-depleted $CaCO_3$, input of $^{14}C$-depleted carbon from the mantle and inaccurate estimation of surface ocean reservoir age (Broecker and Barker, 2007; Lund et al., 2011; Wagner and Hendy, 2015). The feasibility of a large extremely $^{14}C$-depleted deep ocean carbon reservoir has also been contested in terms of atmospheric $CO_2$, deep ocean oxygen (namely the absence of large-scale anoxia) and $CaCO_3$ depth constraints (Hain et al., 2011), and from a dynamical standpoint (Broecker and Clark, 2010). Another strong future test would therefore for our PGACF ensemble to simulate spatially resolved LGM ocean $\Delta^{14}C$, as well as the changes in atmospheric $\Delta^{14}C$ over the deglaciation.

Finally, we have shown that decreasing LGM atmospheric $CO_2$ in our ensemble tends to be accompanied by decreasing POC export production, which all else held constant would result in an increase in deep ocean oxygenation – a feature which we have not assessed. Proxy records to date conversely tend to indicate that there was a decrease, not an increase, in deep ocean oxygen concentration (Jaccard et al., 2016). When observed over sufficiently large areas, the latter supports the presence of an enhanced ocean carbon inventory as deoxygenation can be explained by reduced ocean ventilation (the sole input of oxygen is from the ocean surface) (Wagner and Hendy, 2015). As discussed above, the reduced ventilation is in turn assumed to have led to the accumulation of a significant amount of DIC in the ocean interior. Thus, explaining the lower deep ocean oxygen concentrations without having to reduce ocean ventilation as extensively as suggested by previous studies would as a minimum likely require LGM export production to have increased rather than decreased (assuming the enhanced POC export production does not automatically result in an increase in glacial ocean carbon inventory). Alternatively, it may be possible to increase deep ocean oxygen consumption by increasing organic matter at depth but keeping the surface POC export flux constant. This would require adding processes such as increasing remineralisation depth with decreasing ocean temperature and increasing ballasting into our model (Kohfeld and Ridgwell, 2009; Menviel et al., 2012). We also note that including missing processes affecting the ocean biological pump such as increased oceanic $PO_4$ inventory could potentially result in a net increase in LGM POC export, helping lower oxygen concentrations. However, this would then make reconciliation of our positive $\Delta TerrC$ with the observed negative LGM ocean $\delta^{13}C$ more difficult.

**Acknowledgements**



This work made use of the Darwin Supercomputer of the University of Cambridge High Performance Computing Service (HPCS). K.M.S. Kemppinen thanks staff at HPCS for their technical support, and Antara Banerjee and Alex Archibald for help with R This manuscript was greatly improved by comments from an anonymous reviewer.

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
