# Peer review of "Coupled climate-carbon cycle simulation of the Last Glacial Maximum atmospheric CO2 decrease using a large ensemble of modern plausible parameter sets"

_Climate of the Past, 2017_

## Referee Comment (RC1) · Anonymous Referee #1 · 10 Feb 2018

General comments

I like it very much that the authors provided an comprehensive review of the mechanisms that governing the LGM atm. CO2 drop. And I really appreciate that an extensive body of related work are mentioned during the discussion of model results. However, I do have several major concerns.

1) Research aim:

P1 l15-16 and P3 l19 suggest that the aim of this study is "to investigate the causes

of the LGM atmospheric CO2 drop". To me, this research aim is not appropriate. Previous studies have already proposed some hypotheses regarding the mechanisms that governing the decrease of atmospheric CO2 concentration relative to preindustrial (as have been summarised in the Introduction). What is not yet clear is the interplay of these mechanisms and the relative quantitative contribution of each mechanism to the LGM atmospheric CO2 drop. In this study, the authors did not explicitly propose any new hypothesis. Yet, the quantification of the contribution of different mechanisms using GENIE is not possible due to the simplification of the model and due to that many processes are not accounted for.

I think a more specific research aim/question is needed for this study. When setting up the aim/question, the authors might consider: What are the novel aspects of the model or the EFPC2 ensemble? In this manuscript, is it the first time that an interactive carbon cycle model is applied to LGM? Is it for the first time the sensitivity to process parameters is investigated for LGM climate (Holden et al 2013 pointed out that EMICs are important tools for exploring sensitivities and quantifying uncertainty)? Which mechanism(s) the authors would like to focus on? terrestrial carbon preservation? carbonate weathering?

2) Key findings/conclusions:

The current version of the Conclusion reads more like a summary of the model results. It's not clear what are the key findings of this study. I think the key message would become clear once the research aim/question is given.

3) Comparison of model results to field/proxy data or to results of other models:

A large body of text is devoted to the comparison between model results and data and to the explanation of differences between the two. To me this is a bit overdone. Holden et al. (2010), who used the same model and had the same principle for the design of the ensemble, clearly stated that the ensemble is developed to reproduce the main features, but not the precise observation.

Specific comments (the following comments mainly concern the suggestions for improving the structure and presentation of model results.)

Abstract

P1 l15-16: It would be helpful to specifically state the research aim here.

Introduction

P3 l27: "unknown error" – does it mean the discrepancy between e.g. the modelled circulation and that obtained from proxy data? To me it sounds like a mistake one accidentally made in the model.

P3 l28: "unknown error" – Maybe "missing processes" is more suitable here?

Please add at the end of the Introduction an overview of the upcoming sections, viz., what will be presented in each section.

Method

P5, Table 1: what is OLR?

P6: I think a table or a flow chart summarising the conditions for the four stages would be helpful for readers to understand the set-up of experiments.

How long is the stage 2?

Is the total carbon inventory (that is, sum of atmospheric, terrestrial, ocean and lithospheric carbon inventory) unchanged over the four simulation stage?

LGM ensemble simulations

P9: Please explain here why the subset PGACF-16 is needed. This can be done by moving p37 l30-32 to section 4.1.

It would be very helpful if a brief overview of the following text of this section 4 is presented: which variables of which set will be presented.

P25 l6-7: this is not true because in Table 6 none previous observation/model data study shows negative delta_OceanC.

P36 bottom: Fig. 18 should be Fig. 17.

P36 l3: "North Atlantic" and "North Pacific" should be switched.

Comparison of global-integrated numbers and spatial distribution between model results and data: I suggest to first compare the spatial distribution and then the global-integrated numbers because the latter is just the sum of the former.

Colour slots showing spatial distributions of variables, e.g. Fig 3, 6, 7, ...: please add contour line for land-ocean border.

Plots showing standard deviation in e.g. Fig 3, 6, ...: These plots are shown but never mentioned/used/discussed. So please consider move them to a supplementary information file – there are already many figures in the manuscript.

Conclusions

P39 l15: The positive delta_TerrC has been discussed and justified many times through out the manuscript. Thus, I have the impression that this is one key finding the authors would like to stress. I think this is a bit dangerous because this point is not well supported by data. The authors also seem de-stressing this point several times by stating there are other 4 ways of "achieving a plausible delta_CO2 interns of the sign of individual carbon reservoir changes (although Table 5 suggests those 4 ways are much likely to occur). I have to say I am confused by the above statements.

P39 l39 - P41 l15: I understand that it is a pity that carbon isotopes were not simulated. However, I don't think it is appropriate to extensively present inferred results in the Conclusion section.

---

## Referee Comment (RC2) · P.J. Buchanan (Referee) · 11 Feb 2018

article

**1  General comments**

This study provides some unique perspectives on the glacial drawdown of atmospheric $p$CO$_2$. This problem has been at the forefront of climate research for many decades. So far, many mechanisms have been proposed, but a recipe of changes that is physical and biogeochemically consistent with proxy records and known mechanisms is still elusive. The authors of this study set out to try and achieve this difficult task.

The authors employ a unique set of methods to attack their exploration of what might *plausibly* reduce atmospheric $p$CO$_2$ under glacial conditions. The method involves simulating 315 individual parameter sets using the same Earth System Model (ESM) of intermediate complexity in four stages. From what I can tell, each stage involves the full 315-member ensemble, unless numerical instability problems were encountered. Each ensemble member was therefore independent from another at all stages through the study. Each stage was initialised from the final year solution of its previous stage, except stage 1 which I am unsure about what fields have been used for intialisation. The only boundary conditions that were prescribed to the Earth System Model that I can tell were orbital parameters, aeolian iron deposition rates, detrital flux rate to ocean sediments, and ice-sheet fraction and their orography. The 4 stages are as follows:

1. Stage 1 (PI 10,000 years): relaxed $p$CO$_2$ to 278 ppmv; no interaction between carbon reservoirs; conserved alkalinity in the ocean.

2. Stage 2 (PI 10,000 years): freely evolving $p$CO$_2$; interacting reservoirs; freely evolving ocean alkalinity.

3. Stage 3 (LGM 10,000 years) freely evolving $p$CO$_2$; interacting reservoirs; freely evolving ocean alkalinity; ice-sheet growth and corresponding sea level loss

years 0-1000.

4. Stage 4 (LGM 10,000 years) freely evolving $pCO_2$; interacting reservoirs; freely evolving ocean alkalinity.

Following on from this methodology, the authors lead the reader through the results in a methodical manner, taking on changes in some key environmental variables. In each address of a variable, the authors undergo a detailed assessment of PI versus LGM changes in their model and in proxy records. This is done both in a global sense and a regional sense, sometimes being highly specific. The ground-truthing and comparison to observations and other studies is commendable. It one of the largest comparisons that I have come across in a modelling paper. However, the sheer size of comparison makes the paper unwieldy, and loses the major findings amongst the details. It is also unexpected that the authors concentrate on highly specific regional comparisons because they use an ESM of intermediate complexity that will simply not perform well in many ways. Rather, the authors should focus on the sheer size of their parameter spread and in diagnosing the effect that certain changes have on carbon and climate. I therefore suggest that the authors make an effort to reduce the emphasis on comparison, and focus on how their interesting results might explain the LGM carbon cycle in only a broad sense.

It occurs to me that the authors seem to forget some obvious strengths of their study. Their parameter space is enormous, includes an interactive carbon cycle with carbon-climate feedbacks, as well as many other processes (sea level) that have not been included in other model studies of the LGM. This is of interest to the LGM $pCO_2$ drop problem. I suggest that the authors try and convey these strengths more clearly in their abstract and the final paragraph of the introduction.

A particularly interesting result of this study is the increase in terrestrial carbon under LGM conditions. The authors more often than not find that LGM conditions

are associated with greater carbon held in soils because (1) ice-sheet growth covers previously fertile regions and traps the carbon in the soil, (2) soil respiration rates decrease more than primary production, and (3) terrestrial carbon in exposed continental shelves due to sea level drop increases. The authors do not simulate point 3, and there is a possible methodological problem with point 1, in that the authors grow the LGM ice sheets from pre-industrial to LGM state in only 1,000 years. It is unclear in the manuscript presented here whether this rapid growth of ice sheets that is *prescribed* overtakes highly productive regions that have not yet evolved to become arid tundra land, which have lower carbon content in the soils. A rapid trapping of soil carbon by the unrealistic growth of the ice sheets (almost 100x faster than in reality) would likely overestimate the terrestrial carbon reservoir at the LGM. The authors state in the methods that "Sensitivity simulations were performed to verify the simulated equilibrium state is insensitive to the choice of timescale of ice-sheet growth", and yet the authors do not present any evidence of this sensitivity. I am therefore sceptical of the validity of this result, which I would say is their most important and interesting contribution to the field.

Finally, the conclusion needs complete re-writing. It reads as a re-stating of the methods and then the results, which is simply not useful for the reader. The authors do discuss carbon isotope changes, which are appropriate records to discuss given the main result of an increase terrestrial carbon reservoir. I strongly suggest that the authors re-assess their strengths and major findings, present them concisely, and provide some comments regarding how a higher terrestrial carbon reservoir and low ocean reservoir could have occurred despite most studies indicating the opposite. If this is possible, then this would be a useful contribution to the field.

*Major revisions are needed. If the authors can provide some evidence that changes in the timescale over which the ice-sheets are grown do not play a large role in determining how much carbon is present in the terrestrial reservoir, and the manuscript*

*is rewritten addressing the points above and below, then I advocate publication.*

Some other general points:

- Regarding the figures, I strongly suggest that the authors overlay the red (PGACF-16) over the yellow (PGACF) bars in the histograms. This would reduce a lot of unnecessary replication in the figures. The authors could use a transparency setting to ensure that the red and yellow are easily seen if they have the same number of experiments (frequency).

- The writing is generally okay, but this manuscript definitely requires re-writing in some places. There are too many adjectives and unnecessarily difficult acronyms.

- The presentation of results suffers from the use of opaque acronyms that are easily forgotten after reading the methods section. I strongly suggest that the references to EFPC2, PGACF and PGACF-16 are changed to be more interpretable. $PI_{315}$, $LGM_{104}$, $LGM_{16}$... or equivalent, would be much more helpful.

**2 Specific comments**

**Abstract**

The abstract is quite long. The message that this study delivers could be made considerably more poignant if it were condensed for the reader. The authors make some very interesting findings, such as the increase in non-burial carbon in the terrestrail reservoir due to the slow-down in respiration" and "there are 5 different ways to achieve an atmospheric $p\mathrm{CO}_2$ drop". These findings (mostly in para 2) should be

the focus of the abstract, and I advocate for the more technical aspects (para 1) and comparisons with observations (para 3) be removed. In fact, the entire paragraph 3 of the abstract could be reduced to one important sentence without affecting the findings of this study.

**Introduction**

- Page 3, Line 13 - "dissolve organic carbon inventory" of what? the ocean? soils?

- Page 3, Paragraph 2 - This entire paragraph simply lists the changes that may be assocaited with a glacial ocean. If these mechanisms are to be called upon by the authors, they should be accompanied by at least a brief discussion of why they influence atmospheric $p$CO$_2$. For any non-specialist of palaeoceanographic literature specifically relating to the LGM, this paragraph is totally opaque. I would either expand on these mechanisms or remove entirely when accounting for them in the discussion of the results.

- Page 3, Line 19 - "utilises an ensemble of sets of parameters" a bit clunky. What about "uses a large ensemble (471 parameter sets)"?

- Page 3, Line 23 - probably no need to mention Holden (2010a) or cite other literature. Just state philosophy.

- Page 3, Line 26 - move this to methodology. Not necessary here.

**Methods**

- Table 2 - define acronym SAT in caption. What is EFPC2 and EFPC? Must define these here or introduce the Table later on.

- Page 5, Line 5 - please clarify what a closed biogeochemistry system is. Does it mean no interaction between land-ocean-atmosphere-lithorsphere reservoirs? how are these reservoirs initiated in Stage 1? Using the fields from Holden 2013a?

- Page 7, Lines 1-3 - I don't think it's useful to mention this.

**Preindustrial simulations**

- Page 7, Line 5 - First of all, it would be helpful to change the title of section 3 to "Results: Preindustrial simulations".

- Page 7, Lines 7-12 - Why present results and talk about EFPC ensemble when the authors only use the EFPC2 315-member ensemble? It seems to be an unnecessary inclusion that confuses the reader, rather than helps understanding. I think given the length of this study that it would be helpful to simply cut any inclusion of EFPC 471-member ensemble and simply present the results of the 315-member ensemble.

- Table 2 - change "31/12" to $31^{st}$ Dec.

- Table 2 - change "wt% $CaCO_3$" to "wt% ocean $CaCO_3$"

- page 7, Lines 15-23 - Tables 2 and 3 could be moved to supplementary material. Table 2 discussion could be reduced to a single sentence saying that the preindustrial simulations of the 315-member ensemble reproduced all aspects of the Holden 2013a simulations. Table 3 discussion could be reduced to note that there was good agreement with observations of ocean carbon inventory, SSTs and sea ice extent relative to known values.

- Based on what I've said above, I suppose that this section could actually be reduced to one paragraph, or completely removed if the authors wished to use address PI conditions via comparisons with LGM conditions in the next section.

**LGM ensemble simulations**

- Page 9, Line 7 - "104 ensemble members". Are these presented in yellow in Figure 1? If so, mention it here in the text.

- Page 9, Lines 7-16 - These sentences are confusing for the reader. I understood once reading further on in the paragraph that you do not include these processes in the model, and you are saying that their inclusion would push LGM $pCO_2$ decrease even further, which justifies your choice of a -30 ppmv threshold to define a successful solution of LGM conditions. However, this is not clear. Please re-write.

- Page 9, Lines 16-26 - This needs to be moved to the methods section. From how I understand your thinking: lines 5-16 justify your choice of -30 ppmv threshold; lines 16-26 justify your methodology in treating the LGM simulations.

- Page 9, Lines 22-25 - The lower threshold of -60 ppmv should belong with your choice of the -30 ppmv upper limit. discuss these together, not separated by other sentences and concepts.

- Page 9, Lines 28-33 - This is a very interesting results. Why can't you define the mechanisms that lead or do not lead to the snowball Earth scenario? Surely if you can define a plausible set of mechanisms need to achieve glacial conditions, you can do the same by comparing the 471 PIs, 16 LGMs, and 47 snowballs??? This would mark a significant contribution to the field.

- Figure 1 - Overlay red bars on the yellow bars to make 1 panel.

- Page 11, Line 11 - define SAT

- Figure 2 - Again, overlay red on yellow to reduce unnecessary replication. Use transparency perhaps to show where yellow and red are both present at the same frequency.

- Page 12, Lines 4-16 - Comparisons with obs not necessary at this detail given the focus of the work and the fact that you use an ESM of intermediate complexity. I would expect a discussion of global temperature changes, with perhaps a little bit of basin-wide, regional discussion **if** those are important points for later on. Please reduce this paragraph and combine with the next.

- Page 15, Line 5 - I'd say there is no relationship.

- Page 15, Line 10 - too many adjectives.

- Figure 4 - overlay red on yellow.

- Page 18, Lines 11-13 - confusing sentence please re-write.

- Page 19, Lines 11-15 - when talking about AABW formation rates, it is better to present this in positive units. Oceanographers are familiar that the units are negative in the calculation of the streamfunction. It is less confusing for the reader to present your changes as negatives if the AABW formation rate declines. THis also removes the need to explain that a positive anomaly is actually a decrease.

- Page 20, Line 2-3 - Don't understand. I thought you said that weaker AMOC and stronger AABW was coincident with the glacial runs, these being PGACF-16? Please make this clearer. WHat are you comparing?

- Page 20, Lines 3-8 - These relationships are made more confusing for the reader because you define AABW formation rates as being stronger when they are negative. It owuld be more helpful if strong = more positive.

- Page 20, Line 15 - Please see study of Yu et al (2014) Deep ocean carbonate chemistry and glacial-interglacial atmospheric $CO_2$ changes.

- Page 20, final sentence - This has already been covered above? Also, see Ferrari et al (2014) Antarctic sea ice control on ocean circulation in present and glacial climates. PNAS.

- Figure 9 - please ensure that your colour bars are the same scale! I initially thought that your LGM simulation had strong AMOCs, despite the discussion of weaker AMOCs in the text.

- Page 23 - Very interesting result. I think that this is a unique and interesting contribution to the field and should be a focus of this study.

- Table 5 - why does the order change? Please clarify in caption of correct. Also, add the subheadings "Total counts" and "% of total" or their equivalent to the other columns.

- Table 5 - Please quantify the increases and decreases in this table that accompany the scenarios (i.e. mean).

- Page 25, Line 6 - Is table 6 mentioned previously? This introduction of table 6 is jarring.

- Table 6 - Great Table. Could the acronyms for studies you cite be organised into alphabetical order?

- Page 25, Lines 9-10 - I'm (and probably many readers) not sure what "Carbonate compensation of the increased terrestrial carbon storage" means. It's not clear whether you refer to carbon compensation mechanism in the land or the ocean. Do you mean a loss in oceanic DIC due to terrestrial carbon storage, causing an increase in alkalinity that increases $CaCO_3$ burial? If changing terrestrial carbon

reservoir does have a direct effect on ocean alkalinity and CaCO$_3$ burial, maybe by weathering changes you account for, then please explain more fully.

- Page 25, Line 17 - I find this hard to believe.

- Page 25, Line 22 - and yet present day continental shelves that are inundated are also regions of effective carbon burial through marine export production.

- Tables 7 and 8 - Please make "% Total Land" and its relation to Ice-Sheet carbon more clear in the caption. THis takes time to figure out from the reader.

- Tables 7 and 8 - I think that these tables could be combined to solely show the LGM changes.

- Page 27, Lines 1-16 - The relationship between ice-sheet carbon, non-ice sheet carbon and soil burial carbon needs to be made clearer.

- Page 31, Line 7 - re-write sentence please.

- Page 31, Lines 14-15 - Also the relationship with AABW production and decreased AMOC, which you have just discussed.

- Page 33, Section 4.5 - This tight description of the main effects is what the other sections should emulate, and would tighten up the manuscript considerably.

- Page 35, Line 3 - Again, this sentence is awful to read. Too many adjectives.

- Page 36, Line 7 - I don't see a decline in the figure, just no change.

**Conclusion**

- Page 38, Line 26 - A decrease in ocean POC export is not necessarily associated with an increase in atmospheric $p$CO$_2$. Please see Sigman et al (2010) The polar ocean and glacial cycles in atmospheric CO$_2$ concentration. Nature. for an explanation. Briefly: it is not total POC export, but the efficiency of carbon fixation relative to outgassing that matters.

- Page 39, Line 13 - Why are you now talking about terrestrial carbon here?

- Page 40, Lines 1-11 - Please see Menviel et al (2017) Poorly ventilated deep ocean at the Last Glacial Maximum inferred from carbon isotopes: A data-model comparison study. Paleoceanography.

- Page 40, Lines 13-14 - But atmospheric $\delta^{13}$C at the LGM and preindustrial climates were similar at -6.46 ‰ and -6.36 ‰, respectively.

- Page 40, Line 28 - And what do these new records show?

- Page 40, Line 30 - The deglacial decrease in atmospheric $\delta^{14}$C could also be caused by the exchange of highly negative ocean $\delta^{14}$C with the atmosphere. The argument here is not clear.

**3  Technical corrections**

1. Page 7, Line 9 - "did not evidence numerical instability" –> "did not show evidence of numerical instability"

2. Page 7, Line 15 - "has" –> "had". Please use past tense in results sections.

3. Page 9, Line 7 - "impact of error" –> "error"

4. Page 9, Line 7 - "impact of certain" –> "certain"

5. Page 9, Line 7 - ", such as changing" –> ", such as changing" (two spaces)

6. Page 15, Line 4 - "thereis" –> "there is"

7. Page 15, Line 15 - "may have for instance" –> "may have"

8. Page 20, Line 21 - "for instance is that" –> "is that"

9. Page 23, Line 3 - "Most of the ensemble members" –> "Most members"

10. Page 26, first line of table - there is a tab separating -1160 from 530.

11. Page 36, Line 11 - double space –> single space

12. Page 38, Line 19 - "Terr" –> "TerrC"

13. Page 38, Line 27 - remove /

---

## Referee Comment (RC3) · Anonymous Referee #3 · 16 Feb 2018

This article presents an ensemble of Last Glacial Maximum (LGM) simulations using the GENIE intermediate complexity model with varying parameter values. The model simulates the carbon cycle allowing the authors to compare the CO2 values obtained from the model with the 90 ppm decrease from ice core data and analyses carbon stocks, in addition to the study of climate variables. They select two subsets of simulations with increasing constraints and analyses the changes obtained in these simulations in terms of temperature, precipitation, sea ice, ocean circulation and carbon stocks with respect to the pre-industrial. The CO2 drawdown in most simulations is

due to an increase of carbon storage on land and in the lithosphere while the ocean gets depleted in carbon.

Using an ensemble of simulations to study the change of climate and of the carbon cycle during the LGM is a great idea and GENIE is an adequate model as it is fast enough for the long simulations required. However, concerning the carbon cycle part, the lack of carbon isotopes in the simulations prevents any real conclusion to be drawn on the plausibility of the results obtained and the likelihood of the associated mechanisms and carbon stocks changes. As the carbon isotopes are already incorporated within GENIE, providing that this is feasible, I suggest to rerun the simulations and redo the analyses with the isotopes (at least carbon 13) in comparison to data, which is crucial to properly evaluate the results.

General comments

1. As I said before, the main point is the absence of carbon isotopes which precludes any strong conclusion to be drawn, it would be best to redo the simulations with the carbon isotopes (at least C13, if possible C14) and compare with ocean and atmosphere data to evaluate which simulations are really plausible. How long would this take?

2. From the simulations done so far, we can't know which processes are responsible for the carbon changes, it would be great to have a few additional sensitivity tests (for example taking one set of parameters from the PGACF ensemble) to evaluate the impact of each process on CO2.

3. In the figures with maps, it would be good to draw the coastlines of continents to make it easier to see where the changes take place.

4. Âń conversely Âż appears 28 times in the manuscript, it could probably be removed or replaced a few times.

5. On the bar charts, all ensembles (grey, yellow and orange) could be drawn on the same plot to avoid having two subplots, which would help the comparison between the

ensembles and reduce the space taken by figures.

6. The hypothesis that carbon stays below the ice sheets is a strong one, it could be interesting to evaluate its impact by doing one (or a few) simulations from the PGACF ensemble without carbon kept under ice sheets.

Specific comments

p.1-2: The abstract is quite long, and could be better organized, with the problematic explained at the beginning before stating what is the main scientific question raised in this article, how it is raised, and then the main results.

p. 2-3: Permafrost is not mentioned in the introduction; it would be good to include it. It would also be interesting to introduce here which data will be later used to constrain the results.

p. 6: During the second stage of the simulations, how does $CO_2$ evolve, does it stay stable?

p. 7 l. 12: In the EFPC ensemble, are the simulations at equilibrium at the end of stage 3?

p.7 l.21: The SST value is too high compared to data, how does that compare to other models? Is it in the range?

p.7 l.22 The sea ice value is given for the Northern and Southern Hemispheres, how is the comparison with data when split between the North and the South?

p. 7: The vegetation and soil carbon values are given in table 2 but are not discussed. How does it compare to data ? Is the vegetation distribution ok? Given that it plays an important role in the change of $CO_2$ for the LGM it would be good to know if the preindustrial terrestrial biosphere is well represented or if it has important biases. There is also no discussion of the overturning values given, how does it compare to other models?

[Figure]

p. 9 l. 28-29 I'm not sure I understand or agree with this sentence as the simulations are for the LGM and not the other glacial maxima in terms of orbital parameters.

p. 10 figure 1: maybe replace PRE by PI and explain it somewhere: Pre-industrial (PI).

p. 11 l.10 and following: Could you use temperature and salinity data to select ensemble members that are supported by data?

p. 15 line 10: how does sea ice distribution compare with data?

p. 20 l. 3 Is it really "than in the PGACF-16"? Is this not the ensemble that you are talking about?

p. 20 line 10: NADW instead of AABW?

Figure 9: It looks like the NADW is stronger for the LGM than the Pre-industrial , while from the text and figure 8 I understood the opposite. . .

Figure 10: could you add the PGACF-16 ensemble?

p. 23 and following: could you show a map of where the carbon is stored on land ?

p. 37: the conclusion is long and more descriptive than conclusive, it might be good to re-organize it.

---

## Author Response (AR1)

**Changes made to the revised manuscript are described in red.**

**General response**

5 We thank all three references for their constructive comments and suggestions. Overall, there seem to be two main categories of issues/suggestions:

1) Writing: (a) the aims and conclusions of the research are not communicated clearly enough and (b) the comparison against observations/proxies is too detailed for an EMIC. We agree and have addressed

10 as detailed below. We also agree that the strengths of the study have not been sufficiently emphasised, and address this as well.

2) Analyses: It is suggested that doing additional analyses would be helpful if feasible (specifically incorporating 13C and performing sensitivity analyses). These analyses would be interesting, but are unfortunately infeasible as the research was performed during a PhD and there are no resources for

- 15 additional simulations. Note that applying all our ensemble members to the simulation of one stage represents at least 0.5 CPU years of computing. We will instead deal with these helpful suggestions through an existing sensitivity not previously discussed, and re-emphasise caveats as appropriate, and as detailed below.
- 20 We note that this paper was intended as part one of two related papers. This paper describing the relationships between ensemble outputs, and the second, currently being finalised for submission, describing dependencies on ensemble parameters to isolate mechanisms. We are happy to provide a draft of the second manuscript as part of the review process if deemed necessary, as it may help to put this current paper in perspective. Even accounting for the increased focus of the revisions, however, we
- 25 still believe there to be far too much material overall to cram into a single paper.

**Response to referee #1**

30

We thank referee #1 for useful and constructive comments. Our response is provided below in black, with original referee comments in blue.

**General comments**

35 I like it very much that the authors provided an comprehensive review of the mechanisms that governing the LGM atm. CO2 drop. And I really appreciate that an extensive body of related work are mentioned during the discussion of model results. However, I do have several major concerns.

**1) Research aim:**

P1 I15-16 and P3 I19 suggest that the aim of this study is "to investigate the causes of the LGM atmospheric CO2 drop". To me, this research aim is not appropriate. Previous studies have already proposed some hypotheses regarding the mechanisms

- 5 that governing the decrease of atmospheric CO2 concentration relative to preindustrial (as have been summarised in the Introduction). What is not yet clear is the interplay of these mechanisms and the relative quantitative contribution of each mechanism to the LGM atmospheric CO2 drop. In this study, the authors did not explicitly propose any new hypothesis. Yet, the quantification of the contribution of different mechanisms
- 10 using GENIE is not possible due to the simplification of the model and due to that many processes are not accounted for.

We agree that the research aim should be reframed. We are using a large ensemble of LGM forced simulations to explore changes in physical and biogeochemical variables thought to influence the

- 15 drawdown of  $CO_2$ . We now make it clearer that the objective is not to explain definitively the causes of  $CO_2$  drop as we strongly believe the inevitable uncertainty surrounding simulations makes it impossible to accurately resolve the relative contributions of individual processes in models of any complexity as a result of irreducible uncertainty regarding processes. Instead our aim is to account for as many sources of model uncertainty as we can and from there describe the model output space we see. Intermediate
- 20 complexity is demanded for such an uncertainty-based study, which carried out 471x40,000-year simulations, intractable in high complexity models.

**We rephrased our research aim in the abstract and introduction, in the manner proposed above.**

- 25 I think a more specific research aim/question is needed for this study. When setting up the aim/question, the authors might consider: What are the novel aspects of the model or the EFPC2 ensemble? In this manuscript, is it the first time that an interactive carbon cycle model is applied to LGM? Is it for the first time the sensitivity to process parameters is investigated for LGM climate (Holden et al 2013 pointed out that EMICs are
- 30 important tools for exploring sensitivities and quantifying uncertainty)? Which mechanism( s) the authors would like to focus on? terrestrial carbon preservation? carbonate weathering?

Agreed, the revision addresses this and the introduction lays out the research objective and novelties of our approach, being:

 Investigating the range of physical and biogeochemical changes (and hence implicitly also specific mechanisms) which may have accompanied the LGM atmospheric CO2 drop, when taking into account model uncertainty in a large ensemble approach.

- Attempting to simulate the LGM atmospheric  $CO_2$  drop with the simulated  $CO_2$  feeding back to the simulated climate, which is still infrequently done in LGM  $CO_2$  experiments, and the first time it is done with GENIE-1.
- Simulating the burial of land carbon by glacial ice sheets: there have been only a limited number

of studies doing this and none that have attempted it in an LGM equilibrium experiment set-up.

The novelties of our approach have been laid out in the introduction, in the manner proposed above.

2) Key findings/conclusions:

10 The current version of the Conclusion reads more like a summary of the model results. It's not clear what are the key findings of this study. I think the key message would become clear once the research aim/question is given.

Agreed, the revision will address this and summarise the key conclusions, being:

15

5

- Despite our extensive exploration of model uncertainty, we struggle to achieve -90 ppmv. We attribute this to the potential LGM CO2 drivers not included in the model as well as error in the model's process representations (and which was not captured by the ensemble). The total effect on is estimated at up to ~60 ppmv, yielding an acceptable, or "plausible", range of ~-30 to ~90 ppmv.
- 20

30

- Within this plausible  $\Delta CO_2$  range, there are multiple potential  $\Delta CO_2$  "solutions", simply in terms of the sign and magnitude of physical and biogeochemical changes. However, plausible  $CO_2$  is more frequently associated with some changes than others, namely: decreasing SSTs, increasing sea ice area, a weakening of the AMOC, a strengthening of the AABW cell in the Atlantic Ocean,
- 25 a decreasing ocean biological productivity, an increasing CaCO3 weathering flux, an increasing terrestrial biosphere carbon inventory and an increasing deep-sea CaCO3 burial flux.
  - The paper focuses on these dominant changes: (1) showing that the change in terrestrial carbon is positive both because of ice sheet carbon burial and reduced soil respiration, and (2) suggesting ways in which the other physical and biogeochemical changes may have occurred, based on their spatial patterns and relationship to other changes.
  - The dominant changes are also found to be broadly consistent with observations, and based on a qualitative attempt to reconcile the sign of the terrestrial carbon change with carbon isotopes, we show that a positive terrestrial carbon change is not immediately in contradiction. However, a quantitative assessment would be needed to obtain a more definitive answer. Since a detailed
- 35 and comprehensive assessment of modelled results against observations was not an objective of the current paper, this is a direction for future research.

The key conclusions have been incorporated in the revised conclusions section.

3

3) Comparison of model results to field/proxy data or to results of other models: A large body of text is devoted to the comparison between model results and data and to the explanation of differences between the two. To me this is a bit overdone. Holden et al. (2010), who used the same model and had the same principle for the design

5 of the ensemble, clearly stated that the ensemble is developed to reproduce the main features, but not the precise observation.

We agree and will compare modelled results in less detail in the revision, focusing on spatial patterns at a coarser resolution, and making it more explicit that globally aggregated estimates are to be treated in a similar way.

We have reduced the comparison of model results to observations/paleo-proxies and other models, in the manner described above.

Specific comments (the following comments mainly concern the suggestions for improving

15 the structure and presentation of model results.)

**Abstract**

P1 I15-16: It would be helpful to specifically state the research aim here. We agree that our research aim needs to be clearly laid out at the beginning of the abstract. We propose to briefly introduce the research problem, followed by our aim, and then our key results.

20

**We have rearranged the abstract in the manner described above.**

Introduction

25 P3 l27: "unknown error" – does it mean the discrepancy between e.g. the modelled circulation and that obtained from proxy data? To me it sounds like a mistake one accidentally made in the model.

Our meaning here is that there is no direct two-way, persistent, interaction between ocean and ice 30 sheets (as is the case in some higher complexity models), and the error caused by the absence of this process is unknown.

P3 I28: "unknown error" – Maybe "missing processes" is more suitable here? Agreed, this would be clearer and will be changed.

35

We removed the following sentence: "It is recognised, however, that what constitutes an implausible or plausible climate state is somewhat subjective, and that not all sources of model uncertainty, such as the unknown error due to interacting ice sheets and ocean circulation, or the unknown error due to

potential  $\Delta CO_2$  mechanisms not included in the model (although see below for some estimates), are accounted for."

The concern about not all sources of uncertainty being accounted for is largely addressed through: "Despite our ensemble varying many of the parameters thought to contribute to variability in glacial-5 interglacial atmospheric CO2, not all sources of uncertainty can be captured, and this is reflected in our simulated DCO2 distribution. We estimate that up to ~60 ppmv of DCO2 could be due to processes not included in our model and error in our process representations (see section 2.4 for details)."

Please add at the end of the Introduction an overview of the upcoming sections, viz., 10 what will be presented in each section.

Yes, thank you for the useful suggestion. We will include a paragraph detailing our main sections: (i) description of the model, ensemble and simulation set-up used; (ii) brief analysis of preindustrial

- 15 simulations to verify that plausible given that the way we spin-up the model is not exactly the same as in Holden et al., 2013a, and we also wanted to include evaluation of a few metrics not used to constrain the original ensemble; (iii) analysis of LGM simulation results to determine range of physical and biogeochemical changes that may have accompanied LGM drop. This section includes direct drivers of  $CO_2$  change (e.g. terrestrial carbon inventory change) as well as indirect drivers (e.g. precipitation
- 20 change).

**The new sections are broadly as outlined above.**

We have added an overview section at the end of the introduction

25 Method

P5, Table 1: what is OLR? Outgoing longwave radiation

We have replaced OLR with Outgoing longwave radiation

P6: I think a table or a flow chart summarising the conditions for the four stages would 30 be helpful for readers to understand the set-up of experiments.

We made some minor changes to the text and feel like this clarifies the description significantly.

How long is the stage 2? 10,000 years 35

> Is the total carbon inventory (that is, sum of atmospheric, terrestrial, ocean and lithospheric carbon inventory) unchanged over the four simulation stage? We checked that carbon is being approximately conserved over stage 3 in PGCAF-16, by calculating the sum of ATMC, TERRC, OCEANC

and LITHC for each of the 16 ensemble members. We found that each sum, in absolute terms, was less than 10 PgC.

**LGM ensemble simulations**

5 P9: Please explain here why the subset PGACF-16 is needed. This can be done by

moving p37 l30-32 to section 4.1. We agree that the need for PGACF-16 should be explained more clearly when introducing it. The main reason is to determine whether the patterns identified for PGACF as a whole also appear to hold for the lower end of the  $\Delta CO_2$  range more strictly. Assessing variation across the plausible  $\Delta CO_2$  spectrum is not a dominant objective of the study, however, and we will

10 update the writing to reflect this.

We explain the need for subset PGACF-16 towards the end of the Introduction section.

**It would be very helpful if a brief overview of the following text of this section 4 is**

- 15 presented: which variables of which set will be presented. Agreed. In the current manuscript, for all model outputs of interest, we discuss PGACF, PGACF-16 and EFPC2, albeit focusing on the first ensemble. The other two ensembles are discussed to show extent to which similar to PGACF, despite their different  $\Delta CO_2$  ranges and number of ensemble members included. In the revised manuscript, we will reduce discussion of PGACF-16 and EFPC2 and have this reflected in the overview section.
- 20

35

Instead an overview section in Section 4, we include the above information towards the end of the Introduction section.

**P25 I6-7: this is not true because in Table 6 none previous observation/model data**

- 25 study shows negative delta OceanC. We agree that this statement is misleading as two of the studies make use of soil carbon measurements, and the third is a modelling study which does not explicitly report the change in ocean carbon. The statement was based on the assumption that if ~90% of the atmospheric perturbation caused by the reported increase in terrestrial biosphere carbon was removed by oceans and sediments, the change in ocean carbon inventory would still be negative
- (between ~-9 and ~350 PgC), after adding remaining carbon to be lost from the atmosphere to the 30 ocean. We will rephrase our statement in the revision.

The statement has been replaced with "we can see that the mean  $\Delta$ TerrC is only aligned with a handful of estimates and no studies so far report a negative  $\Delta$ OceanC. Instead,  $\Delta$ OceanC is estimated to be positive, primarily based on carbon isotope data".

P36 bottom: Fig. 18 should be Fig. 17. We will update the figure number. The figure number has been updated.

P36 [3: "North Atlantic" and "North Pacific" should be switched. We agree that the text is confusing here and we will clarify this by replacing "large increases" with " maxima" We implemented the aforementioned edit.

- Comparison of global-integrated numbers and spatial distribution between model results 5 and data: I suggest to first compare the spatial distribution and then the globalintegrated numbers because the latter is just the sum of the former. Agreed, we will switch the order in the revision.
- 10 While it is a useful suggestion, we came to the conclusion, in revising the text, that the latter would be easier to follow with the globally integrated numbers presented first.

Colour slots showing spatial distributions of variables, e.g. Fig 3, 6, 7, ...: please add contour line for land-ocean border. These will be included in the revision.

**15**

Plots showing standard deviation in e.g. Fig 3, 6, ...: These plots are shown but never mentioned/used/discussed. So please consider move them to a supplementary information file - there are already many figures in the manuscript. Agreed, we included the standard deviation plots for information but they are not central to our arguments. We will move them to a SI file in the

20 revised manuscript.

**Conclusions**

P39 I15: The positive delta TerrC has been discussed and justified many times through out the manuscript. Thus, I have the impression that this is one key finding the authors

25 would like to stress. I think this is a bit dangerous because this point is not well supported by data. The authors also seem de-stressing this point several times by stating there are other 4 ways of "achieving a plausible delta\_CO2 interns of the sign of individual carbon reservoir changes (although Table 5 suggests those 4 ways are much likely to occur). I have to say I am confused by the above statements.

**30**

We agree that the statements are confusing and we will clarify these in the revision as follows: we find that plausible  $\Delta CO_2$  can be achieved in 5 different ways in terms of the sign of individual carbon reservoir changes. However, positive  $\Delta$ TerrC combined with positive  $\Delta$ LithC and negative  $\Delta$ OceanC is by far the most common way, encompassing 89% of simulations, and we focus our discussion on these

simulations. This includes proposing explanations for the positive  $\Delta$ TerrC and discussing the change in 35 the light of observational constraints.

The aforementioned revisions have been implemented.

P39 I39 - P41 I15: I understand that it is a pity that carbon isotopes were not simulated. However, I don't think it is appropriate to extensively present inferred results in the Conclusion section.

Agreed. In the revision, we will move the discussion of carbon isotopes, focused on the positive  $\Delta TerrC$ ,

5 to the section on carbon reservoir changes.

Instead of moving the discussion of carbon isotopes to the section on carbon reservoirs, we created a separate sub-section for it, at the end of the results and discussion section.

**Response to Pearse Buchanan (referee #2)**

10 We thank Pearse Buchanan for his very detailed and constructive comments. As above, our replies are in black, and original comments in blue.

**1 General comments**

This study provides some unique perspectives on the glacial drawdown of atmospheric

- 15 pCO2. This problem has been at the forefront of climate research for many decades. So far, many mechanisms have been proposed, but a recipe of changes that is physical and biogeochemically consistent with proxy records and known mechanisms is still elusive. The authors of this study set out to try and achieve this difficult task. The authors employ a unique set of methods to attack their exploration of
- 20 what might plausibly reduce atmospheric pCO2 under glacial conditions. The method involves simulating 315 individual parameter sets using the same Earth System Model (ESM) of intermediate complexity in four stages. From what I can tell, each stage involves the full 315-member ensemble, unless numerical instability problems were encountered. Each ensemble member was therefore independent from another at all
- 25 stages through the study. Each stage was initialised from the final year solution of its previous stage, except stage 1 which I am unsure about what fields have been used for initialisation. The only boundary conditions that were prescribed to the Earth System Model that I can tell were orbital parameters, aeolian iron deposition rates, detrital flux rate to ocean sediments, and ice-sheet fraction and their orography. The 4 stages are
- 30 as follows:

1. Stage 1 (PI 10,000 years): relaxed pCO2 to 278 ppmv; no interaction between carbon reservoirs; conserved alkalinity in the ocean.

2. Stage 2 (PI 10,000 years): freely evolving pCO2; interacting reservoirs; freely evolving ocean alkalinity.

35 3. Stage 3 (LGM 10,000 years) freely evolving pCO2; interacting reservoirs; freely evolving ocean alkalinity; ice-sheet growth and corresponding sea level loss

years 0-1000. 4. Stage 4 (LGM 10,000 years) freely evolving pCO2; interacting reservoirs; freely evolving ocean alkalinity.

- 5 We will revise the text to clarify our method, which is the following: In total, 471 runs were applied to preindustrial and then LGM simulations. At the LGM (stage 3), however, 75 runs either "snowballed" or crashed, leaving 396 ensemble members (EM). By "snowballed" we mean that the runs predicted highly implausible global annual SATs, between ca. -67.8 and -56.8°C, likely as a result of global or near-global sea and land ice cover developing in the
- 10 simulations. Out of the remaining 396 EM, we then removed those simulations with preindustrial CO2 outside of 280 ± 10 ppmv (18 EM), and subsequently any EM that predicted large abrupt changes in atmospheric CO2 over the LGM simulation that likely caused by instabilities rather than by some physical mechanism (63 EM).
- 15 Regarding inititialisation, the model spins up from its default state. Only the CaCO3 weathering fluxes are taken from Holden et al., 2013a, diagnosed from their 25 kyr preindustrial spin-up. However, the prescribed fluxes are automatically and continuously rescaled in the model to balance the modelled CaCO3 burial rate. Thus, their value is not important (Ridgwell, 2017).
- 20 The boundary conditions in our simulations are indeed orbital parameters, aeolian iron deposition rates, detrital flux rate to ocean sediments, and ice sheet fraction and their orography. For the latter, we will make it more explicit in the revision that since our ice sheets are as described in Holden et al., 2010b, only the Laurentide and Eurasian Ice Sheets are allowed to change from their preindustrial form and rather than being extracted uniformly, freshwater to build the LGM ice sheets is routed from the Atlantic Pacific and Arctic accuming modern topography.
- 25 Atlantic, Pacific and Arctic, assuming modern topography.

30

35

I. 24-26 on p. 4 should also say that a freshwater flux scaling parameter (FFX) value of 1.5 is applied in GENIE to correct for un-modelled isostatic depression at the ice-bedrock interface due to ice sheet growth, and for assuming a fixed land-sea mask. We vary the parameter in the ensemble to capture the uncertainty in the magnitude of the glacial sea level drop and its effects on the carbon cycle.

The revised manuscript now reads: "(FFX) scales ice sheet meltwater fluxes to correct for un-modelled isostatic depression at the ice-bedrock interface due to ice sheet growth, and for assuming a fixed landsea mask. We vary the parameter in the ensemble to capture the uncertainty in the magnitude of the glacial sea level drop and its effects on the carbon cycle."

The first paragraph of section 2.3 was modified to clarify what happens in stages 1 and 2.

We have described our ice sheet setup in more detail.

For clarity, we moved the section regarding which ensemble runs were removed to section 2.3 and made some minor changes to our wording.

- 5 Following on from this methodology, the authors lead the reader through the results in a methodical manner, taking on changes in some key environmental variables. In each address of a variable, the authors undergo a detailed assessment of PI versus LGM changes in their model and in proxy records. This is done both in a global sense and a regional sense, sometimes being highly specific. The ground-truthing and
- 10 comparison to observations and other studies is commendable. It one of the largest comparisons that I have come across in a modelling paper. However, the sheer size of comparison makes the paper unwieldy, and loses the major findings amongst the details. It is also unexpected that the authors concentrate on highly specific regional comparisons because they use an ESM of intermediate complexity that will simply not
- 15 perform well in many ways. Rather, the authors should focus on the sheer size of their parameter spread and in diagnosing the effect that certain changes have on carbon and climate. I therefore suggest that the authors make an effort to reduce the emphasis on comparison, and focus on how their interesting results might explain the LGM carbon cycle in only a broad sense.
- 20

Agreed, in the revision we will modify the text to better take into account the complexity of our model, and to better emphasize the links between the changes we see and CO2.

In the revised manuscript, we reduce the level of comparison and attempt the link between non-CO2 variables and CO2 more apparent.

It occurs to me that the authors seem to forget some obvious strengths of their study. Their parameter space is enormous, includes an interactive carbon cycle with carbon-climate feedbacks, as well as many other processes (sea level) that have not

30 been included in other model studies of the LGM. This is of interest to the LGM pCO2 drop problem. I suggest that the authors try and convey these strengths more clearly in their abstract and the final paragraph of the introduction.

Agreed, the abstract will be reduced to one paragraph and the aim and strengths of our study will 35 introduced at the beginning at the paragraph. We will also clarify these in the current final paragraph of the introduction section, and which now focuses on a description of the ensemble. The abstract has been reduced to one paragraph, with the aim stated at the beginning. In the introduction, we state more clearly our advantage of big parameter space, interactive carbon cycle and uniqueness of carbon burial.

- 5 A particularly interesting result of this study is the increase in terrestrial carbon under LGM conditions. The authors more often than not find that LGM conditions are associated with greater carbon held in soils because (1) ice-sheet growth covers previously fertile regions and traps the carbon in the soil, (2) soil respiration rates decrease more than primary production, and (3) terrestrial carbon in exposed continental
- 10 shelves due to sea level drop increases. The authors do not simulate point 3, and there is a possible methodological problem with point 1, in that the authors grow the LGM ice sheets from pre-industrial to LGM state in only 1,000 years. It is unclear in the manuscript presented here whether this rapid growth of ice sheets that is prescribed overtakes highly productive regions that have not yet evolved to become arid tundra
- 15 land, which have lower carbon content in the soils. A rapid trapping of soil carbon by the unrealistic growth of the ice sheets (almost 100x faster than in reality) would likely overestimate the terrestrial carbon reservoir at the LGM. The authors state in the methods that "Sensitivity simulations were performed to verify the simulated equilibrium state is insensitive to the choice of timescale of ice-sheet growth", and yet the authors
- 20 do not present any evidence of this sensitivity. I am therefore sceptical of the validity of this result, which I would say is their most important and interesting contribution to the field.

We agree that our statement on sensitivity simulations is confusing, and realise here that the following statement may also need updating in the revision to avoid confusion: "The increases in terrestrial biosphere carbon are predominantly due to our choice to preserve rather than destroy carbon in ice sheet areas. However, the ensemble soil respiration also tends to decrease significantly more than net photosynthesis, resulting in relatively large increases in non-burial carbon". We briefly address the latter point first.

30

The statement is based on an analysis of terrestrial carbon partitioning between ice-sheet and non-ice sheet carbon pools in PGACF-16 and from there inference about what happens in PGACF. If the LGM burial carbon inventories in PGACF-16 were to be removed, DTERRC would be negative in 13 out of 16 simulations, despite the fact that terrestrial carbon also increases outside of the ice sheet areas in 15

35 out of 16 simulations. However, it is not strictly correct to attribute the carbon inventories that are buried to the burial itself. It is the combination of our ice sheet carbon stocks increasing rather than decreasing when exposed to LGM climate, and our choice to preserve rather than destroy this carbon. If most of the carbon that was present in ice sheet areas at the end of the preindustrial runs had been lost to climate forcings, it would not matter much whether the remaining stocks had been destroyed or preserved. From the literature, it is not clear how much of the carbon in ice sheet areas is thought to have been lost strictly in response to ice sheet "bulldozing" versus climate impacts. We will update the revised manuscript to reflect this logic.

- 5 With regard to the statement "Sensitivity simulations were performed to verify the simulated equilibrium state is insensitive to the choice of timescale of ice-sheet growth", This refers to the sensitivity of the LGM burial carbon amount to the timescale of ice sheet growth. To test this, we took one ensemble member from PGACF-16 and applied it to two 11,000-year LGM simulations with 1000-year ice sheet growth and 10,000-year ice sheet growth respectively. Both
- 10 simulations were started from the end of the same equilibrium preindustrial simulation. We then compared the LGM burial carbon inventories in each run and found that these differed by just 34.2 PgC, which, assuming ~90% of removal of atmospheric  $CO_2$  perturbation by ocean and sediments, amounts to a mere 1.6 ppmv  $CO_2$  difference. Our assumption is that applying the same ensemble member to a transient simulation of the full glacial cycle (and therefore a more realistic ice sheet build-up history)
- 15 would not have yielded a dramatically different burial carbon inventory. Given that we did a sensitivity experiment with just one ensemble member, we are also assuming that the diagnosed sensitivity is roughly representative of the ensemble as a whole. In the revision, we will highlight these as potential caveats.
- 20 With regard to the sign of the ice sheet carbon response at the LGM, we argue that it was not necessarily negative and analysis of PGACF-16 indeed suggests that the sign is consistently positive. Most of this increase is due to a reduction in soil respiration as vegetation carbon change is only positive in one simulation. We also find that extending the timescale of ice sheet growth increases rather than decreases the burial carbon inventory. A likely explanation is that the soil carbon inventory was not yet in equilibrium by 1000 years.

**We have clarified our description of LGM terrestrial carbon changes, and our statement on the sensitivity simulations.**

- 30 Finally, the conclusion needs complete re-writing. It reads as a re-stating of the methods and then the results, which is simply not useful for the reader. The authors do discuss carbon isotope changes, which are appropriate records to discuss given the main result of an increase terrestrial carbon reservoir. I strongly suggest that the authors re-assess their strengths and major findings, present them concisely, and
- 35 provide some comments regarding how a higher terrestrial carbon reservoir and low ocean reservoir could have occurred despite most studies indicating the opposite. If this is possible, then this would be a useful contribution to the field.

Agreed, our revised conclusions section will summarise our research objective and the novelties of our approach and then the key conclusions, detailed in the author reply on p. 3. Most of the discussion on carbon isotopes will be shifted to the section on carbon reservoir changes.

**5 The aforementioned changes have been included in the revised manuscript.**

Major revisions are needed. If the authors can provide some evidence that changes in the timescale over which the ice-sheets are grown do not play a large role in determining how much carbon is present in the terrestrial reservoir, and the manuscript is associated below that below the below the

 $10\$  is rewritten addressing the points above and below, then I advocate publication.

**Some other general points:**

• Regarding the figures, I strongly suggest that the authors overlay the red

- 15 (PGACF-16) over the yellow (PGACF) bars in the histograms. This would reduce a lot of unnecessary replication in the figures. The authors could use a transparency setting to ensure that the red and yellow are easily seen if they have the same number of experiments (frequency). Agreed, we will include these changes in the revision.
- 20

• The writing is generally okay, but this manuscript definitely requires re-writing in some places. There are too many adjectives and unnecessarily difficult acronyms. OK, we will revise this aspect of our writing.

**25 We have sought to make the writing more concise and change acronyms where relevant (e.g. below).**

• The presentation of results suffers from the use of opaque acronyms that are easily forgotten after reading the methods section. I strongly suggest that the references to EFPC2, PGACF and PGACF-16 are changed to be more interpretable.

30 PI315, LGM104, LGM16... or equivalent, would be much more helpful. Agreed, the proposed names will be used in the revision.

We changed EPFC2, PGACF and PGACF-16 to ENS315, ENS104 and ENS16 respectively with "EN" denoting ensemble and the number, the associated number of members.

**35**

**2 Specific comments**

**Abstract**

The abstract is quite long. The message that this study delivers could be

made considerably more poignant if it were condensed for the reader. The authors make some very interesting findings, such as the increase in non-burial carbon in the terrestrail reservoir due to the slow-down in respiration" and "there are 5 different ways to achieve an atmospheric pCO2 drop". These findings (mostly in para 2) should be

- 5 the focus of the abstract, and I advocate for the more technical aspects (para 1) and comparisons with observations (para 3) be removed. In fact, the entire paragraph 3 of the abstract could be reduced to one important sentence without affecting the findings of this study.
- Agreed, we will shorten the abstract, highlighting our key conclusions, with brief sentences on comparison against observations, and before this our research problem and aim.
   The abstract has been rewritten in the manner described above, and reduced to just one paragraph.

**Introduction**

- Page 3, Line 13 "dissolve organic carbon inventory" of what? the ocean? soils? Ocean
   We have made the connection to the ocean explicit in the revision: "decreasing dissolved organic carbon inventory due to a more stratified deep ocean"
  - Page 3, Paragraph 2 This entire paragraph simply lists the changes that may
- 20 be assocaited with a glacial ocean. If these mechanisms are to be called upon by the authors, they should be accompanied by at least a brief discussion of why they influence atmospheric pCO2. For any non-specialist of palaeoceanographic literature specifically relating to the LGM, this paragraph is totally opaque. I would either expand on these mechanisms or remove entirely when accounting for them
- 25 in the discussion of the results. Agreed, we will briefly describe the referenced mechanisms in the revision.

**We have provided brief descriptions of the referenced mechanisms.**

30 • Page 3, Line 19 - "utilises an ensemble of sets of parameters" a bit clunky. What about "uses a large ensemble (471 parameter sets)"? OK, we will include this change in the revision.

**This sentence got removed as we re-wrote the introduction.**

**35**

• Page 3, Line 23 - probably no need to mention Holden (2010a) or cite other literature. Just state philosophy. OK.

We simply state the philosophy in the revised manuscript.

• Page 3, Line 26 - move this to methodology. Not necessary here. Agreed, this would make the message clearer.

5 We have deleted this section from the revised manuscript.

**Methods**

• Table 2 - define acronym SAT in caption. What is EFPC2 and EFPC? Must define these here or introduce the Table later on. It is true that the acronyms are difficult to remember and we

10 will change EFPC2 and EFPC to PI315 and PI471, and also add surface air temperature to the caption.

We have replaced "SAT" with "surface air temperature" in the table. We do not refer to EFPC by any acronym in the revised manuscript and as such no renaming was required. EFPC2 on the other hand is referred to as ENS315 (and as explained above).

15

• Page 5, Line 5 - please clarify what a closed biogeochemistry system is. Does it mean no interaction between land-ocean-atmosphere-lithorsphere reservoirs? It means  $CaCO_3$  weathering and deep sea sediment burial forced into balance, no sediment-ocean interactions. We will clarify this in the revision.

20

We have clarified this in the revision.

how are these reservoirs initiated in Stage 1? Using the fields from Holden 2013a?

25

30

As described above (general comments), the model spins up from its default state and takes the weathering fluxes diagnosed from 25 kyr preindustrial spin-up of Holden et al., 2013a.

• Page 7, Lines 1-3 - I don't think it's useful to mention this. Yes, we agree and will delete the sentences from the revision.

These sentences are excluded from the revised manuscript.

**Preindustrial simulations**

 Page 7, Line 5 - First of all, it would be helpful to change the title of section 3 to "Results: Preindustrial simulations". OK, we will make the change.

In the revised manuscript, we keep the title of Preindustrial Simulations but the section now is also numbered 3.1 and falls under Section 3. Results and discussion.

• Page 7, Lines 7-12 - Why present results and talk about EFPC ensemble when the authors only use the EFPC2 315-member ensemble? It seems to be an unnecessary inclusion that confuses the reader, rather than helps understanding.

5 I think given the length of this study that it would be helpful to simply cut any inclusion of EFPC 471-member ensemble and simply present the results of the 315-member ensemble.

We use EFPC to refer to both our 471-member ensemble and that of Holden et al., 2013a. Our EFPC ensemble is mentioned in the context of explaining where the EFPC2 ensemble came from, and we

- 10 subsequently compare the response of the latter ensemble with that of Holden et al., 2013a (Table 2), to verify that the values taken by the eight modern plausibility metrics are similar to their values in Holden et al., 2013a. To avoid confusion in the revision, we will move the details of the "ensemble filtering" to a SI file, alongside Table 2 as suggested below.
- 15 For enhanced clarity, we decided to move the details of the ensemble filtering to section 2.3 and move Table 2 to a SI file.

• Table 2 - change "31/12" to 31st Dec. OK, will include in the revision.

20

We have incorporated the change in the revised manuscript.

In revising Table 2 – we spotted a mistake. Namely, that the values we reported for the Atlantic Overturning stream function maximum and minimum were erroneous. These have been rectified in the revised manuscript.

25

• Table 2 - change "wt% CaCO3" to "wt% ocean CaCO3" wt% refers to surface sediment wt% . We will clarify in the revision.

The revised manuscript incorporates the above clarification.

30

• page 7, Lines 15-23 - Tables 2 and 3 could be moved to supplementary material. Table 2 discussion could be reduced to a single sentence saying that the preindustrial simulations of the 315-member ensemble reproduced all aspects of the Holden 2013a simulations. Table 3 discussion could be reduced to note that

35 there was good agreement with observations of ocean carbon inventory, SSTs and sea ice extent relative to known values.

• Based on what I've said above, I suppose that this section could actually be reduced to one paragraph, or completely removed if the authors wished to use address PI conditions via comparisons with LGM conditions in the next section.

Agreed, we will reduce discussion of the results, and move Tables 2 and 3 to a SI file.

The section has been reduced to a couple of sentences and Tables 2 and 3 have been moved to a SI file.

**5 LGM ensemble simulations**

• Page 9, Line 7 - "104 ensemble members". Are these presented in yellow in Figure 1? If so, mention it here in the text. Yes, "as shown in yellow in Fig. 1" will be added in the revision.

10 This section has now been moved to a new section, Section 2.4, titled Ensemble subsets.

• Page 9, Lines 7-16 - These sentences are confusing for the reader. I understood once reading further on in the paragraph that you do not include these processes in the model, and you are saying that their inclusion would push LGM pCO2

15 decrease even further, which justifies your choice of a -30 ppmv threshold to define a successful solution of LGM conditions. However, this is not clear. Please re-write.

• Page 9, Lines 16-26 - This needs to be moved to the methods section. From how

20 I understand your thinking: lines 5-16 justify your choice of -30 ppmv threshold; lines 16-26 justify your methodology in treating the LGM simulations.

• Page 9, Lines 22-25 - The lower threshold of -60 ppmv should belong with your choice of the -30 ppmv upper limit. discuss these together, not separated by

25 other sentences and concepts.

With regard to the above 3 comments, we will address these jointly through revision of p9 l6-26: firstly describing our  $\Delta CO_2$  results, and secondly justifying our choice of  $\Delta CO_2$  ranges to focus on. Our general approach to analyzing the results will be laid out in the introduction.

30

In the revised manuscript, the different ensemble subsets are now described in a new, separate section (Section 2.4).

Page 9, Lines 28-33 - This is a very interesting results. Why can't you define the
mechanisms that lead or do not lead to the snowball Earth scenario? Surely if
you can define a plausible set of mechanisms need to achieve glacial conditions,
you can do the same by comparing the 471 PIs, 16 LGMs, and 47 snowballs???
This would mark a significant contribution to the field.

In this paragraph what we were trying to convey, and we will rephrase in the revision, is that in our EFPC2 ensemble we struggle to achieve LGM atmospheric  $CO_2 \le 200$  ppmv (only ~1.6% of simulations). Not included in this ensemble are 47 LGM simulations which completed but which also predicted global annual average SATs between ca. -67.8 and -56.8°C (and which we assumed are the result of global or

- near-global sea and land ice cover developing in the simulations, i.e. "snowball earth" type conditions). 5 In 96% of these simulations, atmospheric  $CO_2$  drops to  $\leq$  200 ppmv at least temporarily. We thus hypothesised that the  $CO_2$  and the "snowballing" may be linked. Establishing causal mechanisms would be interesting but analysis of stage 3  $CO_2$  time series suggests that the  $CO_2$  is far from equilibrium in many of the "snowball earth" simulations by 10 kyr. We expect different dynamics to operate in the
- 10 snowball and non-snowball earth states. We hope to investigate this further in the future, in a separate manuscript.

There are significant complexities here and interpretation is not straightforward, thus we have removed this tangential discussion from the revised manuscript.

15

• Figure 1 - Overlay red bars on the yellow bars to make 1 panel. We will implement these changes in the revision for this figure, as well as figures 2 and 4 as suggested below.

• Page 11, Line 11 - define SAT . We will specify that SAT means surface air temperature.

20

**In the revised manuscript, SAT is replaced with surface air temperature (SAT).**

• Figure 2 - Again, overlay red on yellow to reduce unnecessary replication. Use transparency perhaps to show where yellow and red are both present at the same 25 frequency.

• Page 12, Lines 4-16 - Comparisons with obs not necessary at this detail given the focus of the work and the fact that you use an ESM of intermediate complexity. I would expect a discussion of global temperature changes, with perhaps a little

bit of basin-wide, regional discussion **if** those are important points for later on. 30 Please reduce this paragraph and combine with the next.

We evaluate the spatial distributions of SAT and SST changes as these are likely to influence our CO2 solution through impacts on the solubility pump, land carbon storage etc. We will articulate this more

clearly in the revision and make the comparison with observations less detailed to reflect the focus of 35 our work and the complexity of our model. We will also combine SAT and SST evaluations into one paragraph as the two variables are closely linked.

In the revised manuscript, the description is less detailed and SAT and SST changes are dealt with jointly.

• Page 15, Line 5 - I'd say there is no relationship. Our statement of a weak relationship is based on r

5 greater or equal than 0.12, in line with our chosen 0.05 significance level. We will clarify this in the revision.

We clarify our significance level earlier in the manuscript in the revision.

• Page 15, Line 10 - too many adjectives. We will change "The PGACF ensemble LGM global annual sea

10 ice area anomaly (SIA) has a mean of" to "the mean LGM global annual sea ice area anomaly in the PGACF ensemble is"

The revised manuscript now reads "The  $\text{ENS}_{104}$  mean LGM global annual sea ice area anomaly ( $\Delta \text{SIA})$  is"

15 • Figure 4 - overlay red on yellow.

• Page 18, Lines 11-13 - confusing sentence please re-write. What we wanted to convey here is that the observed precipitation decreases are as negative as -1.1 mm in Southern Europe (SE) and Middle East (ME) whereas the ensemble mean decreases are no greater than ~- 0.5 mm in SE and increases of up

20 to +0.1 mm are simulated in ME. We will make the sentences in the precipitation section more concise and also adapt them to better reflect the complexity of model and the focus of our study.

We have reduced the level of detail in the section on precipitation changes.

- Page 19, Lines 11-15 when talking about AABW formation rates, it is better to present this in positive units. Oceanographers are familiar that the units are negative in the calculation of the streamfunction. It is less confusing for the reader to present your changes as negatives if the AABW formation rate declines. This also removes the need to explain that a positive anomaly is actually a decrease. Agreed, we will convert
- 30 our values into negative units in the revision.

• Page 20, Line 2-3 - Don't understand. I thought you said that weaker AMOC and stronger AABW was coincident with the glacial runs, these being PGACF-16? Please make this clearer. WHat are you comparing?

35

There is a typo in the text which we will fix in the revision. The sentence currently reads: "Although not show here, the PGACF-16 ensemble members tend to exhibit a shoaling of the AMOC and enhanced penetration of AABW. With regard to  $\Delta$  and  $\Delta$ , these tend to be more negative (i.e. weaker AMOC and

stronger AABW) than in the PGACF-16. The  $\Delta$  and  $\Delta$  in the EFPC2 ensemble tend to conversely be more positive".

The second mention of "PGACF-16" should be "PGACF".

5

**We have rectified this in the revised manuscript.**

• Page 20, Lines 3-8 - These relationships are made more confusing for the reader because you define AABW formation rates as being stronger when they are negative.

10 It owuld be more helpful if strong = more positive. In the revision, we will make sure that changes and relationships are reported in a way that avoids confusion about direction.

Page 20, Line 15 - Please see study of Yu et al (2014) Deep ocean carbonate

chemistry and glacial-interglacial atmospheric CO2 changes. We will add an appropriate citation.

15

We reference Yu et al., 2014 at the end of the sentence.

• Page 20, final sentence - This has already been covered above? The relationship between sea ice and ocean circulation changes has already been discussed so we agree that the sentence is somewhat

20 redundant. The aim was simply to point out that  $CO_2$  may have had an impact on the ocean circulation changes as well as vice versa. We will make this clearer in the revision, and also include discussion of the proposed link between Antarctic sea ice expansion and ocean circulation in Ferrari et al., 2014 cited below.

25 We removed the final sentence from the revised manuscript.

Also, see Ferrari

et al (2014) Antarctic sea ice control on ocean circulation in present and glacial climates. PNAS.

30

• Figure 9 - please ensure that your colour bars are the same scale! I initially thought that your LGM simulation had strong AMOCs, despite the discussion of weaker AMOCs in the text. The scales of the colour bars will be updated as part of the revision.

**• Page 23 - Very interesting result. I think that this is a unique and interesting**

contribution to the field and should be a focus of this study. We will highlight these and other key results by reorganizing the abstract in manner suggested above, and by rewriting conclusions section.

• Table 5 - why does the order change? Please clarify in caption of correct. There is no particular reason for this. For clarity, we will update the table so that column 1 shows  $\Delta$ TerrC, column 2  $\Delta$ OceanC and column 3  $\Delta$ LithC.

5 We have updated the table in manner described above.

Also,

add the subheadings "Total counts" and "% of total" or their equivalent to the other columns. OK, we will apply the subheadings to all relevant columns in the revision.

10

The table has been changed in manner described above.

• Table 5 - Please quantify the increases and decreases in this table that accompany the scenarios (i.e. mean). Ok, we will include these in the revised table.

15

• Page 25, Line 6 - Is table 6 mentioned previously? This introduction of table 6 is jarring. It is indeed mentioned for the first time on line 6. Rather than referring to it at the beginning of a new sentence, we will introduce table 6 more explicitly.

20 Table 6 now introduced as follows: "The ENS104 mean  $\Delta$ TerrC,  $\Delta$ OceanC and  $\Delta$ LithC, the signs of which are consistent with scenario 1, are reported in Table 6, alongside previous estimates from observational data- and model-based studies"

• Table 6 - Great Table. Could the acronyms for studies you cite be organised into

25 alphabetical order? Yes, we will change their order.

The acronyms have been organized in alphabetical order in the revised manuscript.

Page 25, Lines 9-10 - I'm (and probably many readers) not sure what "Carbonate compensation of the increased terrestrial carbon storage" means. It's not clear whether you refer to carbon compensation mechanism in the land or the ocean. Do you mean a loss in oceanic DIC due to terrestrial carbon storage, causing an increase in alkalinity that increases CaCO3 burial? If changing terrestrial carbon reservoir does have a direct effect on ocean alkalinity and CaCO3 burial, maybe

35 by weathering changes you account for, then please explain more fully.

We mean here carbonate compensation in response to/of the terrestrial carbon uptake: the loss of  $CO_2$  from the ocean leads to an increase in surface  $[CO_3^{2-}]$  and subsequently deep ocean  $[CO_3^{2-}]$ , which reduces  $CaCO_3$  dissolution. The latter in turn decreases  $[CO_3^{2-}]$  and increases  $[CO_2]$ , which is

communicated back to the surface, with a resultant increase in atmospheric  $CO_2$ . The modelled change in  $CaCO_3$  deep sea burial flux causes ALK to change (Kohfeld and Ridgwell, 2009). We will clarify the above in the revised manuscript.

5 The term carbonate compensation is not utilized in the revised manuscript but a description of the underlying process is still included.

• Page 25, Line 17 - I find this hard to believe. "The PGACF ensemble mean  $\Delta$ TerrC is of the same sign and order of magnitude as the  $\Delta$ TerrC predicted by Zimov et al. (

- 10 2006, 2009), Zech et al. (2011) and Zeng (2003, 2007). ΔOceanC is not directly calculated in these studies". The issue here is likely with the second rather than first sentence. We will remove the sentence from the revised manuscript as our meaning here was that was not reported in the modelling-based studies, and was not calculated in the observations-based studies.
- 15 The sentence "ΔOceanC is not directly calculated in these studies" has been removed from the revised manuscript.

• Page 25, Line 22 - and yet present day continental shelves that are inundated

- are also regions of effective carbon burial through marine export production. Yes, agreed.
- 20

• Tables 7 and 8 - Please make "% Total Land" and its relation to Ice-Sheet carbon more clear in the caption. THis takes time to figure out from the reader. In the revised manuscript, we will change the captions to something like "Ice sheet carbon: amount stored (PgC) and % of total land carbon stock"

25

As suggested below, Tables 7 and 8 have been combined into one table and this table has been renamed Ice sheet and non-ice sheet terrestrial carbon stocks, and includes a description of each column heading.

**30** • Tables 7 and 8 - I think that these tables could be combined to solely show the LGM changes.

We include Table 7 for 2 reasons: (1) to make it possible to compare our preindustrial ice sheet carbon values with those of previous studies (e.g. Zeng, 2003 or O'ishi and Abe-Ouchi, 2013); (2) to allow estimation of what the impact on  $\Delta CO_2$  would have been if this carbon was assumed to be released to the atmosphere as has been done previously (e.g. O'ishi and Abe-Ouchi, 2013). For the revision, we

- propose combining table 7 and table 8 into one table, but keeping the preindustrial ice sheet carbon inventory column. Columns 3-5 of table 7 can be removed as we do not discuss these in-text.
  - 22

**We combine Tables 7 and 8 in the manner described above, in the revised manuscript.**

We also note here that the following sentence will be changed for improved understanding (p27 l1-3): "Analysis of the PGACF-16 ensemble members' terrestrial carbon reservoirs suggests that if the

- 5 preindustrial ice sheet carbon inventory (the terrestrial biosphere carbon in grid cells to be buried by the LGM ice sheets) were to be destroyed instead of preserved at the LGM, the  $\Delta$ TerrC would be negative in all but 3 simulations (Tables 7 and 8)". What we mean here is that if we did not bury carbon,  $\Delta$ TerrC would be negative in all but 3 simulations. This includes the amount of carbon initially present in ice sheet areas (preindustrial ice sheet carbon inventory) and the subsequent increase in terrestrial
- 10 carbon over the ice sheet growth period. If only the preindustrial ice sheet carbon inventory was substracted from  $\Delta TerrC$ , the latter would be negative in all but 7 simulations. As a reminder, one of the 16 simulations predicts negative  $\Delta TerrC$  to begin with.

The revised manuscript now reads: "However, if this "extra" carbon (accumulated in response to 15 climate forcings), and the carbon already present in the ice sheet areas at the end of the preindustrial spin-up, were to have been destroyed rather than preserved, ΔTerrC would be negative in all but 3 simulations, as opposed to positive in all but one simulation (Tables 7 and 8)."

• Page 27, Lines 1-16 - The relationship between ice-sheet carbon, non-ice sheet

20 carbon and soil burial carbon needs to be made clearer. We will clarify this through the use of subscripts: , , , etc.

We have re-arranged the text on ice sheet vs non-ice sheet carbon changes and this will hopefully make the text easier to read.

25

• Page 31, Line 7 - re-write sentence please. We re-write the sentence as "The mean LGM total POC export flux anomaly () in the PGACF ensemble is"

The manuscript now reads "The ENS104 mean LGM total POC export flux anomaly ( $\Delta POC_{exp}$ ) is -0.19 ± 30 1 PgC yr-1".

• Page 31, Lines 14-15 - Also the relationship with AABW production and decreased AMOC, which you have just discussed. Agreed, we will update this sentence to include sea ice effects on ocean circulation.

35

The revised manuscript reads "One possible mechanism is enhanced deep ocean stratification due to increasing AABW formation leading to not only more efficient trapping of DIC at depth (see above), but also nutrients and therefore reduced availability in the euphotic zone. All else held constant, a weaker

and shallower AMOC cell would also inhibit the transfer of nutrients from the deep ocean to the surface."

- Page 33, Section 4.5 This tight description of the main effects is what the other
- 5 sections should emulate, and would tighten up the manuscript considerably. Agreed. The discussion of spatial changes will be considerably reduced in the revised manuscript.

We reduced the text in other sections but to make section 4.5 more consistent with the other sections, we removed the sentence that says: "The total effect of varying GWS over its full range is ~ 40 ppmv
(Fig. 15)", as well as fig. 15.

• Page 35, Line 3 - Again, this sentence is awful to read. Too many adjectives. Agreed. We will replace it with "the mean global deep-sea  $CaCO_3$  burial flux anomaly ( $\Delta CaCO3_{bur}$ ) in the PGACF ensemble is"

15 The revised manuscript reads "The  $ENS_{104}$  mean global deep-sea  $CaCO_3$  burial flux anomaly ( $\Delta CaCO3_{bur}$ ) is".

Page 36, Line 7 - I don't see a decline in the figure, just no change. The current colour legend indeed makes it difficult to distinguish between no change, small positive and small negative changes. We will
 center the legend on white and update the text to reflect the plotted changes.

We reduced the level of detail in the comparison against observations, in line with other sections.

**Conclusion**

- Page 38, Line 26 A decrease in ocean POC export is not necessarily associated with an increase in atmospheric pCO2. Please see Sigman et al (2010) The polar ocean and glacial cycles in atmospheric CO2 concentration. Nature. for an explanation. Briefly: it is not total POC export, but the efficiency of carbon fixation relative to outgassing that matters. Agreed, we will clarify in the revision that the assumption here is
- 30 that the impact of our decrease in POC export is not offset by a decrease in the rate at which remineralised carbon is returned to the surface. We will, however, re-discuss potential caveat of no increase in remineralisation depth with decreasing ocean temperature.

In revising the conclusions section, and making it more concise, we removed this section.

35

• Page 39, Line 13 - Why are you now talking about terrestrial carbon here? Terrestrial carbon gets mentioned here as we are summarizing our results. However, the text will become easier to follow as we re-write the conclusions section to summarise our research objective, novelties of our approach and then the key conclusions. • Page 40, Lines 1-11 - Please see Menviel et al (2017) Poorly ventilated deep ocean at the Last Glacial Maximum inferred from carbon isotopes: A data-model comparison study. Paleoceanography.

5 Here we attempt to broadly reconcile our positive  $\Delta TerrC$  with the mean ocean  $\delta^{13}C$  change and do not indeed discuss the spatial distribution of  $\delta^{13}C$ , which is another useful constraint, as highlighted by the study of Menviel et al., 2017 (and which interestingly suggests weaker, not stronger AABW transport). We will acknowledge this as an added source of uncertainty in the revision.

10 We have added the following line "A further test would be to compare the simulated spatial distribution of  $\delta^{13}$ C with observations (e.g. Menviel et al., 2017)."

• Page 40, Lines 13-14 - But atmospheric \_13C at the LGM and preindustrial climates were similar at -6.46 ‰ and -6.36 ‰, respectively.

- 15 Agreed. We discuss the role that a glacial increase in terrestrial carbon inventory may have played in the glacial-interglacial  $\delta^{13}$ C record but do not attempt to definitely close the  $\delta^{13}$ C cycle as a detailed evaluation against observations was not the focus of the paper. In the revision, we will make the similarity between preindustrial and LGM atmospheric  $\delta^{13}$ C levels more explicit when discussing the deglacial record.
- 20

We have changed the sentence "As discussed in Zeng (2003) the glacial increase in terrestrial carbon inventory may also potentially explain the increase in atmospheric  $\delta^{13}$ C over the glaciation, as well as the decrease of about 0.3 ‰ at the beginning of the deglaciation (Smith et al., 1999)." to "As discussed in Zeng (2003) the glacial increase in terrestrial carbon inventory may also potentially explain transient

25 trends in the glacial-interglacal atmospheric  $\delta^{13}$ C record, such as the increase in atmospheric  $\delta^{13}$ C over the glaciation, and the decrease of about 0.3 ‰ at the beginning of the deglaciation (Smith et al., 1999)"

• Page 40, Line 28 - And what do these new records show?

- 30 As above, our aim was not to go into the  $\Delta^{14}$ C records in detail but simply acknowledge that there have so far not been attempts to reconcile glacial increases in terrestrial carbon with higher resolution atmospheric CO2 and  $\Delta^{14}$ C deglacial records. These include a significant decline in atmospheric  $\Delta^{14}$ C around 14.6 kyr BP, which Köhler et al., 2014 attribute to permafrost thawing in high northern latitudes, as well as possibly flooding of the Siberian continental shelf. Marcott et al., 2014 in turn show that a
- 35 significant fraction of the deglacial  $CO_2$  rise direct radiative forcing occurred in steps of 10-15 ppm, over less than two centuries, and was followed by no notable change in atmospheric  $CO_2$  for ~1000-1500 years.
  - Page 40, Line 30 The deglacial decrease in atmospheric \_14C could also be

caused by the exchange of highly negative ocean \_14C with the atmosphere. The argument here is not clear.

Our argument here, which we will clarify in the revision is that if 14C-depleted carbon was released from the land to the atmosphere during deglaciation, and subsequently absorbed by the ocean, one might expect to see this signal in ocean  $\Delta^{14}$ C records. Instead, for the deep ocean at least, we see an increase in  $\Delta^{14}$ C and the size of the perturbation has been argued to support an overall positive  $\Delta$ OceanC. Hence, there are potential caveats with the enhanced LGM terrestrial carbon hypothesis. However, we also discuss limitations of  $\Delta^{14}$ C data: p.41, I5-12.

10

We have added "a terrestrial biosphere-induced" in front of "early deglacial decrease in atmospheric  $\Delta^{14}$ C would have also led to a decrease in ocean  $\Delta^{14}$ C and this is yet to be reconciled with ocean  $\Delta^{14}$ C data".

**15 3 Technical corrections**

We will revise the manuscript to include the technical corrections below. 1. Page 7, Line 9 - "did not evidence numerical instability" -> "did not show evidence of numerical instability"

```
20
```

2. Page 7, Line 15 - "has" -> "had". Please use past tense in results sections.

3. Page 9, Line 7 - "impact of error" -> "error"

25 4. Page 9, Line 7 - "impact of certain" -> "certain"

5. Page 9, Line 7 - ", such as changing" -> ", such as changing" (two spaces)

6. Page 15, Line 4 - "thereis" -> "there is"

**30**

7. Page 15, Line 15 - "may have for instance" -> "may have"

8. Page 20, Line 21 - "for instance is that" -> "is that"

35 9. Page 23, Line 3 - "Most of the ensemble members" -> "Most members"

10. Page 26, first line of table - there is a tab separating -1160 from 530.

11. Page 36, Line 11 - double space -> single space

```
12. Page 38, Line 19 - "Terr" -> "TerrC"
```

```
13. Page 38, Line 27 - remove /
```

```
5
```

The above corrections have been incorporated in the revised manuscript.

**Response to referee #3**

10 We thank referee #3 for the constructive comments and suggestions. As above, our replies are in black, and original comments in blue.

This article presents an ensemble of Last Glacial Maximum (LGM) simulations using the GENIE intermediate complexity model with varying parameter values. The model

- 15 simulates the carbon cycle allowing the authors to compare the CO2 values obtained from the model with the 90 ppm decrease from ice core data and analyses carbon stocks, in addition to the study of climate variables. They select two subsets of simulations with increasing constraints and analyses the changes obtained in these simulations in terms of temperature, precipitation, sea ice, ocean circulation and carbon
- 20 stocks with respect to the pre-industrial. The CO2 drawdown in most simulations is due to an increase of carbon storage on land and in the lithosphere while the ocean gets depleted in carbon.

Using an ensemble of simulations to study the change of climate and of the carbon cycle

- 25 during the LGM is a great idea and GENIE is an adequate model as it is fast enough for the long simulations required. However, concerning the carbon cycle part, the lack of carbon isotopes in the simulations prevents any real conclusion to be drawn on the plausibility of the results obtained and the likelihood of the associated mechanisms and carbon stocks changes. As the carbon isotopes are already incorporated within
- 30 GENIE, providing that this is feasible, I suggest to rerun the simulations and redo the analyses with the isotopes (at least carbon 13) in comparison to data, which is crucial to properly evaluate the results.

As suggested in replies to previous comments, we will clarify our research aim, which was not to explain definitively the causes of  $CO_2$  but rather take an uncertainty-based approach to exploring the physical and biogeochemical changes which may have accompanied the LGM  $CO_2$  decrease. Given our focus, we seek to compare our simulation results with observations only more broadly.

**General comments**

1. As I said before, the main point is the absence of carbon isotopes which precludes any strong conclusion to be drawn, it would be best to redo the simulations with the carbon isotopes (at least C13, if possible C14) and compare with ocean and atmosphere data to evaluate which simulations are really plausible. How long would this take?

5

We address this comment in the general response but also note here that spinning up the  $\delta^{13}$ C of the ocean would take > 100,000 years, and that the model ensemble does not simulate  $\Delta^{14}$ C and  $\delta^{13}$ C in the terrestrial biosphere.

- 10 2. From the simulations done so far, we can't know which processes are responsible for the carbon changes, it would be great to have a few additional sensitivity tests (for example taking one set of parameters from the PGACF ensemble) to evaluate the impact of each process on CO2.
- 15 We agree that these experiments would be interesting but are not essential to our study, which focuses on the ensemble as a whole rather than the response of individual ensemble members. We also note again here that while this paper describes the relationships between ensemble outputs, the second (related) paper to be submitted describes dependencies on ensemble parameters to isolate mechanisms.
- 20

35

3. In the figures with maps, it would be good to draw the coastlines of continents to make it easier to see where the changes take place. Agreed, we will add these in the revised manuscript.

25 4. Â'n conversely Â'z appears 28 times in the manuscript, it could probably be removed or replaced a few times. There seems to be a typo here which precludes understanding?

5. On the bar charts, all ensembles (grey, yellow and orange) could be drawn on the same plot to avoid having two subplots, which would help the comparison between the

30 ensembles and reduce the space taken by figures. Agreed, we will combine all 3 ensembles in the revision.

**6**. The hypothesis that carbon stays below the ice sheets is a strong one, it could be interesting to evaluate its impact by doing one (or a few) simulations from the PGACF ensemble without carbon kept under ice sheets.

We agree that it would be interesting to test this directly if there were resources for additional simulations. However, we do show how much carbon is stored below the ice sheets in PGACF-16 and from there one can at least try to estimate what the impact of releasing it would have been on CO2.

Indeed, the LGM burial carbon inventory varies between ~300 and 1300 PgC and if we assume that 90% of the initial atmospheric  $CO_2$  perturbation removed by the oceans and sediments, atmospheric  $CO_2$  would increase by between ~14 and 61 ppmv.

5 However, it is important to note here (as also emphasized in an earlier reply) that the importance of assumptions regarding what happens to carbon in ice sheet areas depends on how much carbon is there. Our LGM burial carbon estimates include the initial preindustrial carbon inventories, plus carbon accumulated in response to glacial forcings.

**10 Specific comments**

p.1-2: The abstract is quite long, and could be better organized, with the problematic explained at the beginning before stating what is the main scientific question raised in this article, how it is raised, and then the main results.

15

Agreed, the abstract will be reorganized in manner suggested above and reduced to just one paragraph.

**The changes to the abstract described above have been implemented.**

20 p. 2-3: Permafrost is not mentioned in the introduction; it would be good to include it.

Yes, thank you, permafrost growth should here be mentioned as a separate mechanism. The text will be revised accordingly.

25 We have revised the second paragraph of the introduction to include permafrost growth.

It would also be interesting to introduce here which data will be later used to constrain the results.

30 Agreed. We will include an overview of upcoming sections at the very end of the introduction section, and in this overview describe which model variables will be compared against observations.

We include which variables will be compared against observations/paleo-proxies towards the end of the introduction.

35

p. 6: During the second stage of the simulations, how does CO2 evolve, does it stay stable? We plotted the evolution of atmospheric  $CO_2$  from stage 1 through 4 in PGACF-16:  $CO_2$  stays stable, except for maybe 3 runs where  $CO_2$  changes by ~< 10 ppmv but then reaches an equilibrium again.

p. 7 l. 12: In the EFPC ensemble, are the simulations at equilibrium at the end of stage 3? What is meant here is probably the EFPC2 ensemble. We again have plots of atmospheric  $CO_2$  and surface sediment %wt CaCO3 for a subset of this ensemble (PGACF-16). Both metrics either in or

5 nearing equilibrium by 10 kyr.

p.7 l.21: The SST value is too high compared to data, how does that compare to other models? Is it in the range?

- 10 We deem the mean value (18.9 °C) to be comparable to other previous model-based estimates, in that e.g. Kim et al., 2003 predict modern SST of ~18 °C, Zhang et al., 2012 predict preindustrial SST of 17.1 °C. The range is 16.4 to 21.9 °C, which is potentially larger than the range of previous model-based estimates.
- 15 p.7 l.22 The sea ice value is given for the Northern and Southern Hemispheres, how is the comparison with data when split between the North and the South?

Our model is set up to output time series of annual average global sea ice area, 31/12 NH and 31/12 SH sea ice areas. The latter is one of the modern plausibility metrics used in Holden et al., 2013a and as

- 20 shown in Table 2, our estimates are comparable to those of Holden et al., 2013a. For 31/12 NH sea ice area, the mean of the EFPC2 ensemble is  $15.1 \pm 1.4$  million km2, and the range is 12.6 to 19.2 million  $\mathrm{km}^2$ . The mean is within the range of typical late winter Arctic sea ice cover today (14-16 million  $\mathrm{km}^2$ ) (NSIDC). Note that the preindustrial NH sea ice extent during the month of maximum (winter) extent simulated by the 13 PMIP2 and PMIP3 models shown in Fig. 4 of Goosse et
- al., 2013 ranges between ~ 13 and 27 million  $km^2$ .

p. 7: The vegetation and soil carbon values are given in table 2 but are not discussed. How does it compare to data ? Is the vegetation distribution ok? Given that it plays an important role in the change of CO2 for the LGM it would be good to know if the

30 preindustrial terrestrial biosphere is well represented or if it has important biases. There is also no discussion of the overturning values given, how does it compare to other models?

We kept discussion of the preindustrial results to a minimum to cut down on the amount of text. It is 35 true that we do not compare any of the values in table 2 (including overturning values) against observations - this is not intentional and we will add a note that there are no major differences between our results (EFPC2) and those of Holden et al., 2013a (EFPC), which meet previously chosen modern plausibility criteria (i.e. modelled values within acceptable distance of observations). We did plot the spatial distribution of vegetation and soil carbon in EFPC2 but did not feel that the

discrepancies were large enough to significantly bias our LGM results and therefore did not include the plots in the manuscript. There is however, one potential exception and we mention this on p. 28: "However, it is also noteworthy that, although not shown here, the regions with the largest decreases in terrestrial carbon density, namely northwest North America, Beringia and the Tibetan plateau area, are

- 5 also the regions with the largest terrestrial carbon densities in the preindustrial simulations". This refers to the terrestrial carbon densities in the EFPC2 ensemble mean and we will clarify in the revision that in the preindustrial EFPC2 ensemble mean: (i) the Tibetan soil carbon peak is overestimated and (ii) the North American soil carbon peak misplaced (compared to observations). We attribute (i) to the lack of soil weathering in the model and the inclusion of land use effects in the observational data-based
- 10 estimate (Holden et al., 2013b; Williamson et al., 2006). We attribute (ii) to the lack of explicit representation of permafrost (instead the model only attempts to capture the soil respiration rates characteristic of permafrost by utilising a distinct soil respiration temperature sensitivity for land temperatures below freezing) (Williamson et al., 2006) and the absence of moisture control on soil respiration.
- 15

We add the above clarification to our description of terrestrial carbon changes, and also include the following: "Comparison of the preindustrial response of  $\text{ENS}_{315}$  (i.e. the original, non- $\Delta CO_2$  filtered ensemble) against the preindustrial ensemble response of Holden et al., 2013a confirms that the two are very similar."

20

p. 9 l. 28-29 I'm not sure I understand or agree with this sentence as the simulations are for the LGM and not the other glacial maxima in terms of orbital parameters.
Agreed, conclusions we draw for the LGM may not be generalizable to other glacial maxima. We will rephrase this in the revision.

25

p. 10 figure 1: maybe replace PRE by PI and explain it somewhere: Pre-industrial (PI). We consistently use PRE to denote preindustrial but can change this to PI for improved understanding.

**We kept the PRE denotation for consistency in the revised manuscript.**

30

35

p. 11 l.10 and following: Could you use temperature and salinity data to select ensemble members that are supported by data?

Although a useful suggestion, it goes against our approach of looking at the ensemble more widely and not putting too much emphasis on individual ensemble members/strongly constraining these. This will be clarified in the revised introduction, which will help understand the current presentation of results.

The introduction has been rewritten to reflect the above.

p. 15 line 10: how does sea ice distribution compare with data?

As mentioned above, our model is set up to output time series of 31/12 NH and 31/12 SH sea ice areas, and we also have maps of the annual average spatial distribution of sea ice. There are no obvious observation-based estimates to compare the latter against, or 31/12 NH sea ice area. For 31/12 SH sea

- 5 ice area, our estimates can be compared against the estimates of Gersonde et al., 2005 and Roche et al., 2012. In the first study, LGM summer sea ice extent is estimated to have increased by between 1-2 million  $km^2$ , which is much smaller than our PGACF ensemble mean of  $11.4 \pm 6$  million  $km^2$ , and falls outside of our 3 to 32.6 million  $km^2$  PGACF ensemble range. However, as noted in Gersonde et al., 2005, major uncertainties concern the reconstruction of summer sea ice extent. Roche et al., 2012
- 10 predict increases in LGM summer sea ice extent between ~2 and 12 million  $km^2$ . We also note here that based on Fig. 4 in Goosse et al., 2013, the LGM change in SH sea ice extent during the month of minimum (summer) extent predicted by the 13 PMIP2 and PMIP3 models ranges between ~ -3 and 25 million  $km^2$ .
- 15 To estimate how our simulated 31/12 NH sea ice compares with data, we will, in the revision, compare our LGM change in the annual average spatial distribution of sea ice with reconstructed changes in winter and summer sea ice extents in the NH. We note here, however, that the mean LGM change in NH 31/12 sea ice area in the PGCAF ensemble is 7.3  $\pm$  2.1 million km2, and the range is 3.6 million to 13.2 million km2. For comparison again, based on Fig. 4 in Goosse et al., 2013, the range of estimates
- 20 predicted by the 13 PMIP2 and PMIP3 models included therein goes from -7 to 4 million  ${
  m km}^2$ .

We also note here an error in the current manuscript: 113-16 p.12. Contrary to our statement, it is unlikely (or at least not more likely than not) that the LGM increase in annual average SH sea ice is underestimated given that the LGM increase in 31/12 SH sea ice lies at the upper end of observed actimates. Winter SH sea ice also here maps 21/12 SH sea ice not auttral winter sea ice Mo will revise

25 estimates. Winter SH sea ice also here means 31/12 SH sea ice not austral winter sea ice. We will revise the paragraph.

In revising the paragraph to reduce the level of detail in the description of LGM SAT changes (and comparison against observations), we deemed this section to no longer be necessary and removed it.

**30**

p. 20 l. 3 Is it really "than in the PGACF-16"? Is this not the ensemble that you are talking about? Thank you, there is a typo, it should indeed say "PGACF".

**We have corrected this in the revision.**

**35**

p. 20 line 10: NADW instead of AABW? We use the brackets here to mean that expanding AABW cell may restrict the AMOC cell (i.e. the upper cell of the AMOC) to lower depths and expanding AMOC cell may restrict AABW to higher latitudes. We will clarify this in the revised manuscript.

**In the revision, we only consider changes in AABW cell strength leading to changes in the AMOC.**

Figure 9: It looks like the NADW is stronger for the LGM than the Pre-industrial , while from the text and figure 8 I understood the opposite: : :

5 We will put the colour bars on the same scale to avoid confusion in the revised manuscript.

Figure 10: could you add the PGACF-16 ensemble? Yes, good suggestion.

p. 23 and following: could you show a map of where the carbon is stored on land?

10 We show the spatial distribution of vegetation, soil and total land carbon changes on p.30. As per suggestion of referee #1, we will show the spatial distributions first, then the globally-integrated numbers, in the revised manuscript.

p. 37: the conclusion is long and more descriptive than conclusive, it might be good to

15 re-organize it.

Agreed. We will shorten the current conclusions section, succinctly summarizing the objective of our research and research strengths, and then describe the key conclusions.

The revised conclusion incorporates the above changes. 20

Correspondence to: Krista M.S. Kemppinen (Krista.Kemppinen@asu.edu)

10

Abstract. During the Last Glacial Maximum (LGM), atmospheric CO2 was around 90 ppmv lower than during the

- preindustrial period. The reasons for this decrease are most often elucidated through factorial experiments testing the impact
  of individual mechanisms. Due to uncertainty in our understanding of the real system, however, the different models used to
  conduct the experiments inevitably take on different parameter values, and different structures. In this paper, the objective
  therefore, is to take an uncertainty-based approach to investigating the LGM\_CO2 drop by simulating it with a large ensemble
  of parameter sets, designed to allow for a wide range of large-scale feedback response strengths. Our aim is not to definitely
   explain the causes of the CO2 drop but rather explore the range of possible responses. Despite years of research, however, the
  exact mechanisms leading to the glacial atmospheric CO2 drop are still not entirely understood. Here, a large (471 member)
- ensemble of GENIE 1 simulations is used to simulate the equilibrium LGM minus preindustrial atmospheric CO2 concentration difference (ΔCO2). The ensemble has previously been weakly constrained with modern observations and was designed to allow for a wide range of large scale feedback response strengths. Out of the 471 simulations, 315 complete without evidence of numerical instability and with a ΔCO2 that centres around -20 ppmy. Roughly a quarter of the 315 runs
- 25 without evidence of numerical instability, and with a ΔCO2 that centres around -20 ppmv. Roughly a quarter of the 315 runs predict a more significant atmospheric CO2 drop, between ~30 and 90 ppmv. This range captures the error in the model's process representations and the impact of processes which may be important for ΔCO2 but are not included in the model. These runs jointly constitute what we refer to as the "plausible glacial atmospheric CO2 change filtered (PGACF) ensemble".
- 30 We find that the LGM CO2 decrease tends to predominantly be associated with Our analyses suggest that decreasing LGM atmospheric CO2 tends to 
[revised manuscript text omitted]

AMD
APM
OL0
OL1 | Atmospheric heat diffusivity ( $m^2 s^{-1}$ )
Atmospheric moisture diffusivity ( $m^2 s^{-1}$ )
Atlantic-Pacific moisture flux scaling
Clear skies OLR reduction (W m -2 )
OLR feedback (W m -2 K -1 ) | 1118875 to 4368143
50719 to 2852835
0.1 to 2.0
2.6 to 10.0
-0.5 to 0.5 | a
a
a
b |

[revised manuscript text omitted]

**3 Results and discussion**

**5 3.1 Preindustrial simulations**

**3 Preindustrial simulations**

**3.1 Ensemble filtering**

After application to stages 2 and 3, the 471-member ("EFPC") ensemble was filtered to 315 simulations, being those with a stage 2 atmospheric CO2 concentration in the range 268 to 288 ppmv (c.f. Prentice et al., 2001), which did not enter a

10 snowball Earth state in stage 3 and which did not evidence numerical instability (c.f. Holden et al., 2013b). These 315 simulations comprise the "EFPC2" ensemble, and form the basis of all the results reported in this study, unless otherwise specified. Individual ensemble members are referred to by their "run IDs", corresponding to the 1000 emulator filtered parameter sets that were originally applied to GENIE-1 (Holden et al., 2013a).

**15 3.2 Modern plausibility and other global metrics**

Comparison of the preindustrial response of  $ENS_{315}$  (i.e. the original, non- $\Delta CO_2$  filtered ensemble) against the preindustrial ensemble response of Holden et al., 2013a confirms that the two are very similar. We additionally evaluate  $ENS_{315}$  against a few additional preindustrial metrics (see S2) and find responses that can be deemed not uncontroversially implausible, following the design principles for the ensemble, outlined in Holden et al., 2013a.

20

EFPC2 preindustrial atmospheric CO2 has a mean concentration of 278.1 ± 1.3 ppmv (standard deviation). The ensemble mean and range for the eight modern climate plausibility metrics are shown in Table 2, and compared against the results of the 471-member EFPC ensemble (Holden et al., 2013a). Metrics are also reported separately for the annual average global ocean carbon inventory, sea surface temperature and sea ice area, and compared with observations (Table 3). The ensemble mean ocean carbon inventory is close to the 36,000 PgC equilibrium preindustrial ocean carbon inventory predicted by GENIE-1 in Lenton et al. (2006), below reconstructed estimates of ca. 38,000 PgC (Houghton et al., 1990), largely attributable to an underestimated ocean volume at our low resolution (Lenton et al 2006). The ensemble mean SST exceeds observations but the error is still comparable to that associated with previous model predictions (e.g. Kim et al., 2003). The ensemble mean sea ice area (SIA) lies within the range of observed estimates.

30

5

Table 2. The EFPC and EFPC2 ensembles modern climate plausibility metrics. The EFPC2 ensemble values
are as reported in Holden et al. (2013a), Table 2. All values are annual averages, except for the Antarctic sea ice
area, which is the end-of-year value. The ensemble means are presented as the mean plus minus one standard
d10deviation.

|                                                         | <del>EFPC2</del>
<del>Ensemble</del>
<del>mean</del> | <del>EFPC2</del>
<del>Ensemble</del>
range | EFPC
Ensemble
mean     | EFPC
Ensemble
range |
|---------------------------------------------------------|------------------------------------------------------------|--------------------------------------------------|------------------------------|---------------------------|
|                                                         |                                                            |                                                  |                              |                           |
| Global SAT (°C)                                         | $\frac{13.7 \pm 1.1}{2}$                                   | <del>11.8 to 16.2</del>                          | $\frac{13.6 \pm 1.1}{1}$     | <del>11.7 to 16.2</del>   |
| Atlantic overturning stream function maximum (Sv)       | <del>16.4 ± 3.8</del>                               | 4.5 to 27.3                                      | <del>17.5 ± 3.2</del>        | <del>10.0 to 25.8</del>   |
| Atlantic overturning stream function minimum (Sv)       | -4.1 ± 1.0                                                 | - <del>6.8 to -0.8</del>                         | -4.1 ± 1.0            | -6.8 to -1.0              |
| 31/12 Antarctic sea ice area (million km 2 ) | <del>6.7 ± 2.7</del>                                       | <del>1.2 to 13</del>                             | <del>6.8 + 2.8</del>  | 1.2 to 12.9               |
| Global VegC (PgC)                                       | 4 <del>99.9 ± 94.5</del>                                   | <del>328.6 to 765.5</del>                        | 4 <del>92 ± 9</del> 4        | <del>326 to 762</del>     |
| Global SoilC (PgC)                                      | <del>1329.7 ± 279.2</del>                                  | <del>896.1 to 2353.2</del>                       | <del>1351 ± 308</del>        | <del>896 to 2430</del>    |
| <del>wt%-CaCO3</del>                         | <del>34.4 ± 7.9</del>                                      | <del>19.1 to 51.5</del>                          | <del>34.1 + 7.8</del> | 20.0 to 50.0              |
| Global occan O 2 (µm kg -1 )      | <del>164.1 ± 19.3</del>                                    | 121.8 to 217.5                                   | <del>165 ± 20</del>          | <del>117 to 216</del>     |

Table 3. Preindustrial ocean carbon inventory, sea surface temperature and sea ice area. All values are annual averages.

|                                     | Ensemble mean              | Ensemble range                | Observations           |
|-------------------------------------|----------------------------|-------------------------------|------------------------|
| Global ocean carbon inventory (PgC) | <del>36056.2 ± 252.4</del> | <del>35280.5 to 36655.7</del> | <del>38000</del>       |
|                                     |                            |                               | Houghton et al. (1990) |
| Global sea surface temperature (°C) | <del>18.9 ± 1.2</del>      | <del>16.4 to 21.9</del>       | <del>15.9</del>        |
|                                     |                            |                               | NCDC, 2015             |

<del>16.3 to 38.6</del>

 $23 \pm 4.2$

<del>19 to 27</del> <del>Lemke et al. (2007)</del>

5

**3.24 LGM ensemble simulationssimulations**

**4.1 Atmospheric carbon dioxide**

- 10 The ensemble ΔCO2 distribution is centered around -20 ppmv, with a range of -88 to 74 ppmv, although the ensemble member corresponding to the latter ΔCO2 is an outlier (Fig. 1). A more negative ΔCO2, between --90 and -30 ppmv is achieved by 104 ensemble members. This range roughly accommodates the impact of error in the model's process representations and the impact of certain potentially important ΔCO2 mechanisms that are missing from the model, such as changing marine bacterial metabolic rate, wind speed (via its effect on gas transfer) and Si fertilization (Kohfeld and Ridgwell, 2009). This is not a
- 15 comprehensive assessment, however, as our model also does not include processes such as the effect of changing winds on ocean circulation (Toggweiler et al., 2006), Si leakage (Matsumoto et al., 2002, 2013, 2014), the effect of decreasing SSTs on CaCO2 production (Iglesias Rodriguez et al., 2002), or changing oceanic PO4 inventory (Menviel et al., 2012). The fact that it is difficult for our model to achieve a ΔCO2 of ~ 90 ppmv without the missing processes is a significant result because of our extensive exploration of model uncertainty when building the model ensemble. Our assumption moving forward is that the
- 20 role of processes which are included in the model add linearly to the missing ones and that our ~ -30 to -90 ppmv ACOx therefore represents a "plausible" glacial atmospheric COx change. Rather than trying to identify the "best" parameter set in this "plausible glacial atmospheric COx change filtered (PGACF) ensemble" (Table 4), we investigate what the emergent model output relationships are and what magnitude and direction responses can be observed. This is to avoid focusing on a candidate parameter with potentially the wrong balance of processes. We also compare the behaviour of the PGACF ensemble
- 25 (with a mean ΔCO2 of -45±13 ppmv) with that of the EFPC2 ensemble to better understand the emergent relationships. Moreoever, we identify which members in the PGACF ensemble predict ΔCO2 from the highly negative end of the ensemble range to help diagnose any significant differences in behaviour along the plausible ΔCO2 spectrum. The lower ΔCO2 limit for this subset of the PGACF ensemble is ~ 60 ppmv, roughly equivalent to allowing for an extra atmospheric CO2 decrease due to changing marine bacterial metabolic rate, wind speed (via its effect on gas transfer) and Si fertilization between the best and
- 30 upper estimate of Kohfeld and Ridgwell (2009). The ensemble members in the subset are referred to collectively as the "PGACF-16" ensemble (Table 4), with the numeral denoting the number of ensemble members.

We also note here that we have apparently succeeded in reproducing one of the most remarkable properties of the 800 kyear temporal record of  $CO_2$ , namely the nearly constant lower bound across different glacial states. There is a significant caveat, however, which is that 47 of the 471 LGM states subsequently diverge to an unrealistic snowball glaciation response to LGM forcing through feedback mechanisms, probably involving low  $CO_2$ , that have to be inferred to be unrealistically strong in this context. These simulations have to be rejected, thus it is difficult to draw any firm conclusions regarding the dynamical controls that maintain the constancy of the lower  $CO_2$  bound in the real system.

---

## Referee Report (RR1)

**Second review of " Coupled climate-carbon cycle simulation of the Last Glacial Maximum atmospheric $CO_2$ decrease using a large ensemble of modern plausible parameter sets"**

The changes made by the authors result in a much better article, but there are a few changes that should have been made and that seem to have been forgotten, in particular concerning figures (changing the scale, merging all three ensembles on the bar plots). Also the ensemble names have been changed but the old names have been used in a few places. In addition, I have a few technical comments. Once these changes have been made this article is suitable for publication.

Technical comments

p4. l.24 "The ensemble": could you be more precise? For example, it could be replaced by "the simulation ensemble"?

p.5 l.3 A point is missing after "(Holden et al., 2010b)"

p.5 l. 5 Replace "global average carbonate preindustrial weathering rates" by "global average preindustrial carbonate weathering rates"

p.7 l.10 replace ".." by "."

p.7 l.18 Define what you call "snowball Earth" (in terms of temperature for example, what threshold you used).

p.7 l.25 I suggest to remind what GWS is: "scaled by GWS (land to ocean bicarbonate flux scaling factor)."

p.9 Figure 1 (and following) As said in the previous stage of review, you could plot all three ensembles (grey, red and yellow) on the same plot.

p.10 l.32. Replace "simulate" by "simulated"

p.12 l.9-10 Put the units in the right place just after the numbers given: "level is 2.84 ± 0.62 and the range is 2 to 4" -> "level is 2.84 ± 0.62 % and the range is 2 to 4 %"

p.12 l.11 Replace "in ENS104 than in both ENS315" by "in ENS104 and in both ENS315"

p.13 l.6 replace ".." by "."

p.15 Figure 6 Could you have the same scale for panels a and c?

p.16 l.8 Replace "(Bartlein et al., 2011 in Alder and Hostetler, 2015)" by "(Bartlein et al., 2011; Alder and Hostetler, 2015)"

p.17 l.9 replace ".." by "."

p.20 Figure 9 Could you also have the same scale for panels a and c here?

p.21 l.4 Replace PGACF by ENS104

p.21 l.8 Re-explain here what scenario 1 is when you mention it.

p.26 l.15 no coma after "further"

p.30 l.7 Replace PGACF by ENS104

p.32 l.8 Re-explain what %LOC is

p.34 l.27 replace "2013 in Köhler et al.," by "2013; Köhler et al.,"

---

## Referee Report (RR2)

Pearse James Buchanan

February 2, 2019

**Institute for Marine and Antarctic Studies, University of Tasmania, Australia.**

In their revised manuscript, Kemppinen and coauthors have made a strong attempt to address the many demands of the three reviewers. Overall, I am pleased with the revisions but am still left unsatisfied. I advocate for further revisions before publication.

The introduction is better, although still needs some work. The authors introduce many mechanisms in the Introduction that might affect CO2. I appreciate that the authors are trying to cover a lot of ground in their introduction, but to simply name a mechanism and then not explain to the reader how that mechanism or change works to affect CO2 is confusing and can be misleading to the reader. I realise that it is beyond the scope of this work to present explain how every mechanism in the biosphere/lithosphere can alter CO2, but I think it is equally poor to name all of them and then not give the reader an explanation of why they are important. For instance, the authors list many factors in the 3rd paragraph in just one sentence and fail to provide any conceptual clarity. Possibly, restricting this initial discussion of mechanisms to just a few important ones would be fine, so long as you detail the links to climate. Alternatively, a table/schematic that describes each mechanism you list in the introduction and takes the reader through its links to global CO2 and climate (i.e. its drivers, processes and effect on CO2) would be a helpful addition.

The methods are clearer.

The results section is clearer and more concise.

The abstract is much clearer, more interesting and conveys the major findings. I would advocate for one more sentence, however, that describes why an increase in TerrC and a decrease in OceanC is associated with a pCO2 drawdown. Something to the tune of "preserving, rather than destroying respired C buried under ice sheets and slower remineralisation rates in soils". I

think this will be of interest to the field and improve the popularity (or notoriety?) of the study.

To this point, I am still confused about what the authors mean when they talk about ice-sheet carbon. My confusion stems from an ignorance of what processes are affecting carbon when the ice sheets grow. The authors state that "The increases in terrestrial biosphere carbon are predominantly due to our choice to preserve rather than destroy carbon in ice sheet areas. However, the ensemble soil respiration also tends to decrease significantly more than net photosynthesis resulting in relatively large increases in non-burial carbon". Great, I follow this. Currently, I understand that carbon is preserved in the soils as ice sheets grow over these areas, and that cooler temperatures globally also tend to increase carbon in soils where vegetation is present.

However, then the authors make statements like "It is the combination of our ice sheet carbon stocks increasing rather than decreasing when exposed to LGM climate, and our choice to preserve rather than destroy this carbon", and I am lost. Why does carbon *increase* under ice sheets? It is entirely possible that carbon is *preserved*, but I do not know what process enables carbon in soil under an ice sheet to increase, and the authors do not offer a process that explains this.

They then go on to say "If most of the carbon that was present in ice sheet areas at the end of the preindustrial runs had been lost to climate forcings, it would not matter much [to what? I assume pCO2] whether the remaining stocks had been destroyed or preserved.". I assume from this statement that the authors mean that carbon tends to accumulate in the terrestrial reservoir regardless of changes under ice sheets because soil respiration is reduced more than photosynthesis under cooler climates, and that this is the most important term. This makes sense when viewed through the lens of the metabolic theory of ecology [by the way it would be good to cite some work of the metabolic theory of ecology (Brown, etc.)]. However, this statement then seems to contradict a previous statement: "If the LGM burial carbon inventories were to be removed, DTERRC would be negative in 13 our of 16 simulations, despite the fact that terrestrial carbon also increases outside of the ice sheet areas in 15 out of 16 simulations." This suggests that burial of carbon from under ice sheets (?) is super important for gains in the terrestrial reservoir, while gains in non-ice sheet areas are not so important.

One of two things are causing confusion and make me sceptical of the results. One, the authors have not provided an explanation of the processes and assumptions of how they treat carbon under ice sheets. More explanation of the processes governing vegetation-soil-burial carbon stocks would be very useful. Such an explanation may be all that is necessary to resolve these apparently conflicting statements that make the results difficult to understand. Second, errors in the treatment of carbon under ice sheets were made in the initial runs, causing carbon to *increase* in non-physical ways, and the authors are trying to hide this, which causes confusion. The authors state in a response to reviewer 3, for instance, that "Our LGM burial carbon estimates include the initial preindustrial carbon inventories, plus carbon accumulated in response to glacial forcings". With all previous information, this statement makes me think that carbon is accumulating under ice-sheets, which is odd. Is this due to some undefined process with physical underpinning, or an unrealistic, non-physical process?

I also have some concerns with regards to the discussion of $\delta^{13}$C in the "Other paleo proxies" section. A strong reason for a lower terrestrial C reservoir under glacial conditions is because of the decrease in $\delta^{13}$C recorded in benthic foraminifera recovered from glacial sediments. The decrease in $\delta^{13}$C is thought to originate from the land, as terrestrial organic matter with very negative $\delta^{13}$C signatures found its way into the atmosphere as the terrestrial reservoir diminished.

Carbon in the atmosphere then moved into the ocean, leading to an explanation of atmospheric CO2 drawdown and the lower oceanic $\delta^{13}$C values. If the major result of this study is to hold, then the authors must present a new and plausible explanation of this decrease in the ocean that does not contradict their simulated increase in terrestrial C.

To do this, the authors invoke a reduced marine productivity, low sea surface temperatures, and greater sea ice extent. However, they do not go further to explain why these features of a glacial climate would decrease $\delta^{13}$C in the ocean without requiring the loss of carbon from the land. My reasoning suggests that:

1. Low marine production would actually increase d13C, because less negative d13C would be transferred via organic matter to depth. NOT CONSISTENT.
2. Low SST would make $\delta^{13}$C more positive because fractionation during outgassing of $CO_2$ (which is what you're invoking) would leave more $^{13}$C in the ocean. NOT CONSISTENT.
3. Greater sea ice area would limit production, as you have stated, which would increase $\delta^{13}$C. NOT CONSISTENT.

The authors then mention other studies that could change $\delta^{13}$C over the glaciation and deglaciation (in the subsequent paragraph), and I suppose their aim here is to introduce uncertainty for the causes of the trends in $\delta^{13}$C (and radiocarbon) to challenge the accepted wisdom. However, with the exception of the Lea 1999 study they cite, I am left unconvinced by their argument and therefore sceptical of their result. Once again, the authors seem to list off many studies without talking through in a conceptually clear manner why that response/mechanism could help to explain their results. Moreover, their mention of Hain et al (2011) is misleading, because this study did not discount depleted $\Delta^{14}$C in a glacial ocean, but rather the "extremely" depleted values found in the Pacific by Stott et al (2009) and Marchitto et al (2007). An increase in the carbon reservoir of the ocean was not challenged by Hain et al (2011).

*Major revisions are once again needed. Overall, I am of the opinion that many of my concerns could be allayed if only more effort was put into making the arguments clearer. So, if the authors can (1) improve the introduction by clarifying why a certain process affects CO2 or do not invoke that process (alternatively schematic/table for reader), (2) clarify the processes that allow carbon to increase under ice sheets over their LGM simulations, and (3) improve their discussion of their results against the carbon isotope and paleo proxy data sets, then I advocate publication.*

I **strongly suggest** that the manuscript provides a more thorough description of how the model treats carbon under ice sheet growth. This may allay my concerns totally and I believe would focus and strengthen the later discussion of alternative processes to explain the carbon isotopes.

**1   General comments**

Some other general concerns that I had as I read the article:

- I think that the Results sections might be better if they were split into "Pre-Industrial conditions", "LGM ensemble conditions", and "Carbon cycling through terrestrial, ocean

and lithospheric reservoirs". This would be a nice way to present the main features of the results, which are pretty cut and dry in the first two sections. The real meat of the paper lies in the consequences for carbon cycling, and an interested reader could flick between "LGM ensemble conditions" and "Carbon cycling through terrestrial, ocean and lithospheric reservoirs" sections to see changes in conditions and consequences for carbon, respectively.

- My initial suggestion regarding overlay of the red (ENS-16) over the yellow (ENS-104) bars in the histograms still stands. This would reduce a lot of unnecessary replication in the figures. The authors could use a transparency setting to ensure that the red and yellow are easily seen if they have the same number of experiments (frequency), and possibly use a break in the vertical axis to emphasise the lower frequency ENS-16 experiments. I note that reviewer 3 also suggested this.

- Another suggestion RE figures is that you use a bimodal colour scheme to present changes in your spatial plots. Reds for positive, blues for negatives and centre the range around zero so places with no change are clearly seen. It is misleading to readers assessing your results to present unbalanced colour schemes when discussion change.

- Once again, the writing needs some attention. There are too many adjectives, unnecessarily difficult acronyms, long subordinate clauses, and double negatives in some instances. Please make it easy for the reader. My native language is english, so I mostly understand with a repeat reading of sentences, but many scientists are not.

**2 Specific comments**

**Abstract**

**Introduction**

- Paragraph 3 is one long sentence. Please break it into more sentences to make it easier to read.

- In my opinion, the paragraphs 5 and 6 of the introduction are unnecessary. You cover these points in the methods and abstract. The content could be reiterated in a small paragraph under the main Results heading.

- The final paragraph is unnecessary, but I understand that reviewer 1 thought it was a good idea. Up to you.

**Methods**

- Page 4, line 16 - What is EFPC? you do not define it. I see you mention it only twice. Just spell it out.

- Page 7, line 23 - You mention that the model requires a detrital flux field that is specific to the ocean component. I strongly suggest you explain what this detrital flux does and how it affects your carbon cycling in the ocean between LGM and PI experiments. It might be explained more thoroughly in Ridgwell and Hargreaves (2007), but I would like clarification here.

- Page 8, line 10 - "If one expects..." Please rephrase this sentence. It took me three times over to make sense of it.
- Page 8, line 24 - "In the latter case,..." Please make it more clear what you mean by this sentence. I logic is not clear.

**Preindustrial simulations**

- Page 10, line 7 - "deemed not uncontrovesially implausible" is a double negative. Make it easy for the reader. "deemed plausible."

**LGM ensemble simulations**

- Page 12, lines 9-12 - can you provide changes in mean salinity in units of psu alongside your percentages?
- Page 21, line 4 - You still use PGACF acronym here despite using ENS-315 elsewhere. Also, PGACF is not defined.
- Page 23, line 17 - But why would ice sheet burial of carbon constitute and *increase* in carbon over that of an active boreal forest, for instance? This needs to be clarified.
- Page 23, line 22 - The increase in terrestrial carbon under ice sheets as a result of LGM climate forcings needs to be clarified both here and up front in the Intro/methods.
- Page 24, line 3 - After reading this sentence many times, I think I now understand what you mean. You mean that if all carbon under ice-sheets were to have been destroyed then TerrC would have decreased under LGM scenarios. Could you make this clearer by being simpler?
- Page 25, line 1 - Why would vegetation carbon increase under an ice sheet?
- Page 25, line 11 - "Here, the increase in terrestrial biosphere carbon both inside and outside of the ice sheet areas, are presumed to reflect the decrease in soil respiration rate due to colder SATs exceeding the decrease in photosynthesis rate due to lower CO2, SAT and precipitation, as they are mainly driven by soil carbon increases." Again, how could an increase in soil carbon occur under an ice sheet that should not have any vegetation providing a carbon input?
- Figure 11 - It would be helpful to see the outline for where the ice sheets were placed in the simulations.
- Page 26, line 11 - "The highly negative LGM terrestrial carbon changes..." This is an example of too many adjectives. "The loss of terrestrial carbon under LGM conditions" is easier to read.
- Page 27 - Table 5 is not simple to read. Perhaps you could represent this in a better way? Maybe a figure? It just takes a long time for me to look at it and understand what is going on given the headings and the lists of numbers.
- Page 29, line 6 - Just for your information, the Buchanan 2016 study did more than just make estimates of POC export during a glacial climate, and integrated global climate changes in physical variables including temperature, sea ice, circulation.
- Page 30, line 7 - again PGACF is used.

**Conclusion**

- The first paragraph is basically the same as what is in the introduction. Simplify and reduce.

- Page 36, line 14 - "equally plausible" of what. Once again, I really advocate for the authors to specify what they mean.

- And you cannot say "broad agreement" until there is a better argument for why your results could be compatible with a lower $\delta^{13}$C in the ocean.

**3  Technical corrections**

1. Page 3, line 14 - replace "gets" with "is"

2. Page 5, line 18 - replace "which" with "those that" in both instances.

---

## Author Response (AR2)

**Authors' response**

We thank all 3 referees for their insightful and constructive comments. Both the second and third referee found our discussion of "other paleoproxies" lacking. We have therefore revised this section and summarized here why our results could potentially be reconciled with ocean d13C despite a higher terrestrial carbon inventory (we also suggest here that increased CaCO3 weathering leads to higher rather than lower ocean d13C due to the input of isotopically heavy weathering products, and acknowledge that lower SSTs would result in lower ocean d13C due to enhanced fractionation at the air-sea interface caused by the cooling):

1) Decrease in marine productivity means lower d13C as phytoplankton discriminate against 13C during photosynthesis, giving the marine organic carbon reservoir a low d13C. Zimov et al., 2009 for instance suggested that their glacial terrestrial carbon increase scenario is only possible if there was a decline in the marine reservoir of organic carbon.

2) Increasing sea ice means lower d13C due to reduced gas exchange

3) Winds are held fixed in our model. However, if these had been lower, the gas exchange at the air-sea interface might have been reduced further.

4) We vary land-ocean CaCO3 fluxes but not land to ocean organic matter fluxes. If our ensemble incorporated enhanced weathering of organic matter and/or reduced deposition of organic carbon at continental margins due to lower sea levels, d13C could potentially have decreased significantly. Wallmann, 2014 for instance attributes the observed decrease primarily to this process.

5) Enhanced glacial carbonate ion concentrations may have reduced the d13C in foraminera shells without altering mean ocean d13C (Lea et al., 1999).

Caveat: LGM deoxygenation over large areas of the deep ocean. This is used in the literature as evidence of a larger LGM ocean carbon inventory (due to reduced ventilation). To challenge this argument, one might show that deoxygenation is partly caused by an increased LGM ocean primary productivity, which means more consumption of O2 at depth. Yet, we are suggesting that LGM POC export fluxes may have been lower. A potential counter-argument therefore is that: yes, productivity was lower but if we were to include processes such as increasing remineralisation depth due to decreasing ocean temperature, or increasing ballasting, more O2 would be consumed at depth. Hence, one could potentially still have deep ocean deoxygenation.

Our conclusion with regard to reconciliation with ocean d13C and oxygen is that conceptually our results can only be reconciled with carbon isotope and oxygen data if additional processes not included in our study are brought into play.

We also note that the manuscript previously included a section on D14C as well as discussion of transient changes in carbon isotopes. In the revision, we remove these components as: (1) we believe these are less crucial, and (2) to keep with the aim of our study, which was not to do a comprehensive assessment of model results against observations. Because of (2), we also remove from the "other paleoproxies" section our suggestions for further simulations.

**Responses to Referee 1**

**General comments**

My main concerns are following.

1) In this study, the burial of land carbon by LGM ice sheets was simulated, which previously has not been done in an LGM equilibrium experiment set-up. However, it's not clear how this process is set up in GENIE. The authors attempt to justify the modelled larger LGM land carbon storage and smaller oceanic carbon storage. However, the arguments still seem weak, since many recent studies have suggested the opposite (for instance, Anderson et al., 2019, Global Biogeochemical Cycles).

2) Simulations were carried out with wide ranges of parameters, and some have captured the dramatic LGM CO2 drop. The relationship among different model variables, without explaining why these relations occur. After reading the whole manuscript, there were still so many question marks left in my mind. As readers of a scientific paper, we often like to see a question got answered within one paper. Thus, I have to say I personally didn't find it a pleasant reading experience.

In the response letter, the author argued: "We note that this paper was intended as part one of two related papers. This paper describing the relationships between ensemble outputs, and the second, currently being finalised for submission, describing dependencies on ensemble parameters to isolate mechanisms. ... " I would argue, that one only need to read the introduction and methodology of this paper and the analysis of the next paper, since the relations have to be mentioned when they are analysed. I do understand that it's not feasible to put everything in one huge paper. However, I don't agree that the value of this manuscript should be so much dependent on the next one.

My suggestion is to re-organise content of this manuscript and of the one in preparation such that in the first one, the methodology is described in detail, the behaviour of the physical states of the (sub)ensembles are described and EXPLAINED. In the second manuscript, the results concerning terrestrial and ocean carbon cycle are described and explained.

While we appreciate and agree that there are advantages and disadvantages to any way of dividing the material, we prefer the present configuration because the physical and carbon-cycle processes are very strongly connected, indeed the principal utility of our modelling framework is the ability to focus on these connections. We therefore prefer to retain physical and carbon-cycle processes in both papers, focusing first on describing the model output variables, potential relationships between them and whether or not in agreement with observations.

3) Section 3.2.6: Other paleo proxies

I think it's inappropriate to discuss extensively about carbon isotopes, which are not at all accounted for in this model. I got two main messages from this session. First, the authors attempt to argue that their modelled lower LGM oceanic carbon storage is possible, although this is not supported by many previous studies. However, I find the supporting arguments rather weak. See the specific comments below.
Second, the authors briefly summarise some previous work on 13C and 14C, and suggest some further simulations (see the specific comments). I would like to ask the authors the purpose of this text. Does it help achieving your research aim?

Our research aim is to describe the range of possible CO2 responses given our uncertainty-based approach, and to a lesser extent, how this our response space compares with observations. To that end, we have reduced the section on other paleo proxies and also modified some of our supporting statements.

Specific comments

Page 1 Line 30 - Page 3 Line 5:
At the beginning of the introduction, the authors spend more than one page to review the hypotheses for the LGM CO2 drop compared to preindustrial. When reading this, I start to expect that this manuscript is about deepening the understanding of these mechanisms or the interplay among them. However, from Page 3 line 6, the authors state that their aim is to examine the sensitivity of LGM CO2 drop to model parameters. So why mentioning at all those mechanisms? At this stage, there seems no link between the mentioned mechanisms and the model parameters to be varied in GENIE. As stated in the comments to the first submitted manuscript, I appreciate that the authors give an comprehensive review of the mechanisms that governing the LGM atmospheric CO2 drop. However, it becomes excessive when it does not prepare the readers for the central topic of this study.
My suggestions for improvement is following:

1) Shorten the paragraphs on LGM CO2 drop mechanisms. Provide the links, if exist, between the mechanisms and the model parameters varied in this study.
2)The authors do state that "There is rarely any investigation of the impact of alternative assumptions regarding parameter values ..." I still think some words should be spent on what has been done in the

(limited) previous studies.

We feel that an overview of the mechanisms is useful since, to the extent possible, we attempt to link these to the model output variables we describe. We also add the following sentence in the introduction to further clarify the aim of this manuscript: "Knowledge of these relationships can in turn inform analysis, in the future, of the relationship between the ensemble parameters and model outputs, in order to isolate individual LGM CO2 mechanisms".

With regard to what has been done in previous studies in terms of investigation of the impact of alternative assumptions regarding parameter values, we added the following sentence: "An example of a relevant study is Bouttes et al., 2011, which varied model parameters controlling the importance of iron fertilisation, bring rejection and stratification-dependent diffusion in an ensemble setting, assessing the agreement of the model output with data."

Page 3 Line 15
It has not yet been implemented in GENIE in an equilibrium set-up, or it has not been implemented in other models?
We have specified that it is the first time it is done in models more generally.
Page 4 Line 29-31

Have the OL1 and VPC been explained in previous GENIE papers? If not, please provide more details about what do these parameters do. e.g., what happens if OL1 is increased.
We clarify that these parameters have been explained in Holden et al., 2013b.

Page 7 Line 10-11
What is effect of bioturbation, to increase vertical diffusivity in the sediment pore water? If so, the sediment equilibrium faster with bioturbation.
The effect is to mix sediment material. Bioturbation is turned off to allow surface sediment composition and CaCO3 burial rates to come to an equilibrium faster.
(http://www.seao2.info/cgenie/docs/cGENIE.muffin.Examples.pdf )

Page 8 Line 3-6
Is it the first time that the burial of terrestrial carbon under LGM ice sheet is implemented in GENIE? If yes, please provide here details about how it is done. If not, please provide references.
We include more details in the revised manuscript.

Page 12 Line 9-12
Here salinity rather than sea level is shown and is (mainly) described. So please consider change the title.
We have revised the title.

Page 16 Line 8

There was no "China" 21k before present. "Eastern Asia" instead?
We use Eastern Asia in the revised manuscript.
Page 25 Line 23
I did not find observational data in Fig 11.
Correct. We have revised the text accordingly.
Page 30 Line 16-18
It has been mentioned many times that "We estimate that up to ~60 ppmv of ΔCO2". Please provide information about how this was done.

We clarify in the revision that this is based on our expert opinion.

Page 33 Line 23-24
I find this argument weak. Indeed the LGM marine primary production is lower in the area of sea ice. However, both modelling and proxy data studies (e.g. Muglia et al., 2018, Earth and Planetary Science Letters, and references therein) have suggested that the LGM primary production in the Southern Ocean is higher compared to pi.
We discuss POC export in more detail on pp29-30 but reference Zimov et al., 2009 here as the authors there argued that the decrease in ocean δ^13C could have been caused by a decrease in marine productivity.

Page 34 Line 1-3
I would expect the opposite: LGM ocean delta_13C is lower when CaCO3 weathering flux is higher. Increased CaCO3 weathering flux results in the uptake of CO2 into the ocean. The atmospheric delta_13C is low (< -6 permil). Thus the uptake of atmospheric CO2 will decrease oceanic delta_13C.

Page 35 Line 5-6
There are other methods to estimate LGM ocean carbon inventory, e.g. based oxygen (see Anderson et al., 2019, Global Biogeochemical Cycles).
Agreed, we discuss this in the final paragraph of section 3.2.6.

Page 35 Line 15-30
Instead of blaming missing processes in the model, why not just start analysing your ensemble results to investigate how oceanic oxygen, carbon storage, and so on, respond to different physical states?
We believe our parameter-space exploration is sufficiently comprehensive to be relatively confident that the discrepancy strongly indicates the importance of processes not represented in the model. Indeed, this is an important result of the extensive computational experiment undertaken and described. As we have explained, our preference is to present the evidence for this in terms of the full analysis of the model states compared to observations in this, already long, paper, and reserve the emulation-based factor analysis for a later paper, which will necessarily be another long paper for a proper exposition.
Attempting to add a very short section on the factor analysis to this paper would, in our view, be necessarily incomplete and hence highly unsatisfactory.

Figures

1) When describing model results shown in a figure (or a table), it would be easier for the readers to follow if "what is shown in which figure(table)" is given at the beginning of the text. The authors have all the pictures in mind but the readers don't.

In several places, we have included a statement of the following kind: "As shown in Fig".

2) There are too many figures. Some figures could be combined. For instance, Fig 1 (a) and (b) can be combined into one figure. And in this way one can clearly compared different ensemble sets. In the case of Fig 1, the authors could also consider using nonlinear vertical axis such that the low frequencies are visible.

Where ENS16 and ENS104 were shown separately, these have been combined into one figure. In the case of Fig. 10 we also noticed that ENS16 was not previously shown so we include it in the revised manuscript.

3) Fig 9

Please use the same colorbar range for (a) and (c), such that it's easier to compare between LGM and pi.

We have modified the figure as suggested.

**Responses to Referee 2**
In their revised manuscript, Kemppinen and coauthors have made a strong attempt to address the many demands of the three reviewers. Overall, I am pleased with the revisions but am still left unsatisfied. I advocate for further revisions before publication.

The introduction is better, although still needs some work. The authors introduce many mechanisms in the Introduction that might affect CO2. I appreciate that the authors are trying to cover a lot of ground in their introduction, but to simply name a mechanism and then not explain to the reader how that mechanism or change works to affect CO2 is confusing and can be misleading to the reader. I realise that it is beyond the scope of this work to present explain how every mechanism in the biosphere/lithosphere can alter CO2, but I think it is equally poor to name all of them and then not give the reader an explanation of why they are important. For instance, the authors list many factors in the 3rd paragraph in just one sentence and fail to provide any conceptual clarity. Possibly, restricting this initial discussion of mechanisms to just a few important ones would be fine, so long as you detail the links to climate. Alternatively, a table/schematic that describes each mechanism you list in the introduction and takes the reader through its links to global CO2 and climate (i.e. its drivers, processes and effect on CO2) would be a helpful addition.

We have clarified the linkages between the processes and CO2 and/or the drivers of the processes.

The methods are clearer.

The results section is clearer and more concise.

The abstract is much clearer, more interesting and conveys the major findings. I would advocate for one more sentence, however, that describes why an increase in TerrC and a decrease in OceanC is associated with a pCO2 drawdown. Something to the tune of "preserving, rather than destroying respired C buried under ice sheets and slower remineralisation rates in soils". I think this will be of interest to the field and improve the popularity (or notoriety?) of the study.

We have modified the abstract to reflect this suggestion.

To this point, I am still confused about what the authors mean when they talk about ice-sheet carbon. My confusion stems from an ignorance of what processes are affecting carbon when the ice sheets grow. The authors state that "The increases in terrestrial biosphere carbon are predominantly due to our choice to preserve rather than destroy carbon in ice sheet areas. However, the ensemble soil respiration also tends to decrease significantly more than net photosynthesis resulting in relatively large increases in non-burial carbon". Great, I follow this. Currently, I understand that carbon is preserved in the soils as ice sheets grow over these areas, and that cooler temperatures globally also tend to increase carbon in soils where vegetation is present.

However, then the authors make statements like "It is the combination of our ice sheet carbon stocks increasing rather than decreasing when exposed to LGM climate, and our choice to preserve rather than destroy this carbon", and I am lost. Why does carbon *increase* under ice sheets? It is entirely possible that carbon is *preserved*, but I do not know what process enables carbon in soil under an ice sheet to increase, and the authors do not offer a process that explains this.

They then go on to say "If most of the carbon that was present in ice sheet areas at the end of the preindustrial runs had been lost to climate forcings, it would not matter much [to what? I assume pCO2] whether the remaining stocks had been destroyed or preserved.". I assume from this statement that the authors mean that carbon tends to accumulate in the terrestrial reservoir regardless of changes under ice sheets because soil respiration is reduced more than photosynthesis under cooler climates, and that this is the most important term. This makes sense when viewed through the lens of the metabolic theory of ecology [by the way it would be good to cite some work of the metabolic theory of ecology (Brown, etc.)]. However, this statement then seems to contradict a previous statement: "If the LGM burial carbon inventories were to be removed, DTERRC would be negative in 13 our of 16 simulations, despite the fact that terrestrial carbon also increases outside of the ice sheet areas in 15 out of 16 simulations." This suggests that burial of carbon from under ice sheets (?) is super important for gains in the terrestrial reservoir, while gains in non-ice sheet areas are not so important.

One of two things are causing confusion and make me sceptical of the results. One, the authors have not provided an explanation of the processes and assumptions of how they treat carbon under ice sheets. More explanation of the processes governing vegetation-soil-burial carbon stocks would be very useful. Such an explanation may be all that is necessary to resolve these apparently conflicting statements that make the results difficult to understand. Second, errors in the treatment of carbon under ice sheets were made in the initial runs, causing carbon to *increase* in non-physical ways, and the authors are trying to hide this, which causes confusion. The authors state in a response to reviewer 3, for instance, that "Our LGM burial carbon estimates include the initial preindustrial carbon inventories, plus carbon accumulated in response to glacial forcings". With all previous information, this statement makes me think that carbon is accumulating under ice-sheets, which is odd. Is this due to some undefined process with physical underpinning, or an unrealistic, non-physical process?

We have clarified the ice-sheet/carbon interaction/link throughout the text. In summary, however, carbon does not get modified further once it is covered by ice.

I also have some concerns with regards to the discussion of $\delta^{13}C$ in the "Other paleo proxies" section. A strong reason for a lower terrestrial C reservoir under glacial conditions is because of the decrease in $\delta^{13}C$ recorded in benthic foraminifera recovered from glacial sediments. The decrease in $\delta^{13}C$ is thought to originate from the land, as terrestrial organic matter with very negative $\delta^{13}C$ signatures found its way into the atmosphere as the terrestrial reservoir diminished.

Carbon in the atmosphere then moved into the ocean, leading to an explanation of atmospheric $CO_2$ drawdown and the lower oceanic $\delta^{13}C$ values. If the major result of this study is to hold, then the authors must present a new and plausible explanation of this decrease in the ocean that does not contradict their simulated increase in terrestrial C.

To do this, the authors invoke a reduced marine productivity, low sea surface temperatures, and greater sea ice extent. However, they do not go further to explain why these features of a glacial climate would decrease $\delta^{13}C$ in the ocean without requiring the loss of carbon from the land. My reasoning suggests that:

1. Low marine production would actually increase d13C, because less negative d13C would be transferred via organic matter to depth. NOT CONSISTENT.
2. Low SST would make $\delta^{13}C$ more positive because fractionation during outgassing of $CO_2$ (which is what you're invoking) would leave more $^{13}C$ in the ocean. NOT CONSISTENT.
3. Greater sea ice area would limit production, as you have stated, which would increase $\delta^{13}C$. NOT CONSISTENT.

The authors then mention other studies that could change $\delta^{13}C$ over the glaciation and deglaciation (in the subsequent paragraph), and I suppose their aim here is to introduce uncertainty for the causes of the trends in $\delta^{13}C$ (and radiocarbon) to challenge the accepted wisdom. However, with the exception of the Lea 1999 study they cite, I am left unconvinced by their argument and therefore sceptical of their result. Once again, the authors seem to list off many studies without talking through in a conceptually clear manner why that response/mechanism could help to explain their results. Moreover, their mention of Hain et al (2011) is misleading, because this study did not discount depleted $\Delta^{14}C$ in a glacial ocean, but rather the "extremely" depleted values found in the Pacific by Stott et al (2009) and Marchitto et al (2007). An increase in the carbon reservoir of the ocean was not challenged by Hain et al (2011).

It may be true that Hain et al., 2011 do not negate the presence of an increased ocean carbon reservoir. However, the authors do cast doubt on both the interpretation of the mid-depth D14C anomalies and the existence of an extremely isolated deep reservoir due to constraints of sediment CaCO3 and ocean O2 (and atmospheric CO2).

*Major revisions are once again needed. Overall, I am of the opinion that many of my concerns could be allayed if only more effort was put into making the arguments clearer. So, if the authors can (1) improve the introduction by clarifying why a certain process affects CO2 or do not invoke that process (alternatively schematic/table for reader), (2) clarify the processes that allow carbon to increase under ice sheets over their LGM simulations, and (3) improve their discussion of their results against the carbon isotope and paleo proxy data sets, then I advocate publication.*

I **strongly suggest** that the manuscript provides a more thorough description of how the model treats carbon under ice sheet growth. This may allay my concerns totally and I believe would focus and strengthen the later discussion of alternative processes to explain the carbon isotopes.

**1 General comments**

Some other general concerns that I had as I read the article:

- I think that the Results sections might be better if they were split into "Pre-Industrial conditions", "LGM ensemble conditions", and "Carbon cycling through terrestrial, ocean

and lithospheric reservoirs". This would be a nice way to present the main features of the results, which are pretty cut and dry in the first two sections. The real meat of the paper lies in the consequences for carbon cycling, and an interested reader could flick between "LGM ensemble conditions" and "Carbon cycling through terrestrial, ocean and lithospheric reservoirs" sections to see changes in conditions and consequences for carbon, respectively.

We think this is a good suggestion. However, since we also discuss changes not impacted by the physical conditions (weathering) and changes impacted by changes in the carbon stocks (CaCO3 burial), we believe that the present structure works better. Our description of the model results has, however, been modified to make for an easier read.

- My initial suggestion regarding overlay of the red (ENS-16) over the yellow (ENS-104) bars in the histograms still stands. This would reduce a lot of unnecessary replication in the figures. The authors could use a transparency setting to ensure that the red and yellow are easily seen if they have the same number of experiments (frequency), and possibly use a break in the vertical axis to emphasise the lower frequency ENS-16 experiments. I note that reviewer 3 also suggested this.

We have included ENS16 and ENS104 in the same figure.

- Another suggestion RE figures is that you use a bimodal colour scheme to present changes in your spatial plots. Reds for positive, blues for negatives and centre the range around zero so places with no change are clearly seen. It is misleading to readers assessing your results to present unbalanced colour schemes when discussion change.

We have modified the figures to reflect this suggestion.

- Once again, the writing needs some attention. There are too many adjectives, unnecessarily difficult acronyms, long subordinate clauses, and double negatives in some instances. Please make it easy for the reader. My native language is english, so I mostly understand with a repeat reading of sentences, but many scientists are not.

As suggested above, we have reviewed the writing throughout the text.

**2   Specific comments**

**Abstract**

**Introduction**

- Paragraph 3 is one long sentence. Please break it into more sentences to make it easier to read.

We have revised the third paragraph.

- In my opinion, the paragraphs 5 and 6 of the introduction are unnecessary. You cover these points in the methods and abstract. The content could be reiterated in a small paragraph under the main Results heading.

- The final paragraph is unnecessary, but I understand that reviewer 1 thought it was a good idea. Up to you.

We indeed feel that the final paragraph adds clarity. For this reason also we keep the preceding paragraphs: they include which model variables will be evaluated (reviewer 3 suggestion), and since we review mechanisms at the beginning of the introduction, we want to mention that not all are included in our ensemble, soon after this.

**Methods**

- Page 4, line 16 - What is EFPC? you do not define it. I see you mention it only twice. Just spell it out.

The revision incorporates the suggested edit.

- Page 7, line 23 - You mention that the model requires a detrital flux field that is specific to the ocean component. I strongly suggest you explain what this detrital flux does and how it affects your carbon cycling in the ocean between LGM and PI experiments. It might be explained more thoroughly in Ridgwell and Hargreaves (2007), but I would like clarification here.

We do not explicitly model all sediment material (e.g. opal) and therefore prescribe these components to get the right wt% distribution of CaCO3 in sediments. The detrital flux does not affect the carbon cycling in the ocean between LGM and PI experiments.

- Page 8, line 10 - "If one expects..." Please rephrase this sentence. It took me three times over to make sense of it.
- Page 8, line 24 - "In the latter case,..." Please make it more clear what you mean by this sentence. I logic is not clear.

The revision takes into account the above suggestions.

**Preindustrial simulations**

- Page 10, line 7 - "deemed not uncontrovesially implausible" is a double negative. Make it easy for the reader. "deemed plausible."

The manuscript has been revised as suggested.

**LGM ensemble simulations**

- Page 12, lines 9-12 - can you provide changes in mean salinity in units of psu alongside your percentages?

We present percentages rather than psu units because the variable we describe is the FFX parameter (freshwater scaling flux factor), expressed as a percentage increase in LGM salinity. A value of 1.5 is designed to yield a sea level drop of 120 m and concomitant 3% increase in global salinity. We vary FFX between 1 and 2 in the ensemble.

In reviewing this section, we noticed that the sentence "there is, however, a weak positive correlation between %S and $\Delta CO_2$ in $ENS_{315}$ (r=0.17)" ought to be removed as the correlation cannot be deemed significant.

- Page 21, line 4 - You still use PGACF acronym here despite using ENS-315 elsewhere. Also, PGACF is not defined.

We have replaced both instances of PGCAF remaining with ENS104.

- Page 23, line 22 - The increase in terrestrial carbon under ice sheets as a result of LGM climate forcings needs to be clarified both here and up front in the Intro/methods.

- Page 24, line 3 - After reading this sentence many times, I think I now understand what you mean. You mean that if all carbon under ice-sheets were to have been destroyed then TerrC would have decreased under LGM scenarios. Could you make this clearer by being simpler?

- Page 25, line 1 - Why would vegetation carbon increase under an ice sheet?

- Page 25, line 11 - "Here, the increase in terrestrial biosphere carbon both inside and outside of the ice sheet areas, are presumed to reflect the decrease in soil respiration rate due to colder SATs exceeding the decrease in photosynthesis rate due to lower CO2, SAT and precipitation, as they are mainly driven by soil carbon increases." Again, how could an increase in soil carbon occur under an ice sheet that should not have any vegetation providing a carbon input?

- Figure 11 - It would be helpful to see the outline for where the ice sheets were placed in the simulations.

As suggested above, we have gone through mentions of the ice-sheet/carbon interaction/link in the manuscript and clarified the text throughout.

5 For the ice sheet outline, we could provide as a supplementary material, the following figure:

The Antarctic, Greenland and LGM ice sheets are identified in Fig. S1 by their surface albedo (0.8), as there is no default ice sheet output. The surface albedo field is from a random LGM simulation, at t=2000 years, and with snow albedo = 0.6 to ensure the ice sheets are distinguishable from snow.

[Figure]

**Fig. S1. Surface albedo.** LGM ice sheets are shown in brown (albedo of 0.8).

- Page 26, line 11 - The highly negative LGM terrestrial carbon changes... This is an example of too many adjectives. "The loss of terrestrial carbon under LGM conditions" is easier to read.

We have modified the sentence in the revision.

- Page 27 - Table 5 is not simple to read. Perhaps you could represent this in a better way? Maybe a figure? It just takes a long time for me to look at it and understand what is going on given the headings and the lists of numbers.

- Page 29, line 6 - Just for your information, the Buchanan 2016 study did more than just make estimates of POC export during a glacial climate, and integrated global climate changes in physical variables including temperature, sea ice, circulation.

- Page 30, line 7 - again PGACF is used.

We have changed the description of the table to make it easier to understand. This should now also be aided by the modifications pertaining to the ice sheet/carbon interaction/link.

**Conclusion**

- The first paragraph is basically the same as what is in the introduction. Simplify and reduce.

The first paragraph has been revised in the manner suggested above.

- Page 36, line 14 - "equally plausible" of what. Once again, I really advocate for the authors to specify what they mean.

This has now been clarified in the revision.

- And you cannot say "broad agreement" until there is a better argument for why your results could be compatible with a lower $\delta^{13}C$ in the ocean.

This section has been revised.

**3    Technical corrections**

1. Page 3, line 14 - replace "gets" with "is"

2. Page 5, line 18 - replace "which" with "those that" in both instances.

5    We have incorporated the technical corrections.

**Responses to referee 3**

Second review of " Coupled climate-carbon cycle simulation of the Last Glacial Maximum atmospheric **CO2** decrease using a large ensemble of modern plausible parameter sets"

The changes made by the authors result in a much better article, but there are a few changes that should have been made and that seem to have been forgotten, in particular concerning figures (changing the scale, merging all three ensembles on the bar plots). Also the ensemble names have been changed but the old names have been used in a few places. In addition, I have a few technical comments. Once these changes have been made this article is suitable for publication.

In the revision, we have changed, as appropriate, the figure scales and merged the ensembles. Where old names were forgotten, these have also been replaced, and the technical comments below addressed.

Technical comments

p4. l.24 "The ensemble": could you be more precise? For example, it could be replaced by "the simulation ensemble"?

p.5 l.3 A point is missing after "(Holden et al., 2010b)"

p.5 l. 5 Replace "global average carbonate preindustrial weathering rates" by "global average preindustrial carbonate weathering rates"

p.7 l.10 replace ".." by "."

p.7 l.18 Define what you call "snowball Earth" (in terms of temperature for example, what threshold you used).

p.7 l.25 I suggest to remind what GWS is: "scaled by GWS (land to ocean bicarbonate flux scaling factor)."

p.9 Figure 1 (and following) As said in the previous stage of review, you could plot all three ensembles (grey, red and yellow) on the same plot.

p.10 l.32. Replace "simulate" by "simulated"

p.12 l.9-10 Put the units in the right place just after the numbers given: "level is 2.84 ± 0.62 and the range is 2 to 4" -> "level is 2.84 ± 0.62 % and the range is 2 to 4 %"

p.12 l.11 Replace "in ENS104 than in both ENS315" by "in ENS104 and in both ENS315"

p.13 l.6 replace ".." by "."

p.15 Figure 6 Could you have the same scale for panels a and c?

p.16 l.8 Replace "(Bartlein et al., 2011 in Alder and Hostetler, 2015)" by "(Bartlein et al., 2011; Alder and Hostetler, 2015)"

p.17 l.9 replace ".." by "."

p.20 Figure 9 Could you also have the same scale for panels a and c here?

p.21 l.4 Replace PGACF by ENS104

p.21 l.8 Re-explain here what scenario 1 is when you mention it.

p.26 l.15 no coma after "further"

p.30 l.7 Replace PGACF by ENS104

p.32 l.8 Re-explain what %LOC is

p.34 l.27 replace "2013 in Köhler et al.," by "2013; Köhler et al.,"

[revised manuscript text omitted]

---

## Author Response (AR3)

We thank the editor for her useful comments.

Dear Dr. Kemppinen,

Thank you for submitting the revised version of your manuscript entitled "Coupled climate-carbon cycle simulation of the Last Glacial Maximum atmospheric CO2 decrease using a large ensemble of modern plausible parameter sets".
I have attentively read the responses to the referees' comments as well as your revised version. I would be happy to accept your manuscript for publication after the points listed below have been taken into account:

1) Figures 3 and 7: Please add land contours so that the maps can be read properly. Please note that this was asked upon first submission of the manuscript, and repeated during all stages of review. If needed, please have a look at "Ferret" (https://ferretop.pmel.noaa.gov/Ferret/ ), which would allow you to plot your data and add land contours on your maps easily.
The figures have been modified as suggested. In revising Fig. 3c, we also noticed that the range for the color bar was too low, leading to a portion of the North Atlantic having no colour. This error was introduced when changing the color bar as part of the previous revision, and has now been rectified.

2) The exact same sentence was added at the end of the abstract and conclusion. Please modify them and please make it clearer that oxygen and carbon isotopes are not simulated here.
We added "not evaluated in this study" behind carbon isotope and oxygen data in the abstract. The abstract now reads "An initial comparison of these dominant changes with observations and paleo-proxies other than carbon isotope and oxygen data (not evaluated directly in this study) suggests broad agreement. However, we advise more detailed comparisons in the future, and also note that, conceptually at least, our results can only be reconciled with carbon isotope and oxygen data if additional processes not included in our model are brought into play."

The conclusion now reads: "The dominant changes were broadly in agreement with observations and paleo-proxies other than carbon isotope and oxygen data, which we did not evaluate directly. However, we advise more detailed comparisons in future studies. It is also likely that our results can only be reconciled with carbon isotope and oxygen data if processes currently missing from our model are taken into account"

Other comments (lines and pages refer to the manuscript with track changes on):
- P19, L16-17: reformulate the iron fertilization part as it is not correct as is

We have changed this sentence to make it clearer that iron fertilization = stimulation of iron-limited biological pump
- P19, L 24-25: As far as I know stronger westerly winds over the North Atlantic have not been put forward as a possible mechanism to decrease atmospheric CO2 over G-IG cycles.
It is true that Muglia and Schmittner do not make any direct connection between wind stress and CO2 via the AMOC. We have removed this reference.
- P21, L. 31; Please directly start as "Section 2 describes…."
We have revised this section as suggested.
- Section 2.4: When describing the different subsets, could you please add the names of the subsets in parathensis.
We have revised this section as suggested
- P28, L.31: Please move "however" to the beginning of the sentence
Done
- P38, L.12-14: To me, a difference of 0.1 Sv +/- 1.2 Sv indicate that there is no statistical difference

between the LGM and PRE Atlantic AABW.
Agreed. We have rephrased this section accordingly.
- P39, L. 3: Do you mean the northern limit of AABW? Why AMOC?
Yes, this sentence should be about the AABW, we have removed the reference to AMOC.
- P39, L. 6: please spell out "less than"

Done
- P43, L. 9-10: I assume here you are referring to an increase in terrestrial carbon due to reduced soil decomposition. Please use this term instead of "growth of biosphere land C"
We have modified the sentence accordingly.
- Section 3.2.2. Please read the recent manuscript by Jeltsch-Thommes et al. 2019, https://www.clim-past-discuss.net/cp-2018-167/
This is an interesting study, taking an ensemble of simulations that includes some of the processes (e.g. organic carbon weathering and burial) which we do not include, and looking at which ensemble members satisfy atm 13C + ocean 13C + deep pacific carbonate ion concentration + atmCO2 constraints. Land C is almost invariably negative at the LGM. To acknowledge this important research we added the following sentence to section 3.2.6 (in addition to referencing the paper in Table 4 of section 3.2.2): "Yet, further research is required here as one recent study suggests that taking into account these missing processes would most likely not (the possibility is not completely ruled out) allow reconciliation of a positive ΔTerrC with the observed mean glacial ocean $\delta^{13}$C value (Jeltsch-Thommes et al., 2018)".

- P47, L. 7: Please take out "biosphere"
Done
- Table 4: Please add at the beginning of the legend that the estimates shown are for LGM compared to PRE. Please add estimates of Delta TerrC of Jeltsch-Thommes et al. 2019 and Menviel et al., 2017 (Paleoceanography).

Done. We also ordered the references by date in the legend, and changed "C12" to CI12 and "A15" to A15 for consistency throughout.
- Table 5: I suppose you are referring to carbon stored under ice-sheets

Yes, during the LGM we mean carbon stored under the Eurasian and Laurentide ice sheets. During preindustrial – it is the carbon in the corresponding grid cells. We have modified the legend of Table 5.

- P53, L. 11: Please rephrase this sentence as its meaning is unclear.

The sentence now reads: "As shown in Fig. 12, compared to ENS104, the POC flux decreases in ENS315 and ENS16 tend to be smaller and larger respectively"

- Figure 14: Please spell out GWS in the legend.

Done. We also noticed that the first sentence in section 3.2.4 read "glacial weathering factor". We changed this to "land-to-ocean bicarbonate flux scaling factor"
- P61, L. 22-23: The impact of changes in marine productivity on the global oceanic d13C is not that straightforward and would depend on the associated changes in remineralisation and burial. Enhanced accumulation of remineralised carbon in the deep ocean would lower d13C.

Yes, agreed. The text now reads: "These include reduced marine productivity (e.g. Zimov et al., 2009), as phytoplankton discriminate against $^{13}$C during photosynthesis, giving the marine organic carbon reservoir a

low $\delta^{13}$C. However, we note that the sign of this impact would additionally depend on the associated changes in organic matter remineralisation and burial".

- Last sentence of the conclusion: please modify as stated above.
Done

**Uploaded Files validated** (11 Apr 2019) by Lorena Grabowski

[revised manuscript text omitted]

---

## Author Response (AR4)

**Response to the Editor**

We have added, in the revised manuscript, a code and data availability section (1) and an author contribution section (2). We also include two additional (3,4) changes.

**1. Code and availability.**

GENIE-1 was checked out via https://source.ggy.bris.ac.uk/wiki/GENIE using Subversion (SVN). The simulations described here are with release version 2-8-0. In addition to the source code, several packages and applications such as the NetCDF libraries are required by GENIE-1 (University of Bristol

10 Geography Source 2014).

The way in which GENIE-1 is run manually is as described in Ridgwell (2012), for the GENIE developmental variant cGENIE: the basic flavour and configuration of GENIE-1 is run from ~/genie/genie-main by issuing the command:

./genie.job

15 genie.job is a shell script which determines the basic ("base") configuration of the model. A different and flavour configuration of the model is obtained by specifying a different base configuration file:

./genie.job -f example.xml

where example.xml is a specified model configuration (/flavour) .xml file (Ridgwell, 2012).

The ensemble parameter sets required to repeat the experiments in this study are available upon request

20 to the corresponding author (Krista.Kemppinen@asu.edu).

**We added the following references to the reference list**:

Ridgwell, A.J., 2012. cGENIE v.0.9 ('muffin') User Manual. October 26, 2012 [PDF document]. Retrieved from http://www.seao2.info/cgenie/docs/cGENIE.User_manual.pdf.

University of Bristol Geography Source 2014, GENIE, available from <https://source.ggy.bris.ac.uk/wiki/GENIE>. [2014].

**2. Author contribution**

KMSK designed the experiments with guidance from PBH, NRE AND AR. KMSK carried them out and analysed the data. KMSK prepared the manuscript with contributions from all co-authors.

10 ## 3. Additional Appendix

We added the following comment in text, and the following appendix, for additional details about how to set-up the simulation ensemble.

**Comment in text**

P5, l15: (see Appendix A for further information).

15 ## Appendix A: the OLR feedback parameter

The ensemble parameter OL1 is varied through all stages but stage 1. OL1 describes the un-modelled response of clouds to global average temperature change, and corresponds to the $K_{LW1}$ constant in the equation below (eq. 1 in Holden et al., 2010a):

$$L^*_{out} = L_{out}(T,q) - K_{LW0} - K_{LW1}\Delta T$$

20 where $L_{out}(T,q)$ is the unmodified "clear skies" OLR term of Thompson and Warren (1982), $K_{LW0}$ is the clear-skies outgoing long-wave radiation parameter (OL0), representing the effects of clouds on the unmodified OLR (with $K_{LW0}$ only ever taking positive values), and $\Delta T$ corresponds to the difference between the globally averaged surface air temperature and the equilibrium pre-industrial temperature (Holden et al., 2010a).

25 In stage 1, OL1 is set to zero and $T_0$ can be set to any value. The temperatures simulated at the end of stage 1 are used to define $T_0$ in stage 2 and in the LGM simulations.

**We added the following reference to the reference list**:

Thompson S. L. and Warren S. G.: Parameterization of outgoing infrared radiation derived from detailed radiative calculations. J Atmos Sci 39:2667–2680. doi:10.1175/1520-0469(1982) 039\2667:POOIRD[2.0.CO;2, 1982.

**4. Removed a sentence**

The manuscript currently reads:

"These 10 kyr simulations are variously referred to here as the "LGM equilibrium simulation" or "stage 3", and the LGM equilibrium state refers to the end of stage 3 (see S1 for more details)."

**However, we note that S1 is not provided**. Based on cp-2017-159-author_response-version2.pdf, this was most likely (the page/line numbers seem off) added in response to the following comment:

> p. 7 l. 12: In the EFPC ensemble, are the simulations at equilibrium at the end of stage
> 3? What is meant here is probably the EFPC2 ensemble. We again have plots of atmospheric $CO_2$ and surface sediment %wt $CaCO_3$ for a subset of this ensemble (PGACF-16). Both metrics either in or nearing equilibrium by 10 kyr.

An excerpt from the MS in cp-2017-159-author_response-version2.pdf reads:

"The ensemble was integrated for another 10 kyr after stage 3 (yielding 20 kyr of LGM climate in total) to simply verify, by analyzing a subset of the ensemble, that the sediments (being the slowest component in the mode) were in equilibrium by 10 kyr. These next 10 kyr of LGM simulation are referred to as "stage 4".

**We removed "(see S1 for more details)" from the revised manuscript.**

An alternative to removing this sentence would have been to include the supplementary material below and reference it in the text. However, we did not deem this to be necessary and provide it here for information only.

Supplement S1

The ensemble was integrated for another 10 kyr after stage 3 (yielding 20 kyr of LGM climate in total) to simply verify, by analyzing a subset of the ensemble, that the sediments (being the slowest

component in the mode) were in equilibrium by 10 kyr. These next 10 kyr of LGM simulation are referred to as "stage 4".

[Figure]

[Figure]

**Fig. S1. Sediment surface CaCO₃ content (wt%) over stages 1 (0-10 kyr) to 4 (30-40 kyr) simulated by ENS16.** Individual ensemble members are represented by a hashtag and a number.

[revised manuscript text omitted]

---

## Author Response (AR5)

In the final round of revisions we included the explanation below for removing the sentence "(see S1 for more details)". We subsequently noticed that we had indeed already provided an S1. We, therefore, reintroduced the sentence in the final manuscript. We also noticed a typo in S2. Table S1.2 should say Table S2.1. We rectify this in the new supplement.

**4. Removed a sentence**

The manuscript currently reads:

"These 10 kyr simulations are variously referred to here as the "LGM equilibrium simulation" or "stage 3", and the LGM equilibrium state refers to the end of stage 3 (see S1 for more details)."

10  **However, we note that S1 is not provided**. Based on cp-2017-159-author_response-version2.pdf, this was most likely (the page/line numbers seem off) added in response to the following comment:

> p. 7 l. 12: In the EFPC ensemble, are the simulations at equilibrium at the end of stage 3? What is meant here is probably the EFPC2 ensemble. We again have plots of atmospheric $CO_2$ and surface sediment %wt $CaCO_3$ for a subset of this ensemble (PGACF-16). Both metrics either in or nearing equilibrium by 10 kyr.

An excerpt from the MS in cp-2017-159-author_response-version2.pdf reads:

"The ensemble was integrated for another 10 kyr after stage 3 (yielding 20 kyr of LGM climate in total)
15  to simply verify, by analyzing a subset of the ensemble, that the sediments (being the slowest component in the mode) were in equilibrium by 10 kyr. These next 10 kyr of LGM simulation are referred to as "stage 4".

**We removed "(see S1 for more details)" from the revised manuscript.**

An alternative to removing this sentence would have been to include the supplementary material below
20  and reference it in the text. However, we did not deem this to be necessary and provide it here for information only.

Supplement S1

The ensemble was integrated for another 10 kyr after stage 3 (yielding 20 kyr of LGM climate in total)
25  to simply verify, by analyzing a subset of the ensemble, that the sediments (being the slowest

component in the mode) were in equilibrium by 10 kyr. These next 10 kyr of LGM simulation are

referred to as "stage 4".

[Figure]

[Figure]

**Fig. S1. Sediment surface CaCO₃ content (wt%) over stages 1 (0-10 kyr) to 4 (30-40 kyr) simulated by ENS16.** Individual ensemble members are represented by a hashtag and a number.

[revised manuscript text omitted]